# Spectral attenuation coefficients from measurements of light transmission in bare ice on the Greenland Ice Sheet

Matthew G. Cooper[1,2], Laurence C. Smith[3,4,1], Asa K. Rennermalm[5], Marco Tedesco[6,7], Rohi Muthyala[5], Sasha Z. Leidman[5], Samiah E. Moustafa[3], Jessica V. Fayne[1]

[1]Department of Geography, University of California, Los Angeles, Los Angeles, California, 90027, USA
[2]Pacific Northwest National Laboratory, Richland, Washington, 99354, USA
[3]Institute at Brown for Environment and Society, Brown University, Providence, Rhode Island, 02912, USA
[4]Department of Earth, Environmental and Planetary Sciences, Brown University, Providence, Rhode Island, 02912, USA
[5]Department of Geography, Rutgers, The State University of New Jersey, New Brunswick, New Jersey, 08854, USA
[6]NASA Goddard Institute for Space Studies, New York, New York, 10025, USA
[7]Lamont Doherty Earth Observatory, Columbia University, New York, New York, 10964, USA

*Correspondence to*: Matthew G. Cooper (guycooper@ucla.edu)

**Abstract.** Light transmission into bare glacial ice affects surface energy balance, biophotochemistry, and light detection and ranging (LiDAR) laser elevation measurements but has not previously been reported for the Greenland Ice Sheet. We present measurements of spectral transmittance at 350–900 nm in bare glacial ice collected at a field site in the western Greenland ablation zone (67.15 °N, 50.02 °W). Empirical irradiance attenuation coefficients at 350–750 nm are ~0.9–8.0 m$^{-1}$ for ice at 12–124 cm depth. The absorption minimum is at ~390–397 nm, in agreement with snow transmission measurements in Antarctica and optical mapping of deep ice at the South Pole. From 350–530 nm, our empirical attenuation coefficients are nearly one order of magnitude larger than theoretical values for optically pure ice. The estimated absorption coefficient at 400 nm suggests the ice volume contained a light absorbing particle concentration equivalent to ~1–2 parts per billion black carbon, similar to pre-industrial values found in remote polar snow. The equivalent mineral dust concentration is ~300–600 ppb, similar to values for Northern Hemisphere warm periods with low aeolian activity inferred from ice cores. For a layer of quasi-granular white ice (weathering crust) extending from the surface to ~10 cm depth, attenuation coefficients are 1.5 to 4 times larger than for deeper bubbly ice. Owing to higher attenuation in this layer of near-surface granular ice, optical penetration depth at 532 nm is 14 cm (20%) lower than asymptotic attenuation lengths for optically pure bubbly ice. In addition to the traditional concept of light scattering on air bubbles, our results imply that the granular near-surface ice microstructure of weathering crust is an important control on radiative transfer in bare ice on the Greenland Ice Sheet ablation zone, and we provide new values of flux attenuation, absorption, and scattering coefficients to support model development and validation.

# 1 Introduction

Understanding the transmission, absorption, and scattering of light in ice is important for snow and ice energy balance modelling (Brandt and Warren, 1993), lidar remote sensing of snow surface elevation and grain size (Deems et al., 2013; Yang et al., 2017), primary productivity beneath sea ice (Frey et al., 2011; Grenfell, 1979), biophotochemistry in ice and snowpack (France et al., 2011), and theoretical predictions of "Snowball Earth" paleoclimates (Dadic et al., 2013; Warren et al., 2002). Each of these applications requires knowledge of the vertical distribution of light attenuation in ice, which for a medium (such as glacier ice) that both absorbs and scatters light is specified by the spectral attenuation coefficient:

$$k_{\mathrm{att}}(\lambda) = k_{\mathrm{abs}}(\lambda) + k_{\mathrm{sca}}(\lambda), \tag{1}$$

where $k_{\mathrm{abs}}$ (m⁻¹) is the absorption coefficient, $k_{\mathrm{sca}}$ (m⁻¹) is the scattering coefficient, and all are functions of wavelength, $\lambda$. This study reports on the irradiance attenuation coefficient $k_{\mathrm{att}}$ of bare glacier ice in the Greenland Ice Sheet ablation zone, a critical parameter needed to calculate subsurface absorption and scattering of transmitted radiation that to our knowledge has received no direct field study.

Measurements of $k_{\mathrm{att}}$ in snowpack and sea ice indicate three main sources of variation with relevance to geophysical applications. First, the magnitude of $k_{\mathrm{att}}$ is primarily controlled by ice microstructure (e.g., the size, shape, orientation, and number of air bubbles, ice grains, and cracks), via its control on $k_{\mathrm{sca}}$ (Dadic et al., 2013; Libois et al., 2013; Light et al., 2004, 2008). For the range of air bubble sizes ($\sim 10^{-3}$–$10^{-4}$) and ice grain sizes ($\sim 10^{-1}$–$10^{-3}$) observed in glacier ice, $k_{\mathrm{sca}}$ is effectively independent of wavelength in the visible and near-infrared spectrum (Bohren, 1983; Dadic et al., 2013; Perovich, 1996). Spectrally, $k_{\mathrm{att}}$ is low in the near-ultraviolet and blue-green ($\sim 250$–600 nm) where $k_{\mathrm{abs}}$ is extremely low ($< 10^{-8}$), and progressively higher for wavelengths $> 600$ nm, where $k_{\mathrm{abs}}$ rapidly increases up to its maximum value ($\sim 10^{-2}$) at the far end of the solar spectrum (Warren and Brandt, 2008). Vertically, $k_{\mathrm{att}}$ is at a maximum at the incident boundary (the snow or ice surface) where a portion of upwelling radiation (i.e., transmitted flux reflected upwards) escapes the ice volume before re-reflection downward. Within this near-surface optical boundary layer (Bohren and Barkstrom, 1974), attenuation rates rapidly decrease with depth to an asymptotic value as multiple scattering establishes an isotropic (diffuse) radiation field (Briegleb and Light, 2007; Warren, 1982). For fine-grained dry snow, a few cm depth is typically sufficient to reach the asymptotic regime where monochromatic $k_{\mathrm{att}}$ is constant (Brandt and Warren, 1993). For sea ice the depth required is typically larger and can exceed $> 20$ cm depending on near-surface ice microstructure and the vertical location of the refractive boundary if present (Grenfell, 1991; Grenfell and Maykut, 1977). Attenuation coefficients are also influenced by the horizontal distribution of ice type and surface cover (Frey et al., 2011) but this source of variation is not examined here.

In addition to experimental values obtained from measurements of light transmission in ice or snow, $k_{\mathrm{att}}$ is obtained analytically from optical theory (Bohren, 1987; Warren et al., 2006). Light attenuation in pure ice is specified analytically by the complex refractive index:

$$m(\lambda) = m_{\text{re}}(\lambda) - \text{i}\, m_{\text{im}}(\lambda), \tag{2}$$

where $m_{\text{re}}$ is the real part of the complex refractive index (~1.31 in the visible), $m_{\text{im}}$ is the imaginary part, and:

$$k_{\text{abs}}^{\text{ice}}(\lambda) = \frac{4\pi}{\lambda} m_{\text{im}}(\lambda) \tag{3}$$

is the absorption coefficient of pure ice (Warren et al., 2006; Warren and Brandt, 2008).

Light attenuation in glacier ice differs from pure ice owing to compositional and structural factors that control scattering and absorption, such as the size, geometry, and vertical distribution of embedded light absorbing particles (LAPs) and light scattering air bubbles and ice grains of size larger than wavelength (Askebjer et al., 1997; Picard et al., 2016; Price and Bergström, 1997b; Warren et al., 2006). Analytical methods typically approximate ice and snowpack as homogeneous plane-parallel slabs of spheres having the same volume-to-surface-area ratio (i.e., optically-equivalent grain size) as the collection of non-spherical ice grains and air bubbles in realistic ice (Brandt and Warren, 1993; Grenfell and Warren, 1999; Wiscombe and Warren, 1980). Mie theory is used to calculate the single-scattering properties and two-stream radiative transfer approximations are used to calculate multiple scattering and bulk absorption in the ice volume (Bohren and Barkstrom, 1974; Mullen and Warren, 1988; Wiscombe and Warren, 1980). The single-scattering properties can also be derived from the ratio of surface area to mass (i.e., specific surface area) with or without the assumption of spherical scattering geometry (Kokhanovsky and Zege, 2004; Malinka, 2014), as applied to the highly scattering granular surface layer on sea ice (Malinka et al., 2016). Models of the prior form have been used to calculate subsurface meltwater production caused by penetration of solar radiation in ice both in Greenland (van den Broeke et al., 2008; Kuipers Munneke et al., 2009) and Antarctica (Brandt and Warren, 1993; Hoffman et al., 2014; Liston et al., 1999a, 1999b; Liston and Winther, 2005). However, theoretical values for $k_{\text{att}}$ are rarely validated experimentally, and to our knowledge no such experimental values exist for near-surface glacier ice.

In addition to ice surface energy balance, understanding light attenuation in ice is important for interpreting interactions between visible-wavelength light sources and ice surfaces, for example laser altimetry measurements of ice surface elevation (Deems et al., 2013; Gardner et al., 2015; Greeley et al., 2017). The reciprocal of $k_{\text{att}}$ is the attenuation length, or the average distance travelled by a photon before attenuation by absorption or scattering (Ackermann et al., 2006). In the context of altimetry, the attenuation length is sometimes referred to as the penetration depth, or the average depth to which the electromagnetic signal penetrates before it is backscattered to the atmosphere (Ridley and Partington, 1988; Rignot et al., 2001; Zebker and Weber Hoen, 2000). The laser altimeter onboard Ice, Cloud, and Land Elevation Satellite-1 (ICESat-1) transmitted 1064 nm laser pulses to measure the distance (range) between the satellite and ice sheet surfaces (Schutz et al., 2005). Photons with wavelength 1064 nm penetrate snowpack no more than a few centimetres (Brandt and Warren, 1993; Järvinen and Leppäranta, 2013). This length scale is smaller than typical laser altimetry surface elevation errors due to ice and snow surface roughness and geolocation uncertainty (Deems et al., 2013). In contrast, the laser altimeter onboard ICESat-2 transmits 532

nm laser pulses (Markus et al., 2017). Ice is ~10 times more transparent at 532 nm than at 1064 nm (Warren and Brandt, 2008), and photons at 532 nm may penetrate many tens of centimetres into glacier ice. These subsurface scattered photons may introduce a range bias in ICESat-2 surface elevation retrievals over glacier ice, similar to radar penetration into snow (Brunt et al., 2016; Gardner et al., 2015; Greeley et al., 2017; Smith et al., 2018). To our knowledge no in situ observations of 532 nm optical penetration depth for bare glacier ice exist, precluding field validation of penetration depth obtained from theoretical radiative transfer models.

The purpose of this investigation is to provide experimental values for $k_{att}$ obtained from measurements of solar flux attenuation in bare ice in the Greenland Ice Sheet ablation zone, and to compare them with theoretical values for $k_{att}$ obtained from the two-stream analytical solution (c.f. Eq. 26 Bohren, 1987; Schuster, 1905). We benchmark our field estimates against the two-stream solution because of its wide use in surface energy balance models applied to snow and ice. In Sect. 2 we describe the field measurements and the optical theory used to interpret the solar flux attenuation. In Sect. 3 we report values for $k_{att}$ obtained from our measurements, compare them with values obtained from two-stream theory, and propose a simple empirical model that accounts for enhanced near-surface attenuation. In Sect. 4 we discuss how our $k_{att}$ values differ from prior experimental values acquired in sea ice, snowpack, and deep South Pole glacial ice, and the implication of these differences for modelling radiative transfer in bare glacier ice. To demonstrate the broader implications of our study, we suggest how our findings can be used to improve models for subsurface heating of ablating glacier ice.

## 2 Methods

### 2.1 Transmittance measurements

Ice transmittance was measured on 20 July and 21 July 2018 in the Kangerlussuaq sector of the western Greenland Ice Sheet. The study site is located ~1 km from the ice sheet margin at 840 m a.s.l. (67.15 ºN, 50.02 ºW). Subsurface (in-ice) spectral irradiance was measured at ~0.35 nm spectral resolution in the wavelength range 350–900 nm with an Ocean Optics® JAZ spectrometer. Light was guided from the ice interior to the spectrometer with a 3 mm diameter Kevlar-sheathed fibre optic cable fitted inside a 2 m long insulated white PVC tube (Fig. 1). The fibre was attached at one end to an irradiance sensor consisting of a 90º collimating lens adapter and a remote cosine receptor (RCR) with a Spectralon™ diffusing element. The RCR lens barrel was wrapped in white PTFE tape and set 2 mm out from the PVC tube exterior to act as a contact horizon between its diffusing element and the ice. The system was operated from a battery-powered computer running the Ocean Optics® OceanView software placed on a tripod platform oriented 180º away from the sun and 2.5 m horizontal distance from the measurement location.

To access the interior of the ice, holes were drilled horizontally into a 2-m high sidewall of a natural ice feature with a battery powered hand drill fitted with a 3 cm diameter Kovacs auger bit. To drill these holes, the auger was advanced into the sidewall

approximately 20 cm, levelled horizontally with a digital spirit level, and the sequence was repeated to 2 m horizontal depth.

The PVC tube-fibre optic assembly was then inserted into the hole, RCR facing upward, and a 2 m long ruler was shimmed under the bottom of the PVC tube to ensure the RCR barrel preserved contact with the overlying ice thus minimizing stray light contamination into the RCR field of view. Ice shavings were packed around the drill hole to prevent light reflection into the hole. Spectral irradiance was measured using a 20-scan average with 0.0228 s integration time per scan, yielding 0.46 s total integration time per irradiance measurement. Irradiance measurements were recorded at 1 Hz frequency for thirty seconds

yielding thirty irradiance profiles at each depth, after which the tube was removed, the next hole was drilled, and the sequence was repeated, working from the top hole toward the bottom on 20 July, and from the bottom hole toward the top on 21 July. The measurements were completed between 13:45 and 14:35 local time (UTC -3) on 20 July, and between 13:09 and 14:00 on 21 July, at solar zenith angles of ~48–51°. Solar noon at this time and location is ~13:26.

Background upwelling and downwelling spectral irradiances were measured continuously at 2 m height above the ice surface ~3 m away from the in-ice measurements with a dual-channel Ocean Optics JAZ spectrometer. These data were recorded at 1 min frequency using a 30-scan average with 0.011 s integration time. Light was guided to the spectrometer via two 3 m fibre optic cables attached to two RCRs mounted in upward-looking and downward-looking orientation on a 2 m long horizontally levelled boom attached to a vertical mast frozen into the ice. The horizontal boom became unstable on 21 July and the

upward-looking RCR was moved to the vertical mast; the downward-looking RCR was decommissioned.

  The surface-based spectrometer was calibrated for absolute irradiance in a controlled setting prior to the field experiment using an Ocean Optics HL-3P radiometrically calibrated halogen light source. During the field experiment, the in-ice spectrometer was cross-calibrated to the surface spectrometer by holding it level above the ice surface in an upward-looking orientation ~3

150  m away from the surface spectrometer. Cross-calibration irradiance profiles were collected on 20 July and 21 July immediately prior to subsequent in-ice measurements. All in-ice irradiances are cross-calibrated to the surface spectrometer as a pre-processing step prior to further analysis.

  Dark current spectra were recorded prior to each irradiance measurement as input to the OceanView automated dark current

correction module. To measure dark current, the RCR lens barrel was capped with a custom-fit opaque metal cap provided by Ocean Optics. OceanView adjusts these spectra in real-time for changes in integration time and for charge leakage if detected, corrects the nonlinear analogue-to-digital response of the linear silicon charge coupled device, and applies a boxcar smoothing over adjacent pixels to further reduce noise. Following these automated corrections, the opaque cap was left in place and residual dark current (noise) was recorded with the reference spectrometer in its identical setup during the experiment and with

the in-ice spectrometer held level above the ice surface in an upward-looking orientation. These residual dark current spectra are treated as systematic errors and are subtracted from all irradiance profiles as a pre-processing step prior to analysis (Fig. 2a).

## 2.2 Weather conditions

The 20 July experiment was conducted under low, thick cloud cover with light rain and no direct sun, ideal conditions for estimating the attenuation of diffuse light in ice. The 21 July experiment was conducted under higher, thinner cloud cover with no rain and very brief periods of intermittent direct sun (see Fig. 1). The effect of intermittent direct sun was easily identified in the in-ice irradiance measurements as a rapid increase in light intensity, which only occurred during the third measurement on 21 July. This was mitigated by averaging over the first ten in-ice irradiance pro files for that measurement, prior to the rapid increase in light intensity, and discarding the remainder.

## 2.3 Ice thickness and density

The ice thickness between detector positions was measured to the nearest millimetre with a metre stick and converted to units of solid ice thickness with the relation:

$$\Delta z = \Delta h \frac{\rho}{\rho_{ice}}, \tag{4}$$

where $\Delta h$ is in-situ ice thickness between detector positions, $\rho$ is in-situ ice density, $\rho_{ice}$ is solid ice density (917 kg m$^{-3}$), and $\Delta z$ is solid ice thickness between detector positions. Two separate observers made ten independent measurements of $\Delta h$. In addition, one observer made 41 replicate measurements of an ablation stake using the same metre stick, yielding a mean difference and standard error on $\Delta h$. The ice density $\rho$ was measured on a 1.2 m ice core extracted at the measurement location with a Kovacs Mark IV corer (www.kovacsicedrillingequipment.com) (Fig. 3). The ice core was split along natural breaks into three segments that were measured to the nearest millimetre with a calliper and weighed to the nearest gram on an Acculab digital scale.

## 2.4 Experimental asymptotic flux attenuation coefficients and ice surface albedo

Spectral asymptotic flux attenuation coefficients are estimated by fitting a Bouguer-law exponential decay model to the in-ice irradiance profiles (Grenfell and Maykut, 1977):

$$I_z(\lambda) = I_0(\lambda) \exp[-k_{att}(\lambda)(z - z_0)], \tag{5}$$

where $k_{att}(\lambda)$ is the asymptotic flux attenuation coefficient, $I_z(\lambda)$ is in-ice spectral irradiance at depth z, $I_0(\lambda)$ is background downwelling spectral irradiance, $z_0$ is the ice surface, and:

$$T_z(\lambda) = I_z(\lambda)/I_0(\lambda) \tag{6}$$

is spectral transmittance. The optical depth $\tau_z(\lambda)$ is a dimensionless path length that scales the physical thickness of a layer by its attenuation rate:

$$\tau_z(\lambda) = -\ln T_z(\lambda) = k_{att}(\lambda)(z - z_0). \tag{7}$$

Estimates of $k_{att}(\lambda)$ for each spectral band are obtained by solving a linear equation of the form:

$$\tau_z(\lambda) = \tau_0(\lambda) + k_{att}(\lambda)(\Delta z + \varepsilon_{\Delta z}) + \varepsilon_{\Delta \tau}, \tag{8}$$

where $\tau_0$ is a parameter (y-intercept), $\Delta z = z - z_0$ is ice thickness, $\varepsilon_{\Delta z}$ is an error term that represents ice thickness measurement uncertainty, and $\varepsilon_{\Delta \tau}$ is an error term that represents optical path measurement uncertainty. Equation 8 is solved by Maximum Likelihood Estimation (MLE), which gives an unbiased estimate of the slope when measurement errors are present in both the independent and dependent variables (see Sect. 2.9) (York et al., 2004).

The attenuation length $l_{att}(\lambda)$ is the inverse of $k_{att}(\lambda)$ and is analogous to the photon mean free path or transport length (Ackermann et al., 2006). It is equivalent to the path length in ice required to attenuate irradiance to 37% ($1/e$) of its incident intensity, i.e., the path length at which $T = 1/e$ and $\tau = 1$:

$$l_{att}(\lambda) = \frac{1}{k_{att}(\lambda)}. \tag{9}$$

The ice surface spectral albedo is the ratio of the upwelling spectral irradiance to the downwelling spectral irradiance:

$$\alpha(\lambda) = \frac{I\uparrow(\lambda)}{I\downarrow(\lambda)}, \tag{10}$$

and the broadband albedo is:

$$\alpha = \int_{\lambda_1}^{\lambda_2} \alpha(\lambda)\, I_0(\lambda)\, d\lambda \left/ \int_{\lambda_1}^{\lambda_2} I_0(\lambda)\, d\lambda \right. . \tag{11}$$

## 2.5 Asymptotic flux attenuation coefficients

Theoretical $k_{att}(\lambda)$ values are calculated using the asymptotic solution to the delta-Eddington two-stream radiative transfer approximation (Joseph et al., 1976; Schuster, 1905):

$$k_{att}(\lambda) = \frac{3}{4} \frac{Q_{ext}(\lambda)}{r_{eff}} \sqrt{3\big(1 - \overline{\omega}(\lambda)\big)\big(1 - g(\lambda)\overline{\omega}(\lambda)\big)}, \tag{12}$$

where $Q_{ext}(\lambda)$ is the extinction efficiency, $r_{eff}$ is the effective scattering particle radius (m), $g(\lambda)$ is the average cosine of the scattering angle, also referred to as the asymmetry parameter, and $\overline{\omega}(\lambda)$ is the single-scattering albedo:

$$\overline{\omega}(\lambda) = \frac{\sigma_{sca}(\lambda)}{\sigma_{att}(\lambda)}, \tag{13}$$

where $\sigma_{att}(\lambda)$ and $\sigma_{sca}(\lambda)$ are the single-scattering attenuation coefficient (m$^{-1}$) and scattering coefficient (m$^{-1}$), respectively. Equation (12) describes light attenuation by multiple scattering and absorption in a homogeneous plane-parallel slab of absorbing spheres far from any boundaries (Bohren, 1987).

To estimate $r_{eff}$, Eq. (12) is inverted and solved by iteration for the value of $r_{eff}$ that minimizes the difference between measured and calculated $k_{att}$ at $\lambda = 600$ nm. This method assumes that all absorption at 600 nm is due to ice (Warren et al., 2006). If absorption was influenced by LAPs $r_{eff}$ would be over-estimated. Values for $Q_{ext}(\lambda)$, $g(\lambda)$, and $\overline{\omega}(\lambda)$ are obtained

from Mie scattering algorithms provided as MATLAB® code (Mätzler, 2002) with input $m(\lambda)$ from Warren and Brandt (2008). The Mie solutions at each wavelength are integrated over a Gaussian size distribution (N=1000) of scattering radii $\mathcal{N}(\mu_r = r_{\text{eff}}, \sigma_r = 0.15r_{\text{eff}})$ to eliminate ripples associated with Bessel function solutions to the Mie equations (Gardner and Sharp, 2010). The optimal $r_{\text{eff}}$ values are ~9.3 mm and ~10.6 mm with corresponding specific surface areas ~0.35 m$^2$ kg$^{-1}$ and ~0.31 m$^2$ kg$^{-1}$ for 20 July and 21 July experimental values, respectively.

## 2.6 Flux absorption coefficients

Warren et al. (2006) developed a method to estimate $k_{\text{abs}}$ for pure ice (i.e. $k_{\text{abs}}^{\text{ice}}$) from measurements of flux attenuation in snow in Antarctica. The method relies on three assumptions: 1) the value of $k_{\text{abs}}^{\text{ice}}$ at the reference wavelength ($\lambda_0 = 600$ nm) is known accurately, 2) the value of $k_{\text{att}}$ at $\lambda_0$ is not affected by LAPs in the measured snow or ice, and 3) $\overline{\omega}(\lambda)$ varies so little as to be effectively independent of wavelength in the spectral range considered (here the near-UV and visible). Warren et al. (2006) verified the validity of these assumptions for the spectral range 350–600 nm and obtained the following relation (Eq. 15 of that paper) between flux attenuation and flux absorption:

$$\left[\frac{k_{\text{att}}(\lambda)}{k_{\text{att}}(\lambda_0)}\right]^2 \approx \left[\frac{k_{\text{abs}}(\lambda)}{k_{\text{abs}}(\lambda_0)}\right]. \tag{14}$$

Equation 14 assumes that $k_{\text{abs}}$ is not affected by LAPs at the reference wavelength (600 nm) but the relation can be used to estimate $k_{\text{abs}}$ at any other wavelength, including those where absorption is affected by LAPs. At those wavelengths, Eq. (14) will predict values for $k_{\text{abs}}$ higher than $k_{\text{abs}}^{\text{ice}}$ if LAPs are present in the measured snow or ice volume, due to the influence of LAPs on $k_{\text{att}}$.

The inferred $k_{\text{abs}}$ values can be related to a mass absorption cross-section (MAC) (Doherty et al., 2010):

$$k_{\text{abs}}(\lambda) = k_{\text{abs}}^{\text{ice}}(\lambda) + \beta c \rho_i, \tag{15}$$

where $\beta$ is the spectral MAC (m$^2$ kg$^{-1}$) and $c$ is the mass-mixing ratio of LAPs in the ice volume (g LAPs g$^{-1}$ ice). We exploit this to interpret differences between our theoretical and experimental values of $k_{\text{att}}$ on the basis of differences between $k_{\text{abs}}^{\text{ice}}$ (Warren et al., 2006) and the $k_{\text{abs}}$ values that we obtain for glacier ice from Eq. (14). To provide context, we use representative values of $\beta$ for black carbon $\beta_{\text{BC}}$ and insoluble mineral dust (hereafter 'dust') $\beta_{\text{dust}}$ to estimate corresponding equivalent mass mixing ratios $c_{\text{eq}}BC$ and $c_{\text{eq}}dust$ (Di Mauro et al., 2017; Doherty et al., 2010). The "equivalent" mass mixing ratio is the mass-mixing ratio of each LAP species required to explain the difference between $k_{\text{abs}}^{\text{ice}}$ and our inferred $k_{\text{abs}}$ values at a reference wavelength, and follows a similar approach used to infer LAP absorption in snowpack (Tuzet et al., 2019). For $\beta_{\text{BC}}$, we use 6 m$^2$ g$^{-1}$ as a representative MAC at 550 nm and an absorption Ångstrom exponent range 0.8–1.9 to scale this value to 400 nm (Doherty et al., 2010). For $\beta_{\text{dust}}$, we use 0.013 m$^2$ g$^{-1}$ at 550 nm (Di Mauro et al., 2017) and an absorption Ångstrom

exponent range 2–3 (Doherty et al., 2010). We note that these descriptive estimates provide context for discussion; actual LAP species concentrations were not measured.

## 2.7 Near surface effects

Equations 7 and 12 are applicable at distances far enough from the incident boundary (here the ice surface) that the radiation field is diffuse and $k_{att}$ is constant with depth. Near the ice surface the radiation field is converted via multiple scattering from direct to diffuse flux, and attenuation may be enhanced by direct reflection, enhanced scattering and/or absorption by the granular near-surface ice microstructure, or specular reflection at the ice surface, depending on its roughness (Dadic et al., 2013; Light et al., 2008; Mullen and Warren, 1988) . To account for non-diffuse near-surface attenuation, we define a piecewise optical depth:

$$\tau(\lambda) = \int_0^{z'} k'(\lambda) \, dz + \int_{z'}^z k_{att}(\lambda) \, dz, \tag{16}$$

where $k'$ is an effective attenuation coefficient for the near-surface non-diffuse layer and $z'$ is a depth chosen to partition this layer from the interior diffuse region. We estimate $k'$ with a centred finite-difference form of Eq. (7):

$$k'(\lambda) = -\frac{1}{\Delta z'} \ln\left[\frac{I_{z'}(\lambda)}{I_0(\lambda)}\right]. \tag{17}$$

Here, $\Delta z' = 12$ cm and $I_{z'}$ is the 12 cm in-ice irradiance measured on 20 July. Accordingly, the asymptotic attenuation length (Eq. 9) is distinguished from an effective penetration depth $d_\lambda$ to include the effect of near-surface attenuation. The attenuation length is the depth at which $\tau = 1$. Setting $\tau(\lambda) = 1$ in Eq. (16) and solving for z yields:

$$z = \frac{1 - \Delta z'[k'(\lambda) - k_{att}(\lambda)]}{k_{att}(\lambda)} = d_\lambda. \tag{18}$$

Equation 16 gives estimates of spectral transmittance that account for non-diffuse near-surface attenuation but relies on knowledge of $k'$, which is sensitive to the spectral composition and directional distribution of $I_0$ and the structure and composition of the near-surface ice (Grenfell and Maykut, 1977; Light et al., 2008). To generalize the magnitude of near-surface attenuation, we calculate the fraction of downwelling spectral irradiance that transmits the non-diffuse layer weighted by the downwelling spectral irradiance:

$$\chi_0 = \int_{\lambda_1}^{\lambda_2} I_0(\lambda) \exp[-k'_\lambda \Delta z'] \, d\lambda \Big/ \int_{\lambda_1}^{\lambda_2} I_0(\lambda) \, d\lambda. \tag{19}$$

The $\chi_0$ parameter is analogous to the $i_0$ parameter introduced by Grenfell and Maykut (1977) to partition the fraction of solar irradiance absorbed in the upper 10 cm of sea ice, which they termed the "Surface Scattering Layer" (SSL), and the ice interior, in which radiation is exponentially absorbed at a constant rate:

$$i_0 = \int_{\lambda_1}^{\lambda_2} [1 - \alpha_\lambda] \, I_0(\lambda) \exp[-k'_\lambda \Delta z'] \, d\lambda \, / \int_{\lambda_1}^{\lambda_2} [1 - \alpha_\lambda] \, I_0(\lambda) d\lambda \,. \tag{20}$$

The $i_0$ parameter has been widely adopted in energy balance models of glaciers and sea ice to compute subsurface flux divergence (heating rates) when radiation penetration is considered important (Bintanja and Van Den Broeke, 1995; Hoffman et al., 2014; Holland et al., 2012). For example, the sea ice component of the Community Earth System Model (CESM) uses $i_0 = 70\%$ for the visible (200–700 nm) and $i_0 = 0\%$ for the infrared (700–5000 nm) (Briegleb and Light, 2007). The important distinction is that $i_0$ partitions the absorbed flux whereas $\chi_0$ partitions the downward flux (Brandt and Warren, 1993). For both $\chi_0$ and $i_0$, we set $\Delta z'$ to 10 cm for consistency with prior work (Grenfell and Maykut, 1977; Light et al., 2008; Maykut and Untersteiner, 1971).

## 2.8 Monte Carlo simulations of detector interference

We developed a Monte Carlo radiative transfer model to estimate the effect of detector interference on measured irradiances and fitted $k_{att}$ values, following methods developed to simulate light propagation in biological tissue, ocean waters, and sea ice (Leathers et al., 2004; Light et al., 2003; Wang et al., 1995). Photon scattering is specified by a Henyey-Greenstein scattering phase function with single-scattering properties $Q_{ext}(\lambda)$, $g(\lambda)$, and $\overline{\omega}(\lambda)$ inferred from our optical measurements (Sect. 2.5). A complete technical description is given in the Supplementary Material, where model accuracy is verified by comparison with benchmark solutions to the radiative transfer equation (van de Hulst, 1980).

In the Monte Carlo simulations, photons are launched from an irradiance sensor on a detector rod with dimensions identical to those reported in this study. In the ideal (baseline) simulation, photons originate from an isotropic point source and propagate through ice until they transmit the surface or are terminated by absorption. Detector interference is investigated by repeating the Monte Carlo with an ideal cosine source function describing the angular response to radiance of the RCR, and with a non-ideal (empirical) angular response function (Fig. 2), with and without scattering and absorption interference by the PVC detector rod. The detector rod albedo $\overline{\omega}_{rod} \approx 0.4$ is calculated from the absorption spectra of polyvinyl chloride (Zhang et al., 2020); scattering by the rod is assumed isotropic. The Monte Carlo is integrated over 10 000 interactions at nine wavelengths in 50 nm increments from 350 nm to 750 nm, allowing us to fit the wavelength dependence of the estimated systematic uncertainty in simulated $k_{att}$ values.

## 2.9 Uncertainty propagation

Unless stated otherwise, all statistical uncertainties reported in this paper are standard errors that correspond to 68% confidence intervals around the mean (Taylor and Kuyatt, 1994). For an individual measurement with standard deviation $s_i$ and sample size $N \geq 30$ the standard error is $s_i/\sqrt{N}$. For $N < 30$, standard errors are scaled by a critical t-value drawn from the Student's

t-distribution. Standard errors for combined quantities are propagated in quadrature and hereafter referred to as combined uncertainty. The combined uncertainty for spectral irradiance $I(\lambda)$ is:

$$\sigma_I = \sqrt{(\sigma_I^*)^2 + (\sigma_D)^2}, \tag{21}$$

where $\sigma_I^*$ is the standard deviation of the high-frequency irradiance spectra before dark-noise correction and $\sigma_D$ is the standard deviation of the high-frequency dark-noise spectra. An analogous procedure is used to estimate the combined uncertainty for calibrated irradiance. The combined uncertainty for spectral transmittance $T(\lambda)$ is:

$$\sigma_T = T \sqrt{\left(\frac{\sigma_{I_z}^{cal}}{I_z^{cal}}\right)^2 + \left(\frac{\sigma_{I_0}}{I_0}\right)^2}, \tag{22}$$

where $\sigma_{I_z}^{cal}$ and $\sigma_{I_0}$ are the combined uncertainties for calibrated in-ice irradiance and dark-noise corrected surface downwelling irradiance, respectively. The combined uncertainty for optical depth $\tau_\lambda$ is:

$$\sigma_\tau = \frac{\sigma_T}{T}, \tag{23}$$

and the combined uncertainty for $k_{att}$ is:

$$\sigma_k = \sqrt{(\sigma_{\Delta\tau})^2 + (\sigma_{\Delta z})^2}. \tag{24}$$

Equation 24 gives a first order description of $\sigma_k$ due to statistical propagation of measurement uncertainties, neglecting higher-order interaction terms. A description of the statistical uncertainty in fitted $k_{att}(\lambda)$ values is given by the MLE estimate of the regression slope of Eq. 8, which can be expressed in terms of an error model as:

$$\hat{t}_z(\lambda) = \tau_0(\lambda) + k_{att}(\lambda) (\Delta\hat{z} + \varepsilon_{\Delta z}) + \varepsilon_{\Delta\tau},$$

where $\hat{t}$ and $\Delta\hat{z}$ are the true but unobserved (due to measurement error) optical depth and ice thickness and $\varepsilon_{\Delta z} \sim \mathcal{N}(0, \sigma_{\Delta z}^2)$ and $\varepsilon_{\Delta\tau} \sim \mathcal{N}(0, \sigma_{\Delta\tau}^2)$ are normally distributed error terms. Unlike Ordinary Least Squares, MLE gives an unbiased estimate of the slope and standard error of a linear functional relationship between two variables measured with error (York et al., 2004). The method has been used in similar studies to infer optical coefficients (Zieger et al., 2011). The MLE standard errors for $k_{att}(\lambda)$ are adjusted for $N-2$ degrees of freedom with a two-sided t-statistic (Cantrell, 2008) and combined in quadrature with systematic uncertainty estimated from Monte Carlo simulation to estimate total combined uncertainty for reported $k_{att}(\lambda)$ values.

## 3 Results

### 3.1 Spectral transmittance

Four in-ice irradiance spectra were collected at 12 cm, 36 cm, 58 cm, and 77 cm depth below the ice surface on 20 July (hereafter referred to as Layer A) (Fig. 4a), and at 53 cm, 67 cm, 82 cm, and 124 cm on 21 July (hereafter referred to as Layer B). At all depths, spectral transmittance $T$ is maximum at 430 nm and maintains relatively stable and high values up to ~500 nm in the visible, beyond which $T$ decreases into the red end of the visible spectrum following the well-known exponential

increase in ice absorptivity (Fig. 4c). Maximum $T$ values vary from 78% at 12 cm to 45% at 77 cm. For wavelengths >500 nm, $T$ rapidly decreases both with wavelength and with depth; beyond ~800 nm nearly all incident light is attenuated within 36 cm of the ice surface, although substantial attenuation is apparent in the 12–36 cm depth region (Fig. 4b). The standard deviation of the 1 Hz raw data is <1 W m$^{-2}$ nm$^{-1}$ at all wavelengths, consistent with field observations of thick cloud cover and diffuse light conditions described in Sect. 2.6. Instrumental noise and high-frequency measurement variations propagate as $\pm 1.6\%$ uncertainty on $T$ for wavelengths between 400–600 nm, $\pm 1$–8% for wavelengths between 350–400 nm, where instrumental noise is higher, and $\pm 1$–12% uncertainty for wavelengths between 600–750 nm, where noise is higher and light levels are low.

## 3.2 Experimental flux attenuation coefficients and albedo

Fitted $k_{att}$ values for Layer A range from 0.98±0.17 m$^{-1}$ to 7.86±0.43 m$^{-1}$ for wavelengths between 350 and 750 nm  (Fig. 5a), with uncertainty bounds that represent combined statistical and systematic uncertainty (see Sect. 4 for a discussion of systematic error). These values correspond to attenuation lengths of 1.02±0.18 m to 0.13±0.007 m, respectively. Layer B $k_{att}$ values are ~12% lower than Layer A values at 350–500 nm and within 1% at 650 nm (see inset Fig. 6). For Layer A, the minimum in $k_{att}$ is at 390 nm, blue-shifted relative to the maximum in $T$ at 430 nm. For Layer B, the minimum is at 397 nm. The coefficient of determination (r$^2$) ranged from 0.96–1.0 (p<0.01), with a median value of 0.98, suggesting the data are described appropriately by the Bouguer-law exponential decay model up to ~700 nm, beyond which measured values of in-ice irradiance at 58 cm and 77 cm depth were too low to reliably fit $k_{att}$ values (see Fig. 4b and Fig. 5c). For Layer B values, low light levels prevented fits beyond ~650 nm.

Albedo spectra correspond closely to patterns in transmittance and attenuation (Fig. 5c). The near-UV and blue wavelengths that efficiently transmit ice mostly re-emerge as reflected light, owing to the extremely low values of ice absorptivity in the wavelength range 350–500 nm where albedo is maximum (Gardner and Sharp, 2010; He and Flanner, 2020; Warren et al., 2006). The maximum measured albedo value is 0.81±0.004 at 452 nm, further red shifted from the minimum in $k_{att}$ and the maximum in $T$. All three quantities have low variability near the minimum; albedo is 0.79 at 390 nm. The broadband albedo $\alpha$ (Eq. 11) for the 350–900 nm wavelength range is 0.70±0.006, which is high but not atypical for melting white ice under overcast skies (Bøggild et al., 2010).

## 3.3 Theoretical flux attenuation coefficients

Asymptotic $k_{att}$ values predicted by two-stream theory for optically-clean bubbly ice are nearly one order of magnitude lower than field estimates for wavelengths <500 nm, where very small concentrations of LAPs in the measured ice volume dominate absorption (compare dotted grey line to solid blue line, Fig. 6) (Warren et al., 2006). In contrast, field-estimates and two-stream theory converge at wavelengths >540 nm where absorption is dominated by grain-size effects (He et al., 2017; Libois et al.,

2013). The magnitude of inferred absorption enhancement in the visible due to LAPs (the quantity $\beta c \rho_i$ in Eq. 15) varies from 0.009–0.015 m$^{-1}$ at 350–530 nm. The equivalent black carbon concentration $c_{eq}$BC inferred at 400 nm is 1–2 ng g$^{-1}$ for both Layer A and Layer B, where the range covers uncertainty in both the absorption spectra and the absorption Ångstrom exponent (Doherty et al., 2010). The equivalent mineral dust concentration $c_{eq}$dust is ~344–620 ng g$^{-1}$ for Layer A and 303–545 ng g$^{-1}$ for Layer B. Monte Carlo simulations without detector interference replicate both asymptotic theory for clean bubbly ice (i.e., when forced with $k_{abs}^{ice}$) and field estimates when forced with $k_{abs}$ values inferred from our optical measurements (solid blue squares, Fig. 6). Monte Carlo simulations of detector interference are discussed further in Sect. 4.

## 3.4 Near-surface attenuation and effective penetration depth

Near the ice surface irradiance is not attenuated exponentially and Bouguer's law does not hold, as indicated by the y-intercepts of the straight lines in Fig. 5b at values <100%. Effective $k'$ values (Eq. 17) for the quasi-granular 0–12 cm layer are ~1.5 times higher than $k_{att}$ values for interior bubbly ice at 12–77 cm depth for wavelengths >570 nm and are up to 4 times higher between 400–570 nm (Fig. 7). Owing to higher near-surface attenuation, transmitted irradiance $I_z$ is overestimated by 10–60% if Bouguer's law is applied to the incident downwelling irradiance $I_0$ using asymptotic $k_{att}$ values, with median over-estimation 23% (Fig. 8a). In contrast, the piecewise optical depth (Eq. 16) predicts $I_z$ to within 12% of measured values for all wavelengths between 350–700 nm with median error 3%. Integrated over these wavelengths, $\chi_0$ is 0.68 and $i_0$ is 0.66, suggesting 66% of the total incoming irradiance within this spectral region is absorbed at depths below 10 cm. If $k_{att}$ is used rather than $k'$ to calculate $\chi_0$ and $i_0$, the respective values are 0.81 and 0.79.

Stated in terms of penetration depth, $d_{eff}$ varies from 12–84 cm between 350–700 nm. These values are 13–44% lower than attenuation lengths $l_{att}$ inferred from empirical asymptotic $k_{att}$ values. Specifically at 532 nm, $d_{eff}$ is 52 cm, or 10 cm lower than the 62 cm empirical $l_{att}$ value, and 14 cm lower than the 66 cm theoretical $l_{att}$ value for optically pure bubbly ice. These results point to the potential for reduced optical penetration due to enhanced scattering and absorption on or near the ice surface, as well as within the ice volume where small LAP concentrations reduce optical backscattering due to enhanced absorption.

For smooth ice surfaces, attenuation may be enhanced by refraction at the ice-air interface (Mullen and Warren, 1988). If present, a refractive boundary would enhance near-surface attenuation via external specular reflection, and possibly via enhanced near-surface absorption of the internally reflected downward flux. Following Briegleb and Light (2007), Eq. 20–24, we calculate the external diffuse specular reflectivity for a flat ice surface to be 0.063, meaning specular reflection could enhance attenuation by up to 6.3%. This value is smaller than the 18–44% near-surface attenuation implied by the y-intercepts in Fig. 5b, suggesting specular reflection alone cannot explain the discrepancy. Instead, we suggest that enhanced scattering by the granular near-surface ice microstructure, together with absorptive impurities, enhance near-surface light attenuation at

395 our field site, consistent with observations of the granular and porous surface layer on sea ice (Grenfell and Maykut, 1977; Light et al., 2008).

## 4 Uncertainty analysis

The effect of random and systematic uncertainties on our optical measurements and fitted $k_{att}$ values is evaluated with Monte Carlo simulation and statistical analysis. We considered systematic uncertainties in detector positions, spectrometer sensitivity
to dark current, the non-ideal angular response of the irradiance sensor, and attenuation interference by the PVC detector rod.

The detector positions are known to within 0.9±0.4 cm from independent measurements of the vertical ice thickness Δh. The in-situ ice density $\rho$ varied from 801–888 kg m$^{-3}$ between 4–124 cm where irradiances were measured. The variation in $\rho$ was examined by repeating the analysis with Δz values computed with a single depth-weighted average $\rho$ applied to each Δh, and
405 with $\rho$ values estimated for each Δh from linear and cubic interpolation of the vertical density profile. The maximum Δz difference was 0.9 cm. The $k_{att}$ values differed by <1%, and r$^2$ values were nearly identical. We use the depth-weighted average $\rho$ values to calculate Δz, which are 835 kg m$^{-3}$ and 855 kg m$^{-3}$ for the measurements collected on 20 July and 21 July, respectively.

Detector position uncertainty was further assessed by fitting $k_{att}$ values with an ensemble of 10 000 Δz values perturbed with random errors drawn from a normal distribution $\mathcal{N}(\mu = 0.9 \text{ cm}, \sigma = 0.4 \text{ cm})$. At all wavelengths, the chance of obtaining a fitted $k_{att}$ value >2% from the mean value was <5%. We take 2% as a conservative estimate of systematic uncertainty due to ice thickness measurement bias.

As described in Sect. 3, all irradiance spectra are corrected for residual dark noise. The noise may have varied during the experiment, and dark noise measurements with the in-ice spectrometer were made on the surface, rather than within the ice. To assess possible bias, we fit $k_{att}$ values with and without residual dark noise correction. The mean difference was -0.01±0.13% averaged over the 350–700 nm wavelength range. For a few discrete wavelengths between 350–400 nm and 700–750 nm, differences approached 2%. These wavelengths are those with the highest dark noise in the reference
spectrometer (Fig. 2). At wavelengths between 400–700 nm, differences were <0.5%.

Monte Carlo simulations indicate a possible +1–14% systematic bias due to detector interference for Layer A values, and +2–8% for Layer B values (Fig. 9; also see purple stars minus solid squares, Fig. 6). The high end of this range applies to the wavelength region of minimum absorption ~350–450 nm. The simulated bias is within statistical uncertainty at wavelengths
>450 nm for Layer A and at wavelengths >400 nm for Layer B (Fig. 9). The non-ideal cosine response of the RCR and the presence of the detector rod both tend to increase $k_{att}$ values relative to the ideal case, as expected given the low albedo of the

detector rod. However, detector interference is masked somewhat by the presence of LAPs, as indicated by the larger simulated interference for bubbly ice without LAPs (see dotted grey line and associated Monte Carlo values, Fig. 6). Overall, the combined statistical and systematic uncertainty for the 350–450 nm region is <20% for Layer A values and <14% for Layer B
values, and as low as ~5% for wavelengths >450 nm (Fig. 9).

## 5 Discussion

### 5.1 Comparison with attenuation spectra for sea ice, snowpack, and deep glacial ice

We report spectral measurements of near-UV and visible light transmission in bare ablating glacier ice. These measurements are used to estimate irradiance attenuation coefficients $k_{att}$ for the spectral range 350–750 nm. Prior studies quantified $k_{att}$ for
sea ice and snowpack (c.f. Fisher et al., 2005; Frey et al., 2011; Gerland et al., 2000; Grenfell and Maykut, 1977; Järvinen and Leppäranta, 2013; King and Simpson, 2001; Light et al., 2008; Meirold-Mautner and Lehning, 2004; Pegau and Zaneveld, 2000; Picard et al., 2016; Tuzet et al., 2019; Warren et al., 2006). Scattering and absorption coefficients were quantified for compressed South Pole glacial ice at 800–2350 m depth by the AMANDA (Antarctic Muon and Neutrino Detector Array) experiment (Ackermann et al., 2006; Askebjer et al., 1995, 1997). For South Pole ice at 800–1000 m depth, visible and near-UV
light scatters on air bubbles, below which bubbles transition under pressure to non-scattering clathrates and light scatters on dust grains (Price and Bergström, 1997b). In the bubbly ice regime studied by AMANDA, $k_{sca}$ values at 532 nm are ~1–3 $m^{-1}$, comparable to the 1.6 $m^{-1}$ value quantified in this study. Light scattering in the dusty-ice regime (>1000 m depth) is not comparable to this study; absorption by dust is discussed below.

Fig. 10 compares our $k_{att}$ spectra for glacier ice to seven previously published spectra for snowpack and sea ice. In general,
glacier ice is the most transparent structure examined with the exception of multi-year and first-year interior sea ice at wavelengths >540 nm (Grenfell et al., 2006). Light attenuation in sea ice is controlled by its unique vertical composition including brine inclusions, air pockets, solid salts, sea ice algae, dissolved organic matter, water saturation, and radiative interactions between the ice and underlying ocean (Perovich, 1996). The latter factor, together with differences in
optically-equivalent grain size, may explain the low attenuation at longer wavelengths for sea ice shown here. Relative to snowpack in Greenland and Antarctica (Järvinen and Leppäranta, 2013; Meirold-Mautner and Lehning, 2004; Warren et al., 2006), attenuation by glacial ice has similar spectral structure but is lower at all wavelengths, reflecting the higher specific surface area of fine-grained polar snow. Attenuation within the surface scattering layer (SSL) of sea ice is intermediate, with spectral structure similar to snowpack and glacial ice. Attenuation at 5 cm depth in snow near Summit, Greenland is highest
of all, possibly due to direct light scattering in the near-surface optical boundary layer. The comparison demonstrates that $k_{att}$ values vary by nearly two orders of magnitude at visible wavelengths due to differences in ice structure and composition.

At visible wavelengths between 350–530 nm our field estimates of $k_{att}$ are up to one order of magnitude larger than those obtained from two-stream theory for optically pure bubbly ice, consistent with selective absorption by mineral dust, black carbon, and microorganisms found on glaciers and ice sheet surfaces (Bøggild et al., 2010; Ryan et al., 2018; Stibal et al., 2017; Takeuchi, 2002; Yallop et al., 2012). For context, the absorptivity we document at 400 nm for Layer B can be explained by 1.2–1.8 ng g$^{-1}$ (ppb) equivalent black carbon concentration. Values for Layer A are 1.4–2.0 ppb. Both estimates are relative to pure ice absorptivity values reported by Warren et al. (2006). These values are within the range 2$\pm$2 ppb reported for clean snow near Dye-2 on the interior Greenland Ice Sheet considered representative of pre-industrial fallout rates (Doherty et al., 2010). The equivalent mineral dust concentration is ~344–620 ppb for Layer A and 303–545 ppb for Layer B.

Relative to South Pole ice, our absorptivity values broadly agree with AMANDA values within two depth regions corresponding to peaks in atmospheric dust concentration during the Last Glacial Maximum (LGM) and Marine Isotope Stage 4 (MIS-4) glacial periods ~23 000 and ~66 000 years before present (Fig. 11). For these periods in Earth's history, Southern Hemisphere dust concentrations inferred from the Vostok and Dome-C ice cores are ~300–1500 ppb (Muhs, 2013; Petit et al., 1999). Hemispherical dust fluxes are generally synchronous at these timescales; similar peaks at LGM and MIS-4 are observed in Greenland ice cores (Ruth et al., 2003). However, Northern Hemisphere dust concentrations are several times higher (Muhs, 2013; Ruth et al., 2003), meaning correlation with South Pole absorptivity does not map age at our site. Rather, our optical measurements are consistent with the relatively low dust concentrations during Northern Hemisphere warm periods. For the western Greenland ablation zone, alternating bands of visibly dark and bright outcropping ice are associated with periods of higher and lower aeolian activity during both the early Holocene (post-LGM) and late Pleistocene, with a characteristic band of older brighter interglacial ice ~0.7–1 km from the margin where our field site is located (Bøggild et al., 2010; Petrenko et al., 2006; Reeh et al., 2002; Wientjes et al., 2012). Taken together, this suggests the optical properties documented here are representative of Pleistocene interglacial ice with relatively low volumetric LAP concentration and smaller crystal diameters than Holocene ice associated with the 'dark zone' further inland (Gow et al., 1997; Petrenko et al., 2006; Wientjes et al., 2011).

Regarding pure ice absorptivity, our $k_{abs}$ values provide support for the lower bound pure ice estimate from Warren et al. (2006) (Fig. 12). The steeply sloping high values in the near-UV in the laboratory measurements (Grenfell and Perovich, 1981; Perovich and Govoni, 1991) are now understood as signatures of Rayleigh scattering on nanoscale defects in the laboratory-grown ice (Price and Bergström, 1997a). The South Pole values at 1755 m depth and 830 m depth are contaminated by trace dust deposited during the late Pleistocene and early Holocene, respectively (Ackermann et al., 2006) (Fig. 11). The lowest values reported by Warren et al. (2006) (hereafter W06) were obtained by applying Eq. 7 to measurements of transmitted radiance in a single snow layer at ~90–135 cm depth near Dome C in Antarctica contaminated by ~0.6 ppb black carbon (75$^o$S, 123$^o$E, 3230 m). Picard et al. (2016) (hereafter P16) repeated the W06 experiment on 56 transmitted radiance profiles collected in snow near Dome C with variable impurity content. The values shown in Fig. 12 are their best estimate of pure ice absorptivity from radiance profiles collected in snow with low impurity content (see 'clean' subset, Fig. 17 of that paper). P16 was unable

to reconcile their values with W06, after considering published values for impurity loadings in the vicinity of Dome C, suggesting the W06 values were unreasonably low. Regardless of that discrepancy, our values were undoubtedly influenced by LAPs but are lower than the P16 values. Treating our Monte Carlo estimate of detector interference as a known systematic error would bring our values closer to the AMANDA and W06 values and further from the P16 values (see dotted line, Fig. 12).

Our absorption minimum is at 390 nm for Layer A values and 397 nm for Layer B values, in agreement with W06 and AMANDA. The wavelength shift in the P16 absorption minimum (430 nm) is coincident with our maximum in transmittance. A similar shift is apparent in all attenuation coefficient spectra shown in Fig. 10 that report the surface as a reference horizon, but is absent in those that report an interior reference horizon. P16 used an interior reference horizon, excluding radiance measurements within 8 cm of the surface based on Monte Carlo simulations of detector interference and visual inspection of homogeneous attenuation zones, but their shifted minimum may indicate that radiance profiles in the near-UV and blue were impacted disproportionately by detector rod interference and/or other near-surface effects. The same effects may explain the spectral structure in our near-surface (0–12 cm) effective attenuation coefficient profile between 350–400 nm and its broad minimum between 430–490 nm (dotted line Fig. 10). Similar spectral structure is apparent in diffuse attenuation coefficients obtained in snowpack in the French Alps using the P16 method (c.f. Fig. 3b, Tuzet et al., 2019). Differences aside, our inferred absorption spectrum provides new insight into the magnitude of this fundamental but uncertain optical property, and supports a conclusion that the minimum is likely $<10^{-2}$ m$^{-1}$ and possibly lower (Ackermann et al., 2006; Picard et al., 2016; Warren et al., 2006).

In addition to the traditional concept of surface melt, visible light transmission provides an energy source for subsurface heating and internal melting of near-surface glacier ice (Cooper et al., 2018; Hoffman et al., 2014; Liston and Winther, 2005; Schuster, 2001). Prior estimates of subsurface meltwater production in bare ice used two-stream theory forced with values of $k_{abs}^{ice}$ to calculate $k_{att}$ and the absorbed solar flux as a function of depth below the ice surface in both Greenland and Antarctica (van den Broeke et al., 2008; Hoffman et al., 2014; Kuipers Munneke et al., 2009; Liston and Winther, 2005). The influence of LAPs on subsurface meltwater production has not been quantified to our knowledge and is beyond our scope, but our results suggest LAPs enhance subsurface energy absorption in ablating glacier ice, consistent with enhanced surface melt rates caused by LAPs distributed on bare ice surfaces and within snowpack (Bøggild et al., 1996; Goelles et al., 2015; Goelles and Bøggild, 2017; Tuzet et al., 2019). From a practical perspective, this suggests that $k_{abs}$ values for contaminated ice given here and snowpack elsewhere (Picard et al., 2016) could provide realistic input for radiative transfer models absent explicit knowledge of realistic LAP concentrations. In contrast, simulations that use the canonical $k_{abs}^{ice}$ values compiled in Warren and Brandt (2008) will likely underestimate light attenuation and misrepresent the distribution of subsurface absorbed flux unless LAP concentrations are otherwise accounted for.

## 5 Conclusion

We report first in-situ spectral measurements of near-UV and visible light transmission in near-surface bare glacial ice, collected at a field site in the western Greenland ablation zone on 20–21 July 2018. In general, our empirical irradiance attenuation coefficients are nearly one order of magnitude larger in the range 350–530 nm than predicted by asymptotic two-stream theory using canonical values for the absorption coefficient of pure ice (Warren and Brandt, 2008). The absorption minimum is 0.013–0.014 $\pm$ 0.003 m$^{-1}$ at 390–397 nm, implying absorption length scales of 69–77 m. The volumetric scattering coefficient is 1.6 $\pm$ 0.2 m$^{-1}$ at 532 nm, with an asymptotic attenuation length scale 0.62 $\pm$ 0.08 m. In addition to light scattering on air bubbles, we find that light attenuation is enhanced by a layer of quasi-granular white ice (weathering crust) that extends from the surface to ~10 cm depth at our field site. The effective penetration depth, which accounts for reduced optical transmission through this granular layer relative to deeper bubbly ice, is 0.52 $\pm$ 0.07 m at 532 nm. Our co-located measurements of transmittance and albedo suggest that about 34% of cloudy sky downwelling solar irradiance at 350–700 nm was absorbed within this upper 10 cm surface layer at this time and location, consistent with observations of the semi-granular surface layer on sea ice. The estimated absorption spectrum suggests equivalent black carbon and mineral dust concentrations consistent with pre-industrial and warm interglacial periods with low Northern Hemisphere aeolian activity, and therefore may provide a reasonable lower bound on volumetric absorption enhancement due to impurities embedded in outcropping glacial ice in the western Greenland ablation zone.

**Data availability**

The data are hosted by the Pangaea open access data repository (https://doi.pangaea.de/10.1594/PANGAEA.913508), and are provided as a supplement to this manuscript in Appendix 1.

**Author contribution**

M.G. Cooper and L.C. Smith designed the experiment. M.G. Cooper, A.K. Rennermalm, M. Tedesco, R. Muthyala, S.Z. Liedman, and S.E. Moustafa collected the field data. M.G. Cooper performed the data analysis and wrote the manuscript. L.C. Smith, A.K. Rennermalm, M. Tedesco, S.Z. Liedman, and J.V. Fayne edited the manuscript. The authors declare they have no conflict of interest.

**Acknowledgements**

This project was funded by the NASA Cryospheric Science Program (grants NNX14AH93G and 80NSSC19K0942) managed by Dr. Thomas P. Wagner and Dr. Thorsten Markus; and a graduate fellowship from the NASA Earth and Space Sciences Fellowship Program (grant 80NSSC17K0374) managed by Dr. Lin Chambers. Polar Field Services and Kangerlussuaq

International Science Support (KISS) provided field logistical support. The authors thank Steven G. Warren, Richard E. Brandt, and Ghislain Picard for advice on the experimental design. The authors thank the handling editor and two anonymous reviewers for thorough and constructive reviews of the manuscript.

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

**Figures**

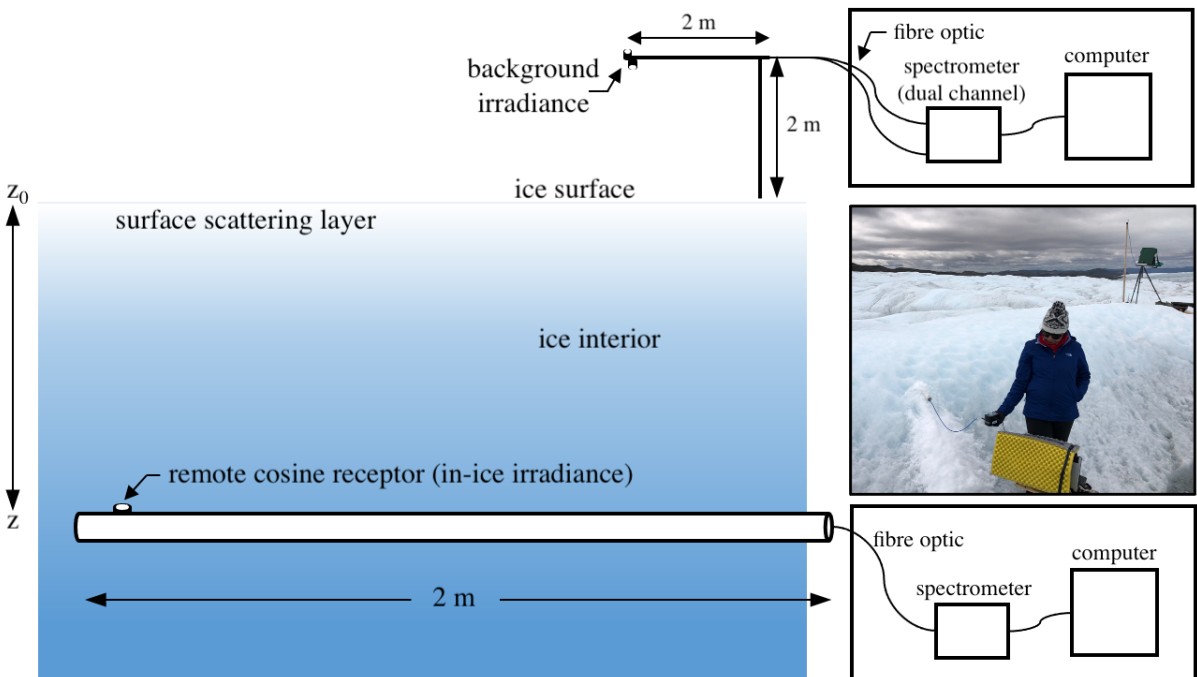

**Fig. 1: The irradiance sensor is comprised of a remote cosine receptor and fibre optic light guide fitted inside an insulated white PVC tube of 2 m length. Holes are drilled level and horizontal into the ice, the tube is inserted, and drill shavings are packed around the hole to prevent stray reflections. The cosine receptor collects the downwelling light, guides it to the fibre optic cable that transmits the light to an Ocean Optics® JAZ spectrometer, and a computer running the Ocean Optics® Ocean View software records the spectra. Background downwelling surface spectra are recorded on a 2 m mast drilled into the ice approximately 3 m to the northwest**
**of the in-ice measurement location (see photo background). This photograph was taken on 21 July 2018 at ~13:22 local time (UTC-3).**

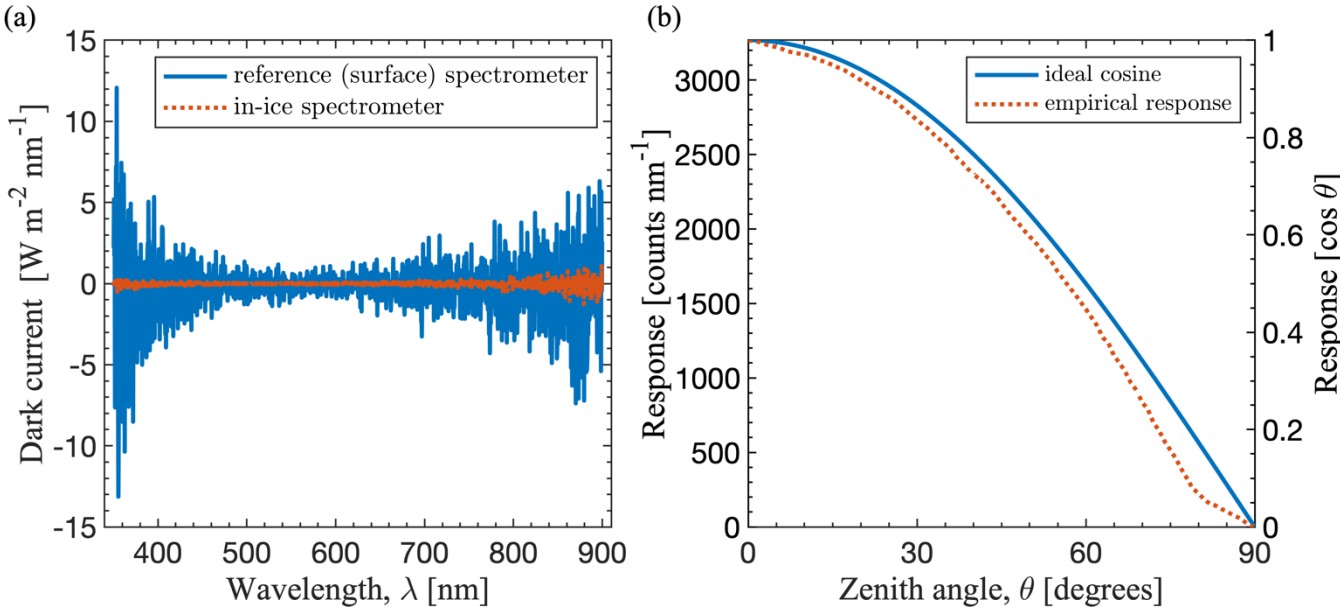

**Fig. 2: (a)** Residual dark-current spectra for the surface-based reference spectrometer and the in-ice spectrometer. Dark current spectra are recorded prior to each absolute irradiance measurement as input to the OceanView software dark-current correction module. Shown here are residual dark-current spectra after automated software correction, which are treated as systematic errors and subtracted from irradiance profiles prior to fitting experimental $k_{att}$ values. **(b)** Ideal angular response function (ideal cosine) and empirical angular response function provided by Ocean Optics from laboratory measurements on the same type of irradiance sensor used in this study. The dashed red line in (b) is used as an empirical probability density function for the angular response of the cosine receptor in our Monte Carlo simulations of detector interference.

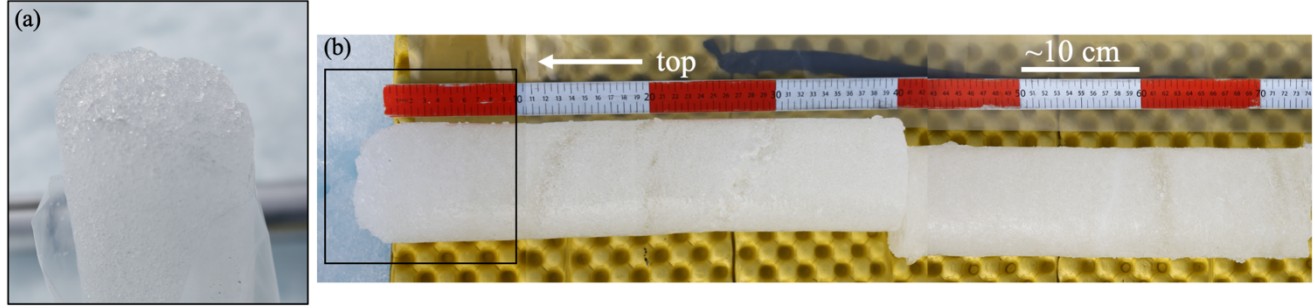

**Fig. 3: Photographs of an ice core collected at the field site. (a) The upper few centimetres of ice is semi-granular, with ~4 cm of unrecovered granular ice not shown. (b) The 122 cm ice core was broken into three segments corresponding to depths of 4–45 cm, 45–74 cm, and 74–122 cm below the ice surface (the far right of the image in (b) is at 74 cm). The density of these segments is 801 kg m$^{-3}$, 884 kg m$^{-3}$, and 888 kg m$^{-3}$, respectively. Below ~10 cm, the bubbly ice appears foliated, indicating variations in bubble density and size distribution that affect scattering. Black box in (b) is approximately the image area in (a).**

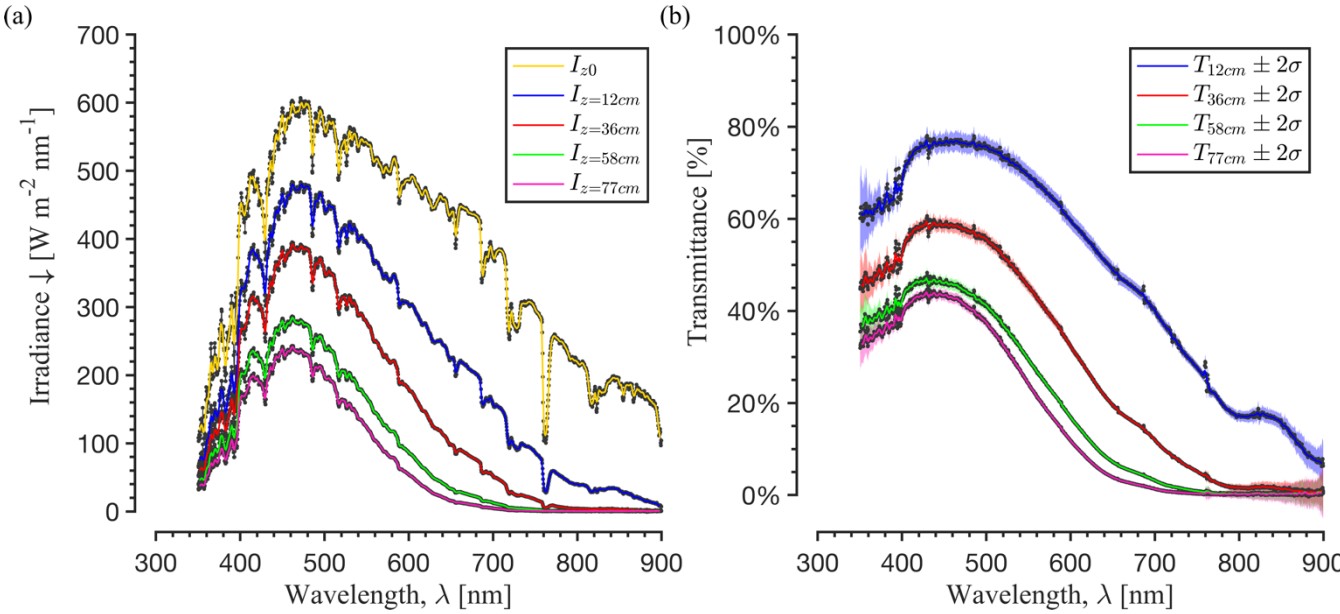

Fig. 4: (a) Field spectra of surface downwelling (z = z0) and in-ice irradiance at four depths below the ice surface collected on 20 July 2018 (Layer A) between 13:45 and 14:35 local time in the western Greenland ablation zone (67.15 ºN, 50.02 ºW). Raw data were recorded at 1 Hz frequency for 30 seconds, yielding 30 irradiance profiles at each depth. Shown here are 30-second averages at ~0.35 nm spectral resolution for each depth (black dots), and 1-nm interpolated values smoothed with a 3-nm centred moving mean filter for clarity (continuous lines). (b) Transmittance at each depth, with 30-second averages (black dots), 1-nm interpolated values (continuous lines), and shaded bounds ($\pm 2\sigma$) representing propagated measurement uncertainty deduced from the standard deviations of the 1 Hz raw data (N=30 for each value). Results for the 21 July 2018 (Layer B) experiment (not shown) are qualitatively similar.

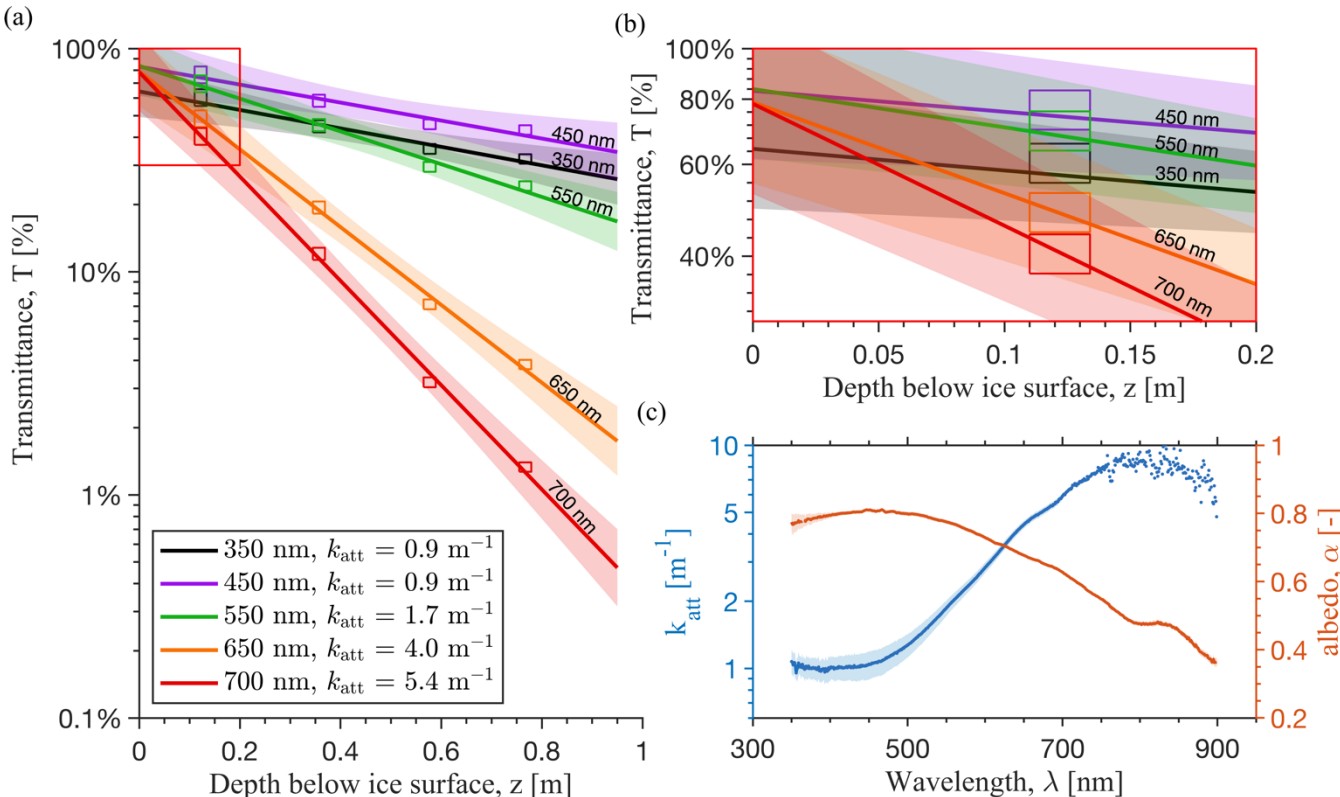

Fig. 5: (a) Sample linear regressions between measured transmittance (indicated by rectangles with width and height proportional to measurement uncertainty in both variables) and depth in the range 12–77 cm (Layer A) at five representative wavelengths spanning the measured spectral range. The slope of each line is the attenuation coefficient $k_{att}$. Shaded bounds are $\pm 95\%$ confidence intervals from a two-sided t-distribution. (b) Red box inset in (a) shows the y-axis intercept of each regression is less than 100%, indicating the magnitude of deviation from Bouguer's law near the surface. (c) Spectral $k_{att}$ (blue dots with shaded uncertainty; left axis) and spectral albedo (red dots with shaded uncertainty; right axis). Beyond ~700 nm, in-ice transmitted irradiance is too low to reliably estimate $k_{att}$ (see Fig. 4a and Fig. 2c), as indicated by the increased scatter in $k_{att}$ values.

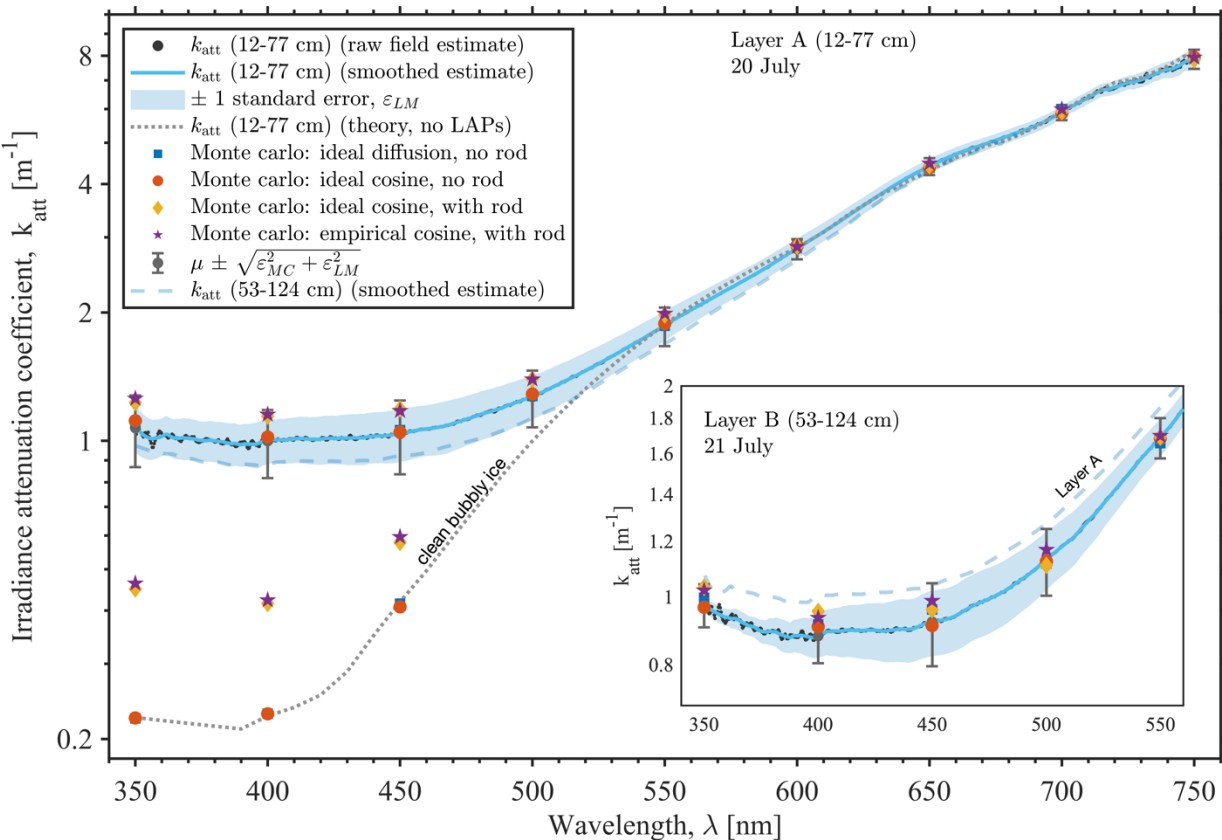

**Fig. 6: Visible and near-UV irradiance attenuation coefficient $k_{att}$ spectra from measurements of light transmission in bare glacier ice collected on 20 July 2018 at 12–77 cm depth below the ice sheet surface. Field estimates are compared with asymptotic two-stream theory for optically clean bubbly ice (continuous dotted line) and with values at nine wavelengths from four simulations with a 3-dimensional Monte Carlo radiative transfer model (solid symbols). Monte Carlo values for clean bubbly ice are shown for 350,**
**400, and 450 nm to demonstrate detector interference at these wavelengths; values at wavelengths >550 nm converge with field spectra and are omitted for clarity. Two measures of uncertainty are shown: 1) statistical linear model uncertainty $\varepsilon_{LM}$ (shaded uncertainty bounds; ±1 standard error in the linear regression) and, 2) $\varepsilon_{LM}$ combined with systematic uncertainty $\varepsilon_{MC}$ due to detector interference estimated with Monte Carlo (error bars; $\mu \pm \varepsilon$). The same comparison for the 21 July experiment (inset; Layer B) suggests detector interference is within statistical uncertainty at wavelengths >400 nm.**

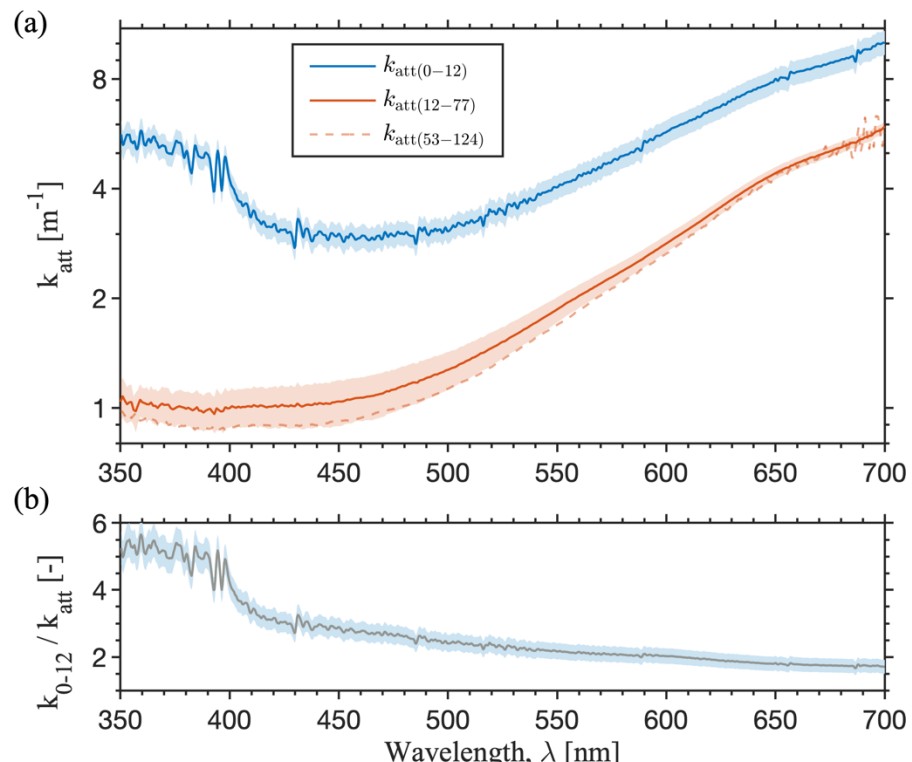

**Fig. 7: (a)** Effective attenuation coefficient $k'$ for the near-surface 0–12 cm region compared to $k_{att}$ values estimated for the interior 12–77 cm (Layer A) and 53–124 cm (Layer B) depth regions. **(b)** Effective $k'$ values are ~1.6 times larger than Layer A values at wavelengths beyond about 600 nm but are ~2–4 times larger between 400–600 nm. The spectral dependence suggests higher influence of absorptive impurities on attenuation enhancement near the ice surface than in the ice interior. The shaded bounds on $k'$ represent propagated $\pm 1.2$ cm vertical measurement uncertainty.

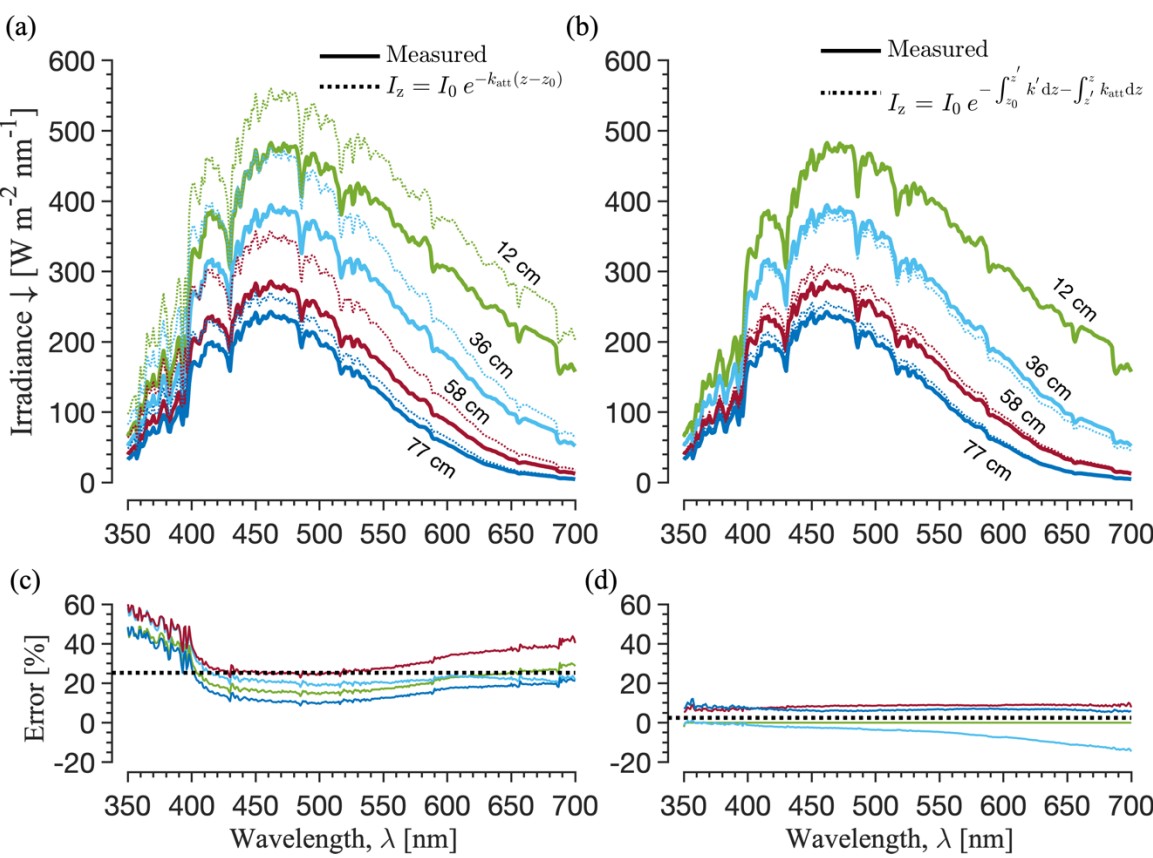

Fig. 8: **Measured in-ice irradiance compared with: (a) Bouguer's law (Eq. 7) with no modification, and (b) the piece-wise Bouguer law (Eq. 16). The error structure (c–d) provides insight into the near-surface attenuation processes: relative errors (%) are positive (model under-predicts attenuation) at all wavelengths but are highest in the near-UV, lowest in the blue, and increase monotonically into the red end of the visible spectrum. (d) errors are small, and generally decrease monotonically with increasing wavelength. Taken together, near-surface attenuation enhancement is ~10–60%.**

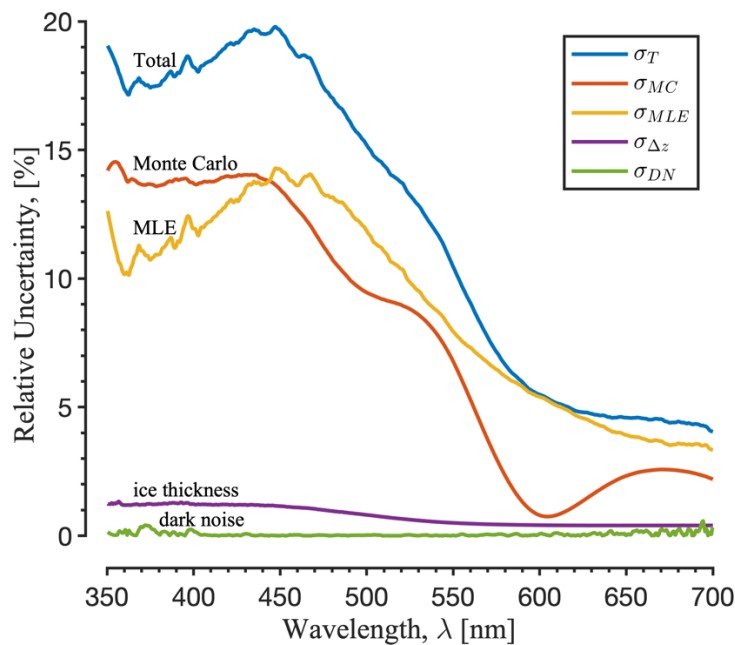

**Fig. 9: Uncertainty budget for reported asymptotic attenuation coefficient $k_{att}$ values for the 12–77 cm depth region (Layer A). Systematic uncertainties examined include spectrometer sensitivity to dark noise, ice thickness (detector position) measurement uncertainty, the non-ideal angular response of the irradiance sensor, and scattering and absorption interference by the polyvinyl chloride detector rod (estimated with Monte Carlo simulation). These systematics are combined with statistical uncertainty represented by one standard error in the Maximum Likelihood Estimate (MLE) linear regression fits. Values for Layer B (53–124 cm) (not shown) are qualitatively similar but lower, with total uncertainty <14% in the region of maximum uncertainty.**

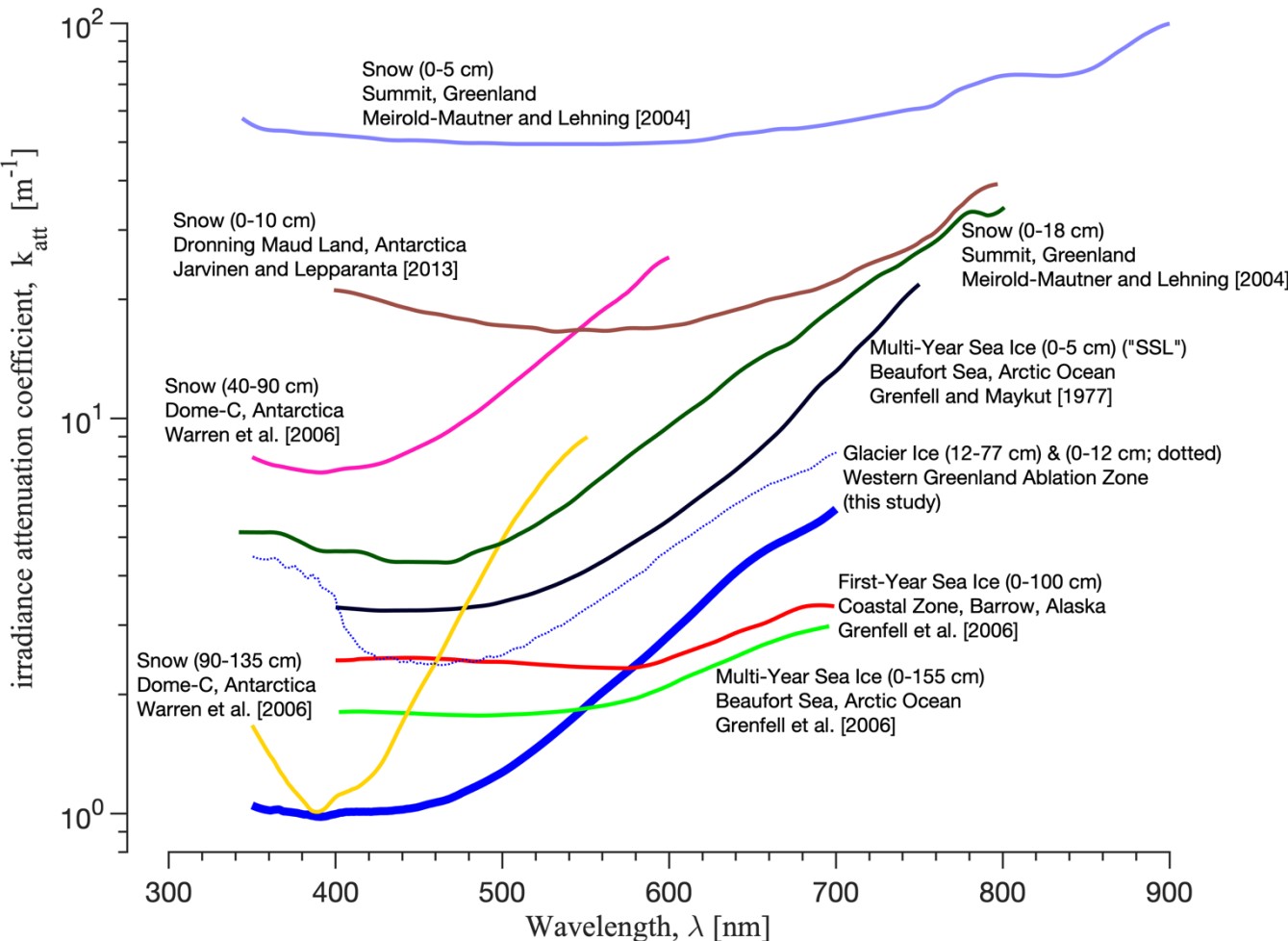

**Fig. 10: Attenuation coefficient spectra for seven distinct ice structures (from lower left clockwise): interior of clean, dry snowpack ($r_{eff} \approx 10^{-5}$ m) near Dome-C in Antarctica for two depth regions (90–135 cm and 40–90 cm) (Warren et al., 2006), near-surface (0–10 cm) dry snowpack ($r_{eff} \approx 10^{-3}$ m) in Dronning Maud Land, Antarctica (Järvinen and Leppäranta, 2013), near-surface (0–5 cm) and interior (0–18 cm) dry snowpack ($r_{eff} \approx 10^{-4}$ m) near Summit, Greenland (Meirold-Mautner and Lehning, 2004), surface scattering layer (SSL; 0–5 cm) of multi-year sea ice in the Arctic Ocean (Grenfell and Maykut, 1977), interior of ablating glacier ice in Greenland (this study) (12–77 cm in solid line; 0–12 cm in dotted line), interior of first-year sea ice in the coastal zone near Barrow, Alaska (Grenfell et al., 2006), and interior of multi-year sea ice in the Arctic Ocean (Grenfell et al., 2006). Differences in attenuation magnitude at each wavelength are mostly controlled by structural differences that control scattering, whereas spectral differences are mostly controlled by differences in type and concentration of absorbing impurities.**

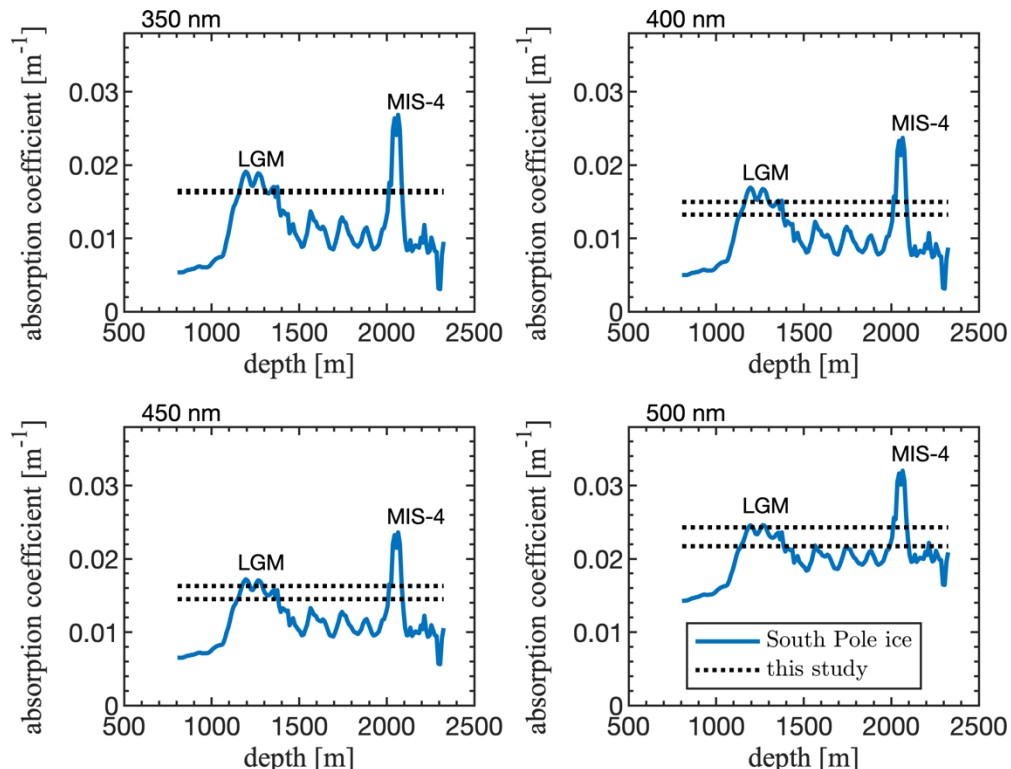

**Fig. 11: Depth profiles of South Pole ice absorption coefficient at four wavelengths obtained from Eq. 26 and Table 3 of Ackermann et al. (2006). The 1% K$^{-1}$ temperature dependence of pure ice absorptivity (Woschnagg and Price, 2001) is removed from South Pole values for comparison with this study's lower (Layer B) and upper (Layer A) absorption coefficient estimate at each wavelength (dashed lines). Values reported in this study are consistent with South Pole values at depths corresponding to the Last Glacial Maximum and Marine Isotope Stage 4 when atmospheric dust concentrations peaked in both hemispheres. Note that South Pole age and dust concentration do not map to ice near the Greenland Ice Sheet margin. Rather, Southern Hemisphere dust concentrations during LGM and MIS-4 are consistent with Northern Hemisphere dust concentrations during warm interglacial periods and/or periods with low aeolian activity (Muhs, 2013; Reeh et al., 2002).**

935

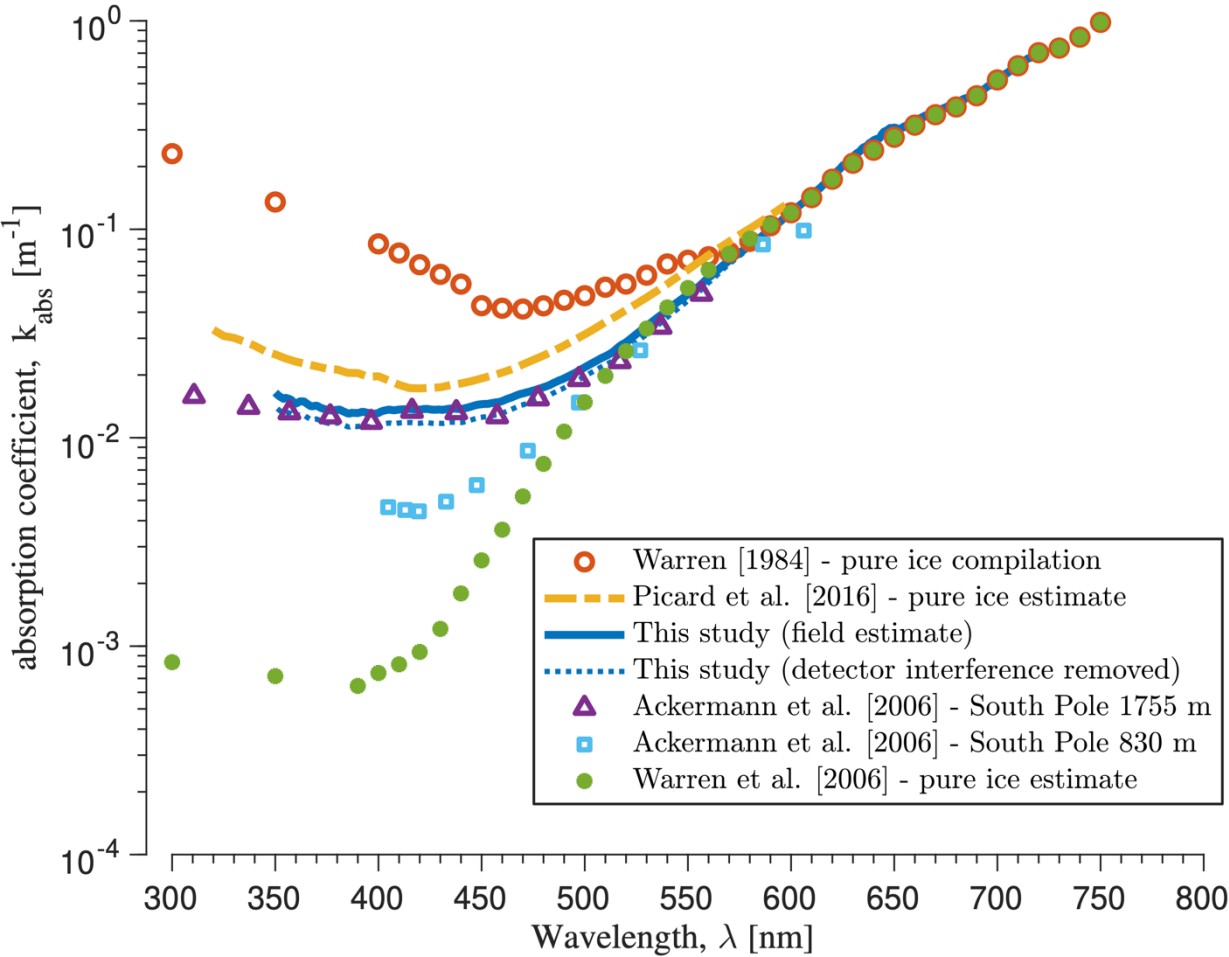

**Fig. 12: Estimates of ice absorption coefficient $k_{abs}$, obtained from five distinct sources: laboratory-grown pure ice (Grenfell and Perovich, 1981; Perovich and Govoni, 1991), as compiled in Warren (1984), snow in Antarctica, contaminated by trace concentrations of light absorbing particles (LAPs) (Picard et al., 2016), glacial ice in Greenland with unknown concentration of LAPs (this study), compressed glacial ice at 1755 m depth and 830 m depth in the Antarctic Ice Sheet contaminated by dust deposited during the late Pleistocene and Early Holocene, respectively (Ackermann et al., 2006), and snow in Antarctica with the effect of LAPs removed (pure ice estimate) (Warren et al., 2006). Values from this study with the detector interference subtracted are shown as dotted blue line.**

Table A2: Estimates of attenuation coefficient and absorption coefficient obtained from solar irradiance transmission measurements in glacier ice (Layer B; 53–124 cm below the ice sheet surface), one standard error in the linear regression, and coefficient of determination.

| wavelength (nm) | $k_{att}$ (m$^{-1}$) | standard error ($k_{att}$) | $r^2$ | $k_{abs}$ (m$^{-1}$) | standard error ($k_{abs}$) |
|---|---|---|---|---|---|
| 350 | 0.975 | 0.021 | 0.999 | 0.0162 | 0.00035 |
| 351 | 0.970 | 0.022 | 0.999 | 0.0160 | 0.00037 |
| 352 | 0.965 | 0.023 | 0.999 | 0.0159 | 0.00038 |
| 353 | 0.960 | 0.024 | 0.999 | 0.0157 | 0.00039 |
| 354 | 0.955 | 0.025 | 0.999 | 0.0155 | 0.00040 |
| 355 | 0.951 | 0.025 | 0.997 | 0.0154 | 0.00041 |
| 356 | 0.947 | 0.026 | 0.998 | 0.0153 | 0.00042 |
| 357 | 0.943 | 0.027 | 0.999 | 0.0152 | 0.00043 |
| 358 | 0.940 | 0.026 | 0.999 | 0.0150 | 0.00041 |
| 359 | 0.936 | 0.026 | 0.999 | 0.0149 | 0.00041 |
| 360 | 0.935 | 0.026 | 0.997 | 0.0149 | 0.00041 |
| 361 | 0.933 | 0.027 | 0.998 | 0.0148 | 0.00042 |
| 362 | 0.931 | 0.029 | 0.999 | 0.0148 | 0.00046 |
| 363 | 0.931 | 0.030 | 0.997 | 0.0148 | 0.00048 |
| 364 | 0.931 | 0.031 | 0.997 | 0.0148 | 0.00049 |
| 365 | 0.927 | 0.031 | 0.998 | 0.0147 | 0.00049 |
| 366 | 0.923 | 0.031 | 0.997 | 0.0145 | 0.00048 |
| 367 | 0.919 | 0.032 | 0.998 | 0.0144 | 0.00050 |
| 368 | 0.915 | 0.031 | 0.997 | 0.0143 | 0.00049 |
| 369 | 0.912 | 0.030 | 0.998 | 0.0142 | 0.00046 |
| 370 | 0.909 | 0.029 | 0.998 | 0.0141 | 0.00045 |
| 371 | 0.908 | 0.029 | 0.998 | 0.0140 | 0.00045 |
| 372 | 0.905 | 0.030 | 0.997 | 0.0140 | 0.00047 |
| 373 | 0.901 | 0.031 | 0.999 | 0.0138 | 0.00048 |
| 374 | 0.897 | 0.033 | 0.998 | 0.0137 | 0.00050 |
| 375 | 0.894 | 0.034 | 0.996 | 0.0136 | 0.00052 |
| 376 | 0.892 | 0.035 | 0.996 | 0.0136 | 0.00053 |

| 377 | 0.892 | 0.035 | 0.997 | 0.0136 | 0.00053 |
| 378 | 0.892 | 0.038 | 0.996 | 0.0136 | 0.00057 |
| 379 | 0.893 | 0.039 | 0.996 | 0.0136 | 0.00060 |
| 380 | 0.892 | 0.040 | 0.997 | 0.0135 | 0.00061 |
| 381 | 0.892 | 0.042 | 0.997 | 0.0136 | 0.00064 |
| 382 | 0.890 | 0.044 | 0.994 | 0.0135 | 0.00067 |
| 383 | 0.886 | 0.045 | 0.993 | 0.0134 | 0.00069 |
| 384 | 0.883 | 0.046 | 0.995 | 0.0133 | 0.00070 |
| 385 | 0.880 | 0.047 | 0.994 | 0.0132 | 0.00070 |
| 386 | 0.878 | 0.046 | 0.995 | 0.0131 | 0.00069 |
| 387 | 0.876 | 0.045 | 0.995 | 0.0131 | 0.00067 |
| 388 | 0.876 | 0.044 | 0.995 | 0.0131 | 0.00066 |
| 389 | 0.879 | 0.046 | 0.995 | 0.0132 | 0.00069 |
| 390 | 0.881 | 0.047 | 0.995 | 0.0132 | 0.00071 |
| 391 | 0.880 | 0.048 | 0.994 | 0.0132 | 0.00073 |
| 392 | 0.881 | 0.050 | 0.993 | 0.0132 | 0.00076 |
| 393 | 0.881 | 0.052 | 0.992 | 0.0132 | 0.00078 |
| 394 | 0.880 | 0.054 | 0.993 | 0.0132 | 0.00081 |
| 395 | 0.878 | 0.055 | 0.992 | 0.0131 | 0.00082 |
| 396 | 0.877 | 0.056 | 0.991 | 0.0131 | 0.00083 |
| 397 | 0.875 | 0.056 | 0.991 | 0.0131 | 0.00084 |
| 398 | 0.875 | 0.057 | 0.992 | 0.0130 | 0.00085 |
| 399 | 0.877 | 0.059 | 0.992 | 0.0131 | 0.00088 |
| 400 | 0.881 | 0.060 | 0.991 | 0.0132 | 0.00091 |
| 401 | 0.883 | 0.062 | 0.990 | 0.0133 | 0.00093 |
| 402 | 0.886 | 0.064 | 0.990 | 0.0134 | 0.00096 |
| 403 | 0.890 | 0.066 | 0.988 | 0.0135 | 0.00100 |
| 404 | 0.892 | 0.067 | 0.988 | 0.0136 | 0.00102 |
| 405 | 0.892 | 0.068 | 0.988 | 0.0136 | 0.00104 |
| 406 | 0.893 | 0.069 | 0.988 | 0.0136 | 0.00105 |
| 407 | 0.894 | 0.070 | 0.989 | 0.0136 | 0.00107 |
| 408 | 0.894 | 0.071 | 0.988 | 0.0136 | 0.00108 |
| 409 | 0.893 | 0.072 | 0.987 | 0.0136 | 0.00110 |
| 410 | 0.894 | 0.074 | 0.987 | 0.0136 | 0.00112 |
| 411 | 0.895 | 0.075 | 0.986 | 0.0136 | 0.00114 |
| 412 | 0.896 | 0.075 | 0.986 | 0.0137 | 0.00115 |

| 413 | 0.897 | 0.076 | 0.985 | 0.0137 | 0.00116 |
| 414 | 0.897 | 0.076 | 0.985 | 0.0137 | 0.00117 |
| 415 | 0.897 | 0.077 | 0.987 | 0.0137 | 0.00118 |
| 416 | 0.896 | 0.077 | 0.987 | 0.0137 | 0.00118 |
| 417 | 0.896 | 0.078 | 0.984 | 0.0137 | 0.00119 |
| 418 | 0.896 | 0.079 | 0.983 | 0.0137 | 0.00120 |
| 419 | 0.895 | 0.080 | 0.985 | 0.0137 | 0.00122 |
| 420 | 0.896 | 0.081 | 0.984 | 0.0137 | 0.00124 |
| 421 | 0.896 | 0.082 | 0.984 | 0.0137 | 0.00125 |
| 422 | 0.897 | 0.082 | 0.983 | 0.0137 | 0.00125 |
| 423 | 0.897 | 0.082 | 0.983 | 0.0137 | 0.00126 |
| 424 | 0.897 | 0.083 | 0.983 | 0.0137 | 0.00127 |
| 425 | 0.896 | 0.084 | 0.984 | 0.0137 | 0.00128 |
| 426 | 0.896 | 0.084 | 0.982 | 0.0137 | 0.00129 |
| 427 | 0.895 | 0.085 | 0.982 | 0.0137 | 0.00130 |
| 428 | 0.894 | 0.085 | 0.983 | 0.0136 | 0.00130 |
| 429 | 0.893 | 0.086 | 0.982 | 0.0136 | 0.00131 |
| 430 | 0.894 | 0.088 | 0.980 | 0.0136 | 0.00133 |
| 431 | 0.895 | 0.088 | 0.982 | 0.0137 | 0.00134 |
| 432 | 0.896 | 0.089 | 0.981 | 0.0137 | 0.00135 |
| 433 | 0.897 | 0.090 | 0.979 | 0.0137 | 0.00137 |
| 434 | 0.898 | 0.091 | 0.980 | 0.0137 | 0.00139 |
| 435 | 0.899 | 0.091 | 0.981 | 0.0138 | 0.00140 |
| 436 | 0.900 | 0.092 | 0.980 | 0.0138 | 0.00141 |
| 437 | 0.900 | 0.093 | 0.978 | 0.0138 | 0.00142 |
| 438 | 0.899 | 0.093 | 0.979 | 0.0138 | 0.00142 |
| 439 | 0.899 | 0.093 | 0.979 | 0.0138 | 0.00142 |
| 440 | 0.899 | 0.093 | 0.979 | 0.0138 | 0.00143 |
| 441 | 0.900 | 0.093 | 0.980 | 0.0138 | 0.00142 |
| 442 | 0.901 | 0.093 | 0.979 | 0.0138 | 0.00143 |
| 443 | 0.904 | 0.093 | 0.979 | 0.0139 | 0.00144 |
| 444 | 0.907 | 0.094 | 0.979 | 0.0140 | 0.00146 |
| 445 | 0.910 | 0.095 | 0.979 | 0.0141 | 0.00148 |
| 446 | 0.913 | 0.096 | 0.978 | 0.0142 | 0.00150 |
| 447 | 0.916 | 0.097 | 0.978 | 0.0143 | 0.00152 |
| 448 | 0.918 | 0.098 | 0.978 | 0.0144 | 0.00153 |

| | | | | | |
|---|---|---|---|---|---|
| 449 | 0.920 | 0.099 | 0.977 | 0.0144 | 0.00155 |
| 450 | 0.922 | 0.099 | 0.977 | 0.0145 | 0.00155 |
| 451 | 0.922 | 0.099 | 0.978 | 0.0145 | 0.00155 |
| 452 | 0.923 | 0.099 | 0.978 | 0.0145 | 0.00156 |
| 453 | 0.925 | 0.099 | 0.977 | 0.0146 | 0.00157 |
| 454 | 0.926 | 0.100 | 0.978 | 0.0146 | 0.00157 |
| 455 | 0.927 | 0.100 | 0.977 | 0.0146 | 0.00158 |
| 456 | 0.929 | 0.100 | 0.976 | 0.0147 | 0.00158 |
| 457 | 0.931 | 0.100 | 0.977 | 0.0148 | 0.00158 |
| 458 | 0.933 | 0.100 | 0.978 | 0.0148 | 0.00159 |
| 459 | 0.936 | 0.101 | 0.978 | 0.0149 | 0.00161 |
| 460 | 0.939 | 0.101 | 0.978 | 0.0150 | 0.00162 |
| 461 | 0.942 | 0.102 | 0.977 | 0.0151 | 0.00164 |
| 462 | 0.946 | 0.103 | 0.975 | 0.0153 | 0.00166 |
| 463 | 0.951 | 0.104 | 0.976 | 0.0154 | 0.00169 |
| 464 | 0.955 | 0.105 | 0.976 | 0.0155 | 0.00171 |
| 465 | 0.960 | 0.105 | 0.977 | 0.0157 | 0.00172 |
| 466 | 0.964 | 0.105 | 0.977 | 0.0158 | 0.00172 |
| 467 | 0.968 | 0.105 | 0.977 | 0.0160 | 0.00173 |
| 468 | 0.972 | 0.105 | 0.977 | 0.0161 | 0.00174 |
| 469 | 0.975 | 0.105 | 0.977 | 0.0162 | 0.00175 |
| 470 | 0.979 | 0.106 | 0.977 | 0.0163 | 0.00177 |
| 471 | 0.981 | 0.107 | 0.977 | 0.0164 | 0.00178 |
| 472 | 0.984 | 0.107 | 0.977 | 0.0165 | 0.00180 |
| 473 | 0.987 | 0.107 | 0.976 | 0.0166 | 0.00181 |
| 474 | 0.991 | 0.107 | 0.976 | 0.0167 | 0.00181 |
| 475 | 0.994 | 0.107 | 0.977 | 0.0168 | 0.00182 |
| 476 | 0.997 | 0.107 | 0.978 | 0.0169 | 0.00182 |
| 477 | 1.001 | 0.107 | 0.978 | 0.0171 | 0.00183 |
| 478 | 1.004 | 0.107 | 0.977 | 0.0172 | 0.00183 |
| 479 | 1.008 | 0.107 | 0.978 | 0.0173 | 0.00184 |
| 480 | 1.012 | 0.107 | 0.978 | 0.0175 | 0.00185 |
| 481 | 1.016 | 0.107 | 0.978 | 0.0176 | 0.00186 |
| 482 | 1.021 | 0.107 | 0.978 | 0.0178 | 0.00187 |
| 483 | 1.026 | 0.108 | 0.978 | 0.0179 | 0.00188 |
| 484 | 1.031 | 0.108 | 0.979 | 0.0181 | 0.00189 |

| | | | | | |
|---|---|---|---|---|---|
| 485 | 1.037 | 0.108 | 0.979 | 0.0183 | 0.00190 |
| 486 | 1.042 | 0.107 | 0.979 | 0.0185 | 0.00191 |
| 487 | 1.048 | 0.107 | 0.979 | 0.0187 | 0.00192 |
| 488 | 1.053 | 0.107 | 0.980 | 0.0189 | 0.00192 |
| 489 | 1.058 | 0.107 | 0.980 | 0.0191 | 0.00193 |
| 490 | 1.064 | 0.107 | 0.980 | 0.0193 | 0.00194 |
| 491 | 1.069 | 0.107 | 0.980 | 0.0195 | 0.00195 |
| 492 | 1.075 | 0.107 | 0.981 | 0.0197 | 0.00196 |
| 493 | 1.081 | 0.107 | 0.980 | 0.0199 | 0.00198 |
| 494 | 1.087 | 0.107 | 0.981 | 0.0202 | 0.00199 |
| 495 | 1.094 | 0.108 | 0.981 | 0.0204 | 0.00201 |
| 496 | 1.101 | 0.108 | 0.981 | 0.0207 | 0.00203 |
| 497 | 1.108 | 0.108 | 0.982 | 0.0209 | 0.00204 |
| 498 | 1.115 | 0.108 | 0.981 | 0.0212 | 0.00205 |
| 499 | 1.123 | 0.108 | 0.981 | 0.0215 | 0.00207 |
| 500 | 1.130 | 0.108 | 0.982 | 0.0218 | 0.00208 |
| 501 | 1.138 | 0.108 | 0.983 | 0.0221 | 0.00209 |
| 502 | 1.145 | 0.108 | 0.982 | 0.0223 | 0.00210 |
| 503 | 1.152 | 0.107 | 0.983 | 0.0226 | 0.00211 |
| 504 | 1.159 | 0.107 | 0.983 | 0.0229 | 0.00212 |
| 505 | 1.167 | 0.108 | 0.983 | 0.0232 | 0.00214 |
| 506 | 1.174 | 0.108 | 0.983 | 0.0235 | 0.00216 |
| 507 | 1.181 | 0.108 | 0.983 | 0.0238 | 0.00218 |
| 508 | 1.188 | 0.108 | 0.984 | 0.0240 | 0.00220 |
| 509 | 1.195 | 0.109 | 0.984 | 0.0243 | 0.00222 |
| 510 | 1.202 | 0.109 | 0.983 | 0.0246 | 0.00223 |
| 511 | 1.209 | 0.109 | 0.984 | 0.0249 | 0.00224 |
| 512 | 1.216 | 0.109 | 0.984 | 0.0252 | 0.00225 |
| 513 | 1.226 | 0.109 | 0.984 | 0.0256 | 0.00227 |
| 514 | 1.237 | 0.108 | 0.985 | 0.0261 | 0.00228 |
| 515 | 1.248 | 0.108 | 0.985 | 0.0265 | 0.00229 |
| 516 | 1.259 | 0.108 | 0.986 | 0.0270 | 0.00231 |
| 517 | 1.269 | 0.108 | 0.986 | 0.0274 | 0.00233 |
| 518 | 1.279 | 0.108 | 0.986 | 0.0279 | 0.00234 |
| 519 | 1.290 | 0.108 | 0.986 | 0.0283 | 0.00236 |
| 520 | 1.300 | 0.108 | 0.986 | 0.0288 | 0.00238 |

| 521 | 1.310 | 0.107 | 0.987 | 0.0292 | 0.00240 |
|-----|-------|-------|-------|--------|---------|
| 522 | 1.320 | 0.107 | 0.987 | 0.0297 | 0.00241 |
| 523 | 1.333 | 0.107 | 0.987 | 0.0303 | 0.00242 |
| 524 | 1.345 | 0.106 | 0.988 | 0.0308 | 0.00243 |
| 525 | 1.358 | 0.105 | 0.988 | 0.0314 | 0.00244 |
| 526 | 1.370 | 0.105 | 0.988 | 0.0320 | 0.00245 |
| 527 | 1.382 | 0.104 | 0.989 | 0.0326 | 0.00246 |
| 528 | 1.394 | 0.104 | 0.989 | 0.0331 | 0.00247 |
| 529 | 1.407 | 0.104 | 0.989 | 0.0337 | 0.00248 |
| 530 | 1.419 | 0.104 | 0.989 | 0.0343 | 0.00250 |
| 531 | 1.432 | 0.103 | 0.990 | 0.0349 | 0.00252 |
| 532 | 1.445 | 0.104 | 0.990 | 0.0356 | 0.00255 |
| 533 | 1.458 | 0.104 | 0.990 | 0.0362 | 0.00258 |
| 534 | 1.471 | 0.103 | 0.990 | 0.0369 | 0.00259 |
| 535 | 1.483 | 0.103 | 0.990 | 0.0375 | 0.00260 |
| 536 | 1.497 | 0.103 | 0.990 | 0.0382 | 0.00262 |
| 537 | 1.511 | 0.102 | 0.991 | 0.0389 | 0.00264 |
| 538 | 1.525 | 0.102 | 0.991 | 0.0396 | 0.00266 |
| 539 | 1.539 | 0.102 | 0.991 | 0.0404 | 0.00268 |
| 540 | 1.553 | 0.102 | 0.991 | 0.0411 | 0.00270 |
| 541 | 1.567 | 0.102 | 0.991 | 0.0419 | 0.00273 |
| 542 | 1.581 | 0.102 | 0.992 | 0.0426 | 0.00275 |
| 543 | 1.595 | 0.102 | 0.992 | 0.0433 | 0.00278 |
| 544 | 1.608 | 0.102 | 0.992 | 0.0440 | 0.00280 |
| 545 | 1.621 | 0.102 | 0.992 | 0.0448 | 0.00282 |
| 546 | 1.634 | 0.103 | 0.992 | 0.0455 | 0.00286 |
| 547 | 1.648 | 0.103 | 0.992 | 0.0463 | 0.00290 |
| 548 | 1.662 | 0.104 | 0.992 | 0.0471 | 0.00294 |
| 549 | 1.677 | 0.104 | 0.992 | 0.0479 | 0.00296 |
| 550 | 1.690 | 0.103 | 0.992 | 0.0487 | 0.00298 |
| 551 | 1.704 | 0.103 | 0.993 | 0.0495 | 0.00299 |
| 552 | 1.718 | 0.103 | 0.993 | 0.0503 | 0.00300 |
| 553 | 1.732 | 0.102 | 0.993 | 0.0511 | 0.00301 |
| 554 | 1.747 | 0.102 | 0.993 | 0.0520 | 0.00303 |
| 555 | 1.763 | 0.102 | 0.993 | 0.0530 | 0.00306 |
| 556 | 1.780 | 0.102 | 0.993 | 0.0540 | 0.00310 |

| | | | | | |
|---|---|---|---|---|---|
| 557 | 1.798 | 0.102 | 0.993 | 0.0551 | 0.00313 |
| 558 | 1.817 | 0.102 | 0.994 | 0.0562 | 0.00316 |
| 559 | 1.835 | 0.102 | 0.994 | 0.0574 | 0.00319 |
| 560 | 1.853 | 0.102 | 0.994 | 0.0585 | 0.00321 |
| 561 | 1.870 | 0.101 | 0.994 | 0.0596 | 0.00322 |
| 562 | 1.888 | 0.101 | 0.994 | 0.0607 | 0.00324 |
| 563 | 1.906 | 0.100 | 0.995 | 0.0619 | 0.00326 |
| 564 | 1.925 | 0.100 | 0.995 | 0.0631 | 0.00328 |
| 565 | 1.944 | 0.099 | 0.995 | 0.0644 | 0.00328 |
| 566 | 1.964 | 0.099 | 0.995 | 0.0657 | 0.00331 |
| 567 | 1.983 | 0.098 | 0.995 | 0.0670 | 0.00333 |
| 568 | 2.001 | 0.098 | 0.995 | 0.0683 | 0.00335 |
| 569 | 2.019 | 0.098 | 0.995 | 0.0695 | 0.00337 |
| 570 | 2.037 | 0.098 | 0.995 | 0.0707 | 0.00338 |
| 571 | 2.055 | 0.097 | 0.995 | 0.0720 | 0.00341 |
| 572 | 2.072 | 0.097 | 0.996 | 0.0732 | 0.00343 |
| 573 | 2.090 | 0.097 | 0.996 | 0.0744 | 0.00347 |
| 574 | 2.108 | 0.098 | 0.996 | 0.0757 | 0.00350 |
| 575 | 2.126 | 0.097 | 0.996 | 0.0770 | 0.00353 |
| 576 | 2.143 | 0.098 | 0.996 | 0.0782 | 0.00357 |
| 577 | 2.160 | 0.098 | 0.996 | 0.0795 | 0.00360 |
| 578 | 2.178 | 0.097 | 0.996 | 0.0808 | 0.00362 |
| 579 | 2.197 | 0.097 | 0.996 | 0.0822 | 0.00364 |
| 580 | 2.216 | 0.097 | 0.996 | 0.0837 | 0.00366 |
| 581 | 2.237 | 0.097 | 0.996 | 0.0853 | 0.00370 |
| 582 | 2.257 | 0.097 | 0.996 | 0.0868 | 0.00373 |
| 583 | 2.276 | 0.097 | 0.996 | 0.0882 | 0.00376 |
| 584 | 2.296 | 0.097 | 0.996 | 0.0898 | 0.00379 |
| 585 | 2.317 | 0.097 | 0.996 | 0.0914 | 0.00382 |
| 586 | 2.337 | 0.097 | 0.997 | 0.0930 | 0.00388 |
| 587 | 2.358 | 0.097 | 0.997 | 0.0947 | 0.00391 |
| 588 | 2.380 | 0.097 | 0.997 | 0.0965 | 0.00393 |
| 589 | 2.402 | 0.097 | 0.997 | 0.0983 | 0.00395 |
| 590 | 2.424 | 0.096 | 0.997 | 0.1001 | 0.00398 |
| 591 | 2.446 | 0.096 | 0.997 | 0.1020 | 0.00400 |
| 592 | 2.468 | 0.096 | 0.997 | 0.1038 | 0.00402 |

| 593 | 2.489 | 0.096 | 0.997 | 0.1055 | 0.00405 |
| 594 | 2.512 | 0.095 | 0.997 | 0.1076 | 0.00407 |
| 595 | 2.536 | 0.095 | 0.997 | 0.1096 | 0.00412 |
| 596 | 2.560 | 0.096 | 0.997 | 0.1116 | 0.00417 |
| 597 | 2.583 | 0.096 | 0.997 | 0.1137 | 0.00421 |
| 598 | 2.606 | 0.096 | 0.997 | 0.1158 | 0.00426 |
| 599 | 2.630 | 0.096 | 0.997 | 0.1179 | 0.00431 |
| 600 | 2.653 | 0.096 | 0.997 | 0.1200 | 0.00435 |