# Peer review of "Spectral attenuation coefficients from measurements of light transmission in bare ice on the Greenland Ice Sheet"

_The Cryosphere, 2020_

## Referee Comment (RC1) · Anonymous Referee #1 · 30 May 2020

**General comments**

Within the manuscript, spectral measurements of light attenuation in Greenland Ice Sheet bare ice are presented. For this purpose, the authors employed spectral irradiance measurements between 350-900 nm wavelength at the surface and at four different depths below the ice surface and calculated the spectral transmittance within the ice. From this, spectral flux attenuation and absorption coefficients are derived and compared to previous studies.

[Figure]

The manuscript is clearly structured and the figures are of good quality, which helps to convey the arguments of the authors. The measurements of in-ice spectral transmittance are very valuable, and the author's efforts to put them into context and to identify possible future applications should be acknowledged. However, there are some aspects that need further focus in my opinion. After some general comments, the more specific comments and suggestions for technical corrections follow below.

In my point of view, the most pressing aspect is the lack of accounting for measurement uncertainties within the manuscript. So far, only statistical variations along the 30 measured irradiance spectra are considered, which need to be clearly separated from instrument uncertainties. However, the latter are not mentioned at all within the manuscript.

I suggest to include a new subsection within the 'Methods'-section that is devoted to instrument and measurement uncertainties. I understand the instrument is calibrated for irradiance measurements, but instrument errors such as the influence of dark and stray light, the wavelength calibration of the spectrometer, the non-ideal cosine response of the RCR diffuser, and the uncertainty of the absolute calibration (to get calibrated irradiance measurements) need to be quantified. Furthermore, as the transmittance is measured calculating the ratio of measured irradiance from two different instruments (the one in the ice, and the permanent one at the surface), differences between the two instruments need to be quantified (e.g., by means of a cross-calibration with the same light source). These instrumental errors need to be put in relation to uncertainties regarding the measurement setup (e.g., is the cosine receptor really in close contact with the ice during the in-ice irradiance measurements?) and the statistical variations as deduced from the 30 subsequently measured spectra.

This uncertainty analysis should eventually lead to vertical uncertainty bars in Figure 3, together with the horizontal uncertainty bars stemming from the depth measurements with the ruler (this second paragraph of Section 3.5 should also be moved to the new

uncertainty subsection in Section 'Methods'). The errors in the linear regression to derive the flux attenuation coefficient should consider both depth and transmittance uncertainty estimates and, eventually, should lead to an uncertainty range attributed to each $k_{att}$ value. A thorough treatment of the measurement uncertainties will definitely increase the value of the measurements for further applications.

Another remark is more structural and concerns the introduction of figures in the text: Some figures need to be described and explained in more detail within the text already. So far, some figures are mentioned for the first time in the text in brackets after an interpretation of the figure is done already. In contrast, the figure captions explain the figures in a lot of detail. I suggest providing the information of the figure captions already within the text of the manuscript.

It is definitely fair to point out that the presented measurements are a valuable contribution to the field as (to my knowledge) such experimental values do not exist for glacier ice. However, the authors should consider to not oversell their work in mentioning it many times throughout the manuscript (e.g, the title, and on Page 2 Line 41, P3 L72, P3 L89,...). In a somewhat related issue, the title reads a bit 'bulky' and would benefit from being shortened in my opinion. However, this is of course only something to consider for the authors. In addition, some shortcomings in explaining the applied methodology should be fixed in order to increase reading comprehension and reproducability of the measurements (see specific comments below).

**Specific comments**

**Introduction**

- Page 2 Line 46: at this point of the introduction, it is helpful to give some values

for typical ranges of air bubble and ice grain sizes.

- P3 L66: While it is true that analytical models typically assume spherical scatterers to calculate the inherent optical properties, the introduction should also point at different approaches and mention efforts by Kokhanovsky and Zege (2004) as well as Malinka (2014) and Malinka et al. (2016) which provide analytical solutions for single-scattering properties but model snow and ice grains as nonspherical.
  Kokhanovsky, A. A. and Zege, E. P.: Scattering optics of snow, Appl. Opt., 43, 1589–1602,
  https://doi.org/10.1364/AO.43.001589, 2004.
  Malinka, A.: Light scattering in porous materials: Geometrical optics and stereological approach, J. Quant. Spectrosc. Radiat. Transf., 141,3514–23, https://doi.org/10.1016/j.jqsrt.2014.02.022, 2014.
  Malinka, A., Zege, E., Heygster, G., and Istomina, L.: Reflective properties of white sea ice and snow, Cryosphere, 10, 2541–2557,https://doi.org/10.5194/tc-10-2541-2016, 2016.

**Methods**

- How did you make sure the cosine receptor is in direct contact with the ice to avoid another ice-air interface? Shimming the ruler underneath the PVC tube seems to help, but have you done any testing in that regard?

- P4 L123: This sentence reads a bit confusing to me and needs to be reformulated. Also: I don't understand the integration time given in Hertz - I think a conversion to an actual time period is useful here.

- I have some suggestions for Figure 1 that would help the reader in my opinion:

- the photograph of the measurement setup within Figure 1 should be enlarged (and maybe put to the right of the schematic) as, right now, it is a bit small.
- a horizontal line should clearly mark the ice surface, as due to the color gradient it is hard to distinguish from the end of the schematic.
- Can you indicate the other vertical positions of the transmittance measurements maybe with some dashed horizontal lines and then also draw an arrow indicating that the measurements were conducted from the bottom to the top?

- P5 L134: The weather situation during the measurements needs to be specified with respect to clouds, temperature, etc.

**Results**

- Figure 2:

  - It would be interesting to include the surface downwelling irradiance at $z_0$ into Figure 2a for comparison.
  - The unit for the standard deviation is missing in Figure 2b.
  - Figure caption: instead of naming it relative irradiance for the first time, I would name it Transmittance like in the rest of the manuscript.

- Figure 2b is not discussed in Section 3.1. Instead of stating in the Figure caption of Fig. 2b that the standard deviation is below 1 W m-2 nm-1 at all wavelengths, I would move this statement to the main text.

- P8 L216: Albedo is mentioned for the first time. Please explain at this point how it was calculated from the surface measurements.

- P8 L231-...: Warren et al. (2006) also show the snow transmission measurements before removing the absorption by impurities - wouldn't using these measurements lead to a similar discrepancy between the theory and field estimate for clean snow than for the glacier ice at smaller wavelengths? I suggest to use the uncorrected measurements for snow as well within Figure 4a, so that it is consistent with the glacier ice measurements.

- P9 L252: The derivation of $\chi$ as the root-mean-squared difference between measured and predicted transmitted irradiance should already be explained in the respective Section 2.5 in the Methods.

- Again P9 L252: I have a general question to the $\chi$-value you applied. As you state in the caption of Figure 5, 'the spectral dependence [of the relative error] suggests a contribution of absorption to near-surface attenuation enhancement'. This calls for a spectral value of $\chi$, and indeed you mention at P9 L253, 'weighted equally at all depths and all $\lambda$' indicating that you derived $\chi$ for each individual wavelength separately. Is this the case? If so, please state already in Equation (7) that $\chi$ is dependent on $\lambda$ (which is the main difference to $i_0$ in my understanding). However, I get confused with the last sentence of the paragraph on P9 L253: it reads as if you apply only one value of $\chi = 15\,\%$ to all wavelengths. Please clarify this in the text. The same applies to P12 L354.

- Figure 5: The second empirical model you use, applying $I(z_0) = I(z_{12\mathrm{cm}})$, seems to perform best as it is only applied in the isotropic region of transmittance. Looking at the formulas in Figs. 5b and 5c for $I_z$, one could think the best possible solution for $\chi$ would be such that $(1 - \chi) = I_{12\mathrm{cm}}$. Which is not possible in my understanding as the exponential part of the equation (namely the 'z') is different. To avoid this confusion, the equation in Fig. 5b needs to be adapted accordingly: $I_z = I_{12\mathrm{cm}} \exp\left[-k(z - 12\,\mathrm{cm})\right]$

- P9 L256: it is unclear to the reader how the effective $k_{\mathrm{att}}$-values are derived using

a finite-difference solution to Eq. (2) (also: contradictory, in the caption of Figure 6 it says Eq. 1)?

- P9 L260: please state how exactly the effective $k_{att}$-values were combined for calculation of the effective penetration depth, e.g. give an equation for that.

- P10 L280: Please clarify in more detail how the external diffuse specular reflectivity for a flat ice surface was calculated in this case.

- Section 3.5: The title 'Uncertainty analysis' is misleading. It is true that the second paragraph of this section is a valuable uncertainty estimate, that should already be part of an 'Uncertainty analysis' subsection of the Section 'Methods' (compare general comment). The first paragraph of 3.5 is well-placed at this point of the manuscript, but I suggest to rename the subsection to e.g. 'Influence of ice density' after moving the second paragraph to the 'Methods'-section.

- Figure 9: I suggest including an additional subsection that compares the $k_{abs}$-values of this study with previous estimates. The first time the authors mention Figure 9 is in the 'Suggestions for further work' part, which definitely undersells this comparison. This is also a very specific case for what I was mentioning in the 'General comments' section: the Figure is mostly described and discussed in the Figure caption and not in the text at all. This should be changed.

- P13 L394: The 'Further work ...'-part of the last sentence is not useful in my opinion, as Section 4.4 already gives suggestions for future studies. I would end the 'Conclusions' section with the new values of attenuation and absorption coefficients that are provided in this study.

- Data availability: please provide the doi of the published dataset in Pangaea.

**Technical corrections**

- P2 Eq. (1): please indicate the spectral dependence already within the equation.

- P2 L47: please make sure the exponents are not split up at a linebreak.

- P2 L61: as you specify the spectral dependence of $m$, you should include it also for $m_{re}$ and $m_{im}$. In addition, naming them the real part and imaginary part of the complex index of refraction seems more appropriate than denoting them 'real and imaginary index'.

- P2 L62: The authors should consider to give $k_{abs,ice}$ a separate, numbered equation.

- P5 L139: The equation for the spectral transmittance should become a separate equation instead of an in-text equation.

- P6 L179: do you mean Warren and Brandt (2008)?

- P7 L183: Equation (2) does not give a direct relation to calculate $k_{att}(\lambda)$. The authors should consider providing a separate equation for this purpose.

- Figure 3:

  – The y-axes of Figs. 3a and 3b don't show the Transmittance T but its logarithm ln T - please adjust the axes titles accordingly.

  – Legend for black line: 350 nm are stated in the text - is '351 nm' a typing error?

- P8 L220: the superscript 1 belongs to the unit of $k_{att}$, please keep it in one line.

- P8 L234: reference should be to Figure 4a not Figure 5.

- P8 L236: 4 μm instead of 4 um.

- P9 L252: missing closing bracket after Eq. (7)

- Figure 6: add y-axis title to Fig. 6b.

- P9 L273-274: don't split the unit on different lines.

- P10 L281: I think this should be a reference to Fig. 3b.

- P10 L287: I guess you don't mean $\omega$?

- P11 L310: to six previously published [...], not seven

- Bibliography: please add how to access the Mätzler (2002) and Perovich (1996) references.

- Appendix 1, P29 L693: the minus '-' in the unit is missing

---

## Referee Comment (RC2) · Anonymous Referee #2 · 23 Jul 2020

This study presents attenuation flux coefficients using spectral irradiance measurements of bare glacial ice in western Greenland. These coefficients are compared with theory and other data sets. The authors conclude that attenuation is enhanced due to a semi-granular near-surface ice layer and by light absorbing impurities at their measurement location. As attenuation flux coefficients for glacial ice are scarce, the data set presented in this study is therefore an important addition for the scientific community. The manuscript is generally well written with clear figures. The following issues should be addressed before publication:

[Figure]

General comments:

1) The title and part of the introduction (line 81-91) looked a bit strange to me. I got the impression that a significant part of the study is a about the ICEsat satellite, while it is only briefly discussed in Sect. 4.3. Therefore, I would suggest to shorten the title and leave out the ICEsat part, and shorten the part of the introduction about ICEsat to take away the confusion.

2) The Kangerlussuaq region is well known to have a high LAP concentration (e.g., Wientjes et al., 2011; Tedstone et al., 2020). Therefore, it is not surprising that impurities impact the results. The authors, however, state on various occasions in the manuscript that there might be LAPs involved, almost like it is a new finding (e.g., line 331-333: "Comparison with the spectral coefficient for pure ice (Figure 4c) suggests the discrepancy we find is likely due to LAPs present in the measured volume, which appear to disproportionately enhance energy absorption near the ice surface" or on line 385-386: "This suggest light absorbing particles enhance visible light absorption and reduce optical penetration depth at our field site"). I think that the manuscript would benefit if more literature is used to determine if the results are in agreement with the observed LAP concentration for this region.

3) The authors state that no asymptotic flux attenuation coefficients are available for glacial ice (e.g., line 72-73), but Ackermann et al. (2006) (which is cited in this manuscript) reported absorption coefficients for glacial ice in Antarctica. Although it is true that Ackermann et al. (2006) measured deep glacial ice in Antarctica while the authors measured bare glacial ice in Greenland, for some cases it compares relatively well with the results presented in the manuscript, as you have shown in Fig. 9. Furthermore, Ackermann et al. (2006) show the absorption coefficient for 532 nm (Fig. 16 of that paper), which does not seem to not match the statement on line 89-91. I would like these issues discussed on the relevant places in this study or an explanation why the authors think it is not comparable (For example, on line 72-73, line 89-91, line 226-229, line 305-307, Sect. 4.3, Fig. 4b, Fig. 9).

[Figure]

4) Figure 4b should be replaced by Fig. 9, as Fig 4b seems redundant. Furthermore, a discussion in more detail about Fig. 9 is desirable. On one hand it shows that the results of this study are in agreement with AMANDA 1755 m, and support the claim that impurities are an important factor (which is mentioned on various places in the manuscript, like on line 226-229 or line 282-283). However, on the other hand it shows that the difference with the pure-ice estimate of Picard et al. 2016 is very small. This is confusing for me. I also think that Fig. 4a and Fig. 8 can be merged.

5) Most Figures are barely introduced in the manuscript, while a highly detailed description is provided in the caption. I would suggest to move some of the caption to the main text.

6) It would have been better for the $\chi$ term that is introduced in Sect. 2.5 to be wavelength dependent, as attenuation in the surface layer strongly depends on wavelength (e.g., Fig. 6 and 7 of Grenfell and Maykut, 1977). This maybe could explain the increasing difference with wavelength for the 12 cm depth fit in Fig. 5c. As the differences are still rather small, I do not think that it is necessary to adjust the results to a wavelength dependent $\chi$, but I think that the manuscript would benefit if the authors state the uncertainty that arises because of this choice. Also, I do not understand line 195-197. Isn't $\chi$ now practically the same as i0 due to the spectral integration?

7) Figure 7, lines 268 – 275 and Sect. 3.5, except for line 285-287, should be moved to the methods.

8) Use the abbreviation 'Fig.' when referring to a figure in running text, unless it is at the beginning of the sentence.

Minor comments:

Line 65: Change "size > wavelength" to "size larger than wavelength"

Line 66-69: Add references to this statement.

For all equations: Use punctuation at the end of the equation, as the equation is part

of a sentence.

Line 148-149: What do you mean with "Solid ice-equivalent values?"

Line 154: g is usually assumed to be independent of wavelength. Also call it the asymmetry factor and define the single-scattering albedo.

Eq 5: Are you sure that Schuster, 1905 is the right reference for this equation? Libois et al. (2013) and Tuzet et al. (2019, cited in this manuscript) describe it relatively well. They also use the Delta-Eddington method, which should be mentioned in the manuscript.

Fig. 2.: Please change "Relative irradiance" to "transmittance" and add the units for the standard deviation.

Fig. 3: Do the authors have any idea why k_att becomes increasingly smaller and very small around 850-900 nm, which does not seem to be in agreement with Warren et al. 2006? I know that in the manuscript it is stated that beyond 700 nm the flux is small and the results become less reliable, but it seems to be odd.

Line 216: Define the albedo, as e.g. the surface reflectivity of solar radiation .

Line 218: Please use more recent references, e.g. Gardner and Sharp (2010), and/or He and Flanner (2020).

Line 230: Add more references.

Line 241. Do the authors mean Eq. 17 instead of Eq. 16 of Warren et al. (2006)?

Fig.6b: Please put k_att(0-12 cm) / k_att on the y-axis and remove the legend.

Line 268: I assume the authors mean with "The field measurements" your observations, and not from Grenfell and Maykut (1977)? Please clarify.

Line 287: "omega > 800 nm". 800 nm does not make sense, as omega is defined in this manuscript as the single-scattering albedo.

Line 323-324: "The comparison demonstrates the tremendous variation in k_att values". The term 'tremendous' is a bit overexaggerated. Besides, the differences are not that large if the absorption coefficient is compared to glacial ice or pure ice (Fig. 9).

Line 374: Change "to modelling light attenuation in glacier ice" to "to modelling light attenuation in near-surface glacier ice".

Line 394-397: This is a bit vague, please reformulate.

Bibliography:

Tedstone, A. J., Cook, J. M., Williamson, C. J., Hofer, S., McCutcheon, J., Irvine-Fynn, T., Gribbin, T. and Tranter, M.: Algal growth and weathering crust state drive variability in western Greenland Ice Sheet ice albedo, The Cryosphere, 14, 521-538, https://doi.org/10.5194/tc-14-521-2020, 2020

Wientjes, I. G. M., Van de Wal, R. S. W., Reichart, G. J., Sluijs, A. and Oerlemans, J.: Dust from the dark region in the western ablation zone of the Greenland ice sheet, The Cryosphere, 5, 589-601, https://doi.org/10.5194/tc-5-589-2011, 2011.

Libois, Q., Picard, G., France, J. L., Arnaud, L., Dumont, M., Carmagnola, C. M., and King, M. D.: Influence of grain shape on light penetration in snow, Cryosphere, 7, 1803–1818, https://doi.org/10.5194/tc-7-1803-2013, 2013.

Gardner, A. S. and Sharp, M. J.: A review of snow and ice albedo and the development of a new physically based broadband albedo parameterization, Journal of Geophysical Research: Earth Surface, 115, https://doi.org/10.1029/2009JF001444, 2010.

He, C. and Flanner, M.: Snow Albedo and Radiative Transfer: Theory, Modeling, and Parameterization, pp. 67–133, Springer International Publishing, Cham, https://doi.org/10.1007/978-3-030-38696-2_3, 2020.

---

## Author Comment (AC1) · 15 Dec 2020

Author response: Thank you for your detailed comments on our manuscript. We believe we have addressed each request. Below, we provide a point-by-point reply to each comment. Please note that the other reviewer requested a comprehensive instrumental and measurement uncertainty analysis. Please see the attached supplementary document that describes the Monte Carlo radiative transfer simulations that we performed for this purpose.

[Figure]

This study presents attenuation flux coefficients using spectral irradiance measurements of bare glacial ice in western Greenland. These coefficients are compared with theory and other data sets. The authors conclude that attenuation is enhanced due to a semi-granular near-surface ice layer and by light absorbing impurities at their measurement location. As attenuation flux coefficients for glacial ice are scarce, the data set presented in this study is therefore an important addition for the scientific community. The manuscript is generally well written with clear figures. The following issues should be addressed before publication:

General comments: 1) The title and part of the introduction (line 81-91) looked a bit strange to me. I got the impression that a significant part of the study is a about the ICEsat satellite, while it is only briefly discussed in Sect. 4.3. Therefore, I would suggest to shorten the title and leave out the ICEsat part and shorten the part of the introduction about ICEsat to take away the confusion.

Author reply: As requested, we shortened the title and removed the ICESat part. We moved the ICESat paragraph from the introduction to the discussion where it provides an example for how our dataset can be used (c.f. Deems et al., 2013; Smith et al., 2018).

2) The Kangerlussuaq region is well known to have a high LAP concentration (e.g., Wientjes et al., 2011; Tedstone et al., 2020). Therefore, it is not surprising that impurities impact the results. The authors, however, state on various occasions in the manuscript that there might be LAPs involved, almost like it is a new finding (e.g., line 331-333: "Comparison with the spectral coefficient for pure ice (Figure 4c) suggests the discrepancy we find is likely due to LAPs present in the measured volume, which appear to disproportionately enhance energy absorption near the ice surface" or on line 385-386: "This suggest light absorbing particles enhance visible light absorption and reduce optical penetration depth at our field site"). I think that the manuscript would benefit if more literature is used to determine if the results are in agreement with the observed LAP concentration for this region.

Author reply: We did not measure LAP concentration directly; therefore, we were cautious in noting that we infer LAP influence. Our intention is not to suggest absorption by LAPs is surprising. However, it is also important to acknowledge that prior studies reported on albedo and/or reflectance, whereas the results referenced here relate to transmittance at >12 cm depth below the ice surface and therefore provide additional context regarding light absorption by LAPs within the ice volume, rather than on or very near the ice surface.

3) The authors state that no asymptotic flux attenuation coefficients are available for glacial ice (e.g., line 72-73), but Ackermann et al. (2006) (which is cited in this manuscript) reported absorption coefficients for glacial ice in Antarctica. Although it is true that Ackermann et al. (2006) measured deep glacial ice in Antarctica while the authors measured bare glacial ice in Greenland, for some cases it compares relatively well with the results presented in the manuscript, as you have shown in Fig. 9. Furthermore, Ackermann et al. (2006) show the absorption coefficient for 532 nm (Fig. 16 of that paper), which does not seem to not match the statement on line 89-91. I would like these issues discussed on the relevant places in this study or an explanation why the authors think it is not comparable (For example, on line 72-73, line 89-91, line 226-229, line 305-307, Sect. 4.3, Fig. 4b, Fig. 9).

Author reply: We removed the claim "first" everywhere to avoid confusion. We acknowledge the need to distinguish our results from those of Ackermann et al. (2006). There are two main distinctions: 1) compressed glacial ice at >800 m depth has no granular structure, and 2) absorptivity of compressed glacial ice at >800 m depth is controlled by factors that are only partly relevant to the ice sheet surface (dust concentrations from past millennia). The Ackermann et al. (2006) results demonstrate that scattering at >800 m depth is mainly controlled by clathrates, indicating that air bubbles are also of little importance. An earlier study, closely related to the Ackerman study, concludes: "Scattering . . . at ice–ice boundaries . . . will be of minor importance" (Price and Bergström, 1997). Consequently, the two main factors that control light scattering near

the ice sheet surface (granularity and air bubbles) are of minor importance to the light scattering results presented by Ackermann et al. (2006).

Although we agree that Fig. 16 of Ackermann et al. (2006) compares relatively well with our results, it is important to acknowledge that this is at least in part incidental, and different physical mechanisms are at play in both cases. It is not our intention to suggest that Ackermann's results are irrelevant to our study or that our study is incomparable to that study. Rather, the underlying physical mechanisms that control light attenuation are different in both cases. As such, we have removed claims of "first" throughout the paper and we added additional context for the difference between our study and the Ackermann et al. (2006) study in the relevant discussion.

4) Figure 4b should be replaced by Fig. 9, as Fig 4b seems redundant. Furthermore, a discussion in more detail about Fig. 9 is desirable. On one hand it shows that the results of this study are in agreement with AMANDA 1755 m, and support the claim that impurities are an important factor (which is mentioned on various places in the manuscript, like on line 226-229 or line 282-283). However, on the other hand it shows that the difference with the pure-ice estimate of Picard et al. 2016 is very small. This is confusing for me. I also think that Fig. 4a and Fig. 8 can be merged.

Author reply: We removed Fig. 4 and we point to Fig. 8 and 9 where we previously pointed to Fig. 4, as requested. We added a new paragraph that concludes the Discussion section focused on Fig. 9.

5) Most Figures are barely introduced in the manuscript, while a highly detailed description is provided in the caption. I would suggest to move some of the caption to the main text.

Author reply: We addressed this throughout the manuscript, as requested.

6) It would have been better for the $\chi$ term that is introduced in Sect. 2.5 to be wavelength dependent, as attenuation in the surface layer strongly depends on wavelength

(e.g., Fig. 6 and 7 of (Grenfell and Maykut, 1977)). This maybe could explain the increasing difference with wavelength for the 12 cm depth fit in Fig. 5c. As the differences are still rather small, I do not think that it is necessary to adjust the results to a wavelength dependent $\chi$, but I think that the manuscript would benefit if the authors state the uncertainty that arises because of this choice. Also, I do not understand line 195-197. Isn't $\chi$ now practically the same as i0 due to the spectral integration?

Author reply: We originally included the spectrally averaged value because large-scale models often need a single value for the visible and a single value for the infrared, or one single broadband value (Briegleb and Light, 2007; Liston and Winther, 2005). We now report spectral $\chi(\lambda)$ values in addition to the average value, as requested by another reviewer.

The reason we distinguish $\chi$ and i_o is because i_o is defined in such a way that it partitions the absorbed solar flux, or the net solar flux divergence, whereas we use $\chi$ to partition the downward flux. Grenfell and Maykut (1977) use their albedo measurements and modeling to extend their albedo and extinction coefficients across the solar spectrum and thereby to calculate the net flux divergence. Overall, our main goal with this part of the paper is to communicate the idea that a surface scattering layer is present on the ice sheet, and that this concept is well-developed in the sea ice literature but is conspicuously absent from the glaciological literature. To that end, we felt that a single $\chi$ value was sufficient to communicate that message.

7) Figure 7, lines 268 – 275 and Sect. 3.5, except for line 285-287, should be moved to the methods.

Author reply: We moved these sections and Fig. 7 to the methods, as requested.

8) Use the abbreviation 'Fig.' when referring to a figure in running text, unless it is at the beginning of the sentence.

Author reply: This has been corrected throughout the manuscript, as requested.

Minor comments:

Line 65: Change "size > wavelength" to "size larger than wavelength"

Author reply: This has been corrected, as requested.

Line 66-69: Add references to this statement.

Author reply: We added the following references to this statement:

Brandt, R. E. and Warren, S. G.: Solar-heating rates and temperature profiles in Antarctic snow and ice, Journal of Glaciology, 39(131), 99–110, doi:10.3189/S0022143000015756, 1993.

Liston, G. E., Bruland, O., Elvehøy, H. and Sand, K.: Below-surface ice melt on the coastal Antarctic ice sheet, Journal of Glaciology, 45(150), 273–285, doi:10.3189/002214399793377130, 1999.

Wiscombe, W. J. and Warren, S. G.: A Model for the Spectral Albedo of Snow. I: Pure Snow, J. Atmos. Sci., 37(12), 2712–2733, doi:10.1175/1520-0469(1980)037<2712:AMFTSA>2.0.CO;2, 1980.

For all equations: Use punctuation at the end of the equation, as the equation is part of a sentence.

Author reply: This has been corrected, as requested.

Line 148-149: What do you mean with "Solid ice-equivalent values?"

Author reply: "Solid ice-equivalent values" refers to normalization of the $k_{att}$ values by the ratio of solid ice density to measured (sample) density: $k_i = _i/ k_{att}$ where is measured ice density, $_i$ is solid ice density (917 kg m-3), $k_i$ is $k_{att}$ in units of (inverse) "solid-ice equivalent thickness" [m-1] and $k_{att}$ is in units of (inverse) in-situ ice thickness.

Line 154: g is usually assumed to be independent of wavelength. Also call it the

asymmetry factor and define the single-scattering albedo.

Author reply: We now call it the asymmetry factor and we defined the single scattering albedo, as requested. For completeness, we retained the wavelength dependence of g.

Eq 5: Are you sure that Schuster, 1905 is the right reference for this equation? Libois et al. (2013) and Tuzet et al. (2019, cited in this manuscript) describe it relatively well. They also use the Delta-Eddington method, which should be mentioned in the manuscript.

Author reply: The equation we use is derived equivalently from the Eddington approximation or the two-stream derivation given in Bohren, (1987). Schuster, (1905) is usually credited with the asymptotic two-stream solution (Mishchenko, 2013). We cite Tuzet et al. (2019) and Libois et al. (2013) in the manuscript.

Fig. 2.: Please change "Relative irradiance" to "transmittance" and add the units for the standard deviation.

Author reply: We changed "relative irradiance" to "transmittance" in the figure caption and we added the units for standard deviation, as requested.

Fig. 3: Do the authors have any idea why k_att becomes increasingly smaller and very small around 850-900 nm, which does not seem to be in agreement with Warren et al. 2006? I know that in the manuscript it is stated that beyond 700 nm the flux is small and the results become less reliable, but it seems to be odd.

Author reply: The values beyond ~700 nm are inaccurate. We show them to help the reader understand why we restrict our k_att values to the range 350–700 nm, whereas transmittance was measured to 900 nm (and is plotted in this range in Fig. 2c).

Line 216: Define the albedo, as e.g. the surface reflectivity of solar radiation .

Author reply: We added a definition for albedo and explained how we calculate it and

how we use it.

The text reads: "The ice surface albedo was estimated as the ratio of the 2 m background upwelling spectral irradiance to the downwelling spectral irradiance. These irradiance data were smoothed with the same 1 nm interpolation filter described [for the in-ice irradiance measurements]. The ice surface albedo is presented in Sect. 4 to qualitatively discuss the in-ice irradiance measurements and the k_att ($\lambda$) estimates."

Line 218: Please use more recent references, e.g. Gardner and Sharp (2010), and/or He and Flanner (2020).

Author reply: We added both of these references. Thank you for alerting us to the review by He and Flanner (2020), it was helpful.

Line 230: Add more references.

Author reply: Line 230 in the discussion paper is a comment that grain size dominates absorption beyond ~530 nm. Line 229 is a comment that LAPs dominate absorption at shorter wavelengths. We are not sure which of these two comments this request is aimed at, but we added the following references that address both comments (He et al., 2017; Libois et al., 2013, 2014):

He, C., Takano, Y., Liou, K.-N., Yang, P., Li, Q. and Chen, F.: Impact of Snow Grain Shape and Black Carbon–Snow Internal Mixing on Snow Optical Properties: Parameterizations for Climate Models, J. Climate, 30(24), 10019–10036, doi:10.1175/JCLI-D-17-0300.1, 2017.

Libois, Q., Picard, G., France, J. L., Arnaud, L., Dumont, M., Carmagnola, C. M. and King, M. D.: Influence of grain shape on light penetration in snow, The Cryosphere, 7(6), 1803–1818, doi:10.5194/tc-7-1803-2013, 2013.

Libois, Q., Picard, G., Dumont, M., Arnaud, L., Sergent, C., Pougatch, E., Sudul, M. and Vial, D.: Experimental determination of the absorption enhancement parameter of snow, Journal of Glaciology, 60(222), 714–724, doi:10.3189/2014JoG14J015, 2014.
Line 241. Do the authors mean Eq. 17 instead of Eq. 16 of Warren et al. (2006)?

Author reply: Yes, thank you, we corrected this.

Fig.6b: Please put k_att(0-12 cm) / k_att on the y-axis and remove the legend.

Author reply: We have made these corrections, as requested.

Line 268: I assume the authors mean with "The field measurements" your observations, and not from Grenfell and Maykut (1977)? Please clarify.

Author reply: Yes, we are referring to our measurements. We clarified this in the revised text.

Line 287: "omega > 800 nm". 800 nm does not make sense, as omega is defined in this manuscript as the single-scattering albedo.

Author reply: This typo has been corrected. The revised text reads: "the maximum difference found was 0.2% for values of $\omega$ at wavelengths greater than 800 nm."

Line 323-324: "The comparison demonstrates the tremendous variation in k_att values". The term 'tremendous' is a bit overexaggerated. Besides, the differences are not that large if the absorption coefficient is compared to glacial ice or pure ice (Fig. 9).

Author reply: We revised the text as follows: "The comparison demonstrates that k_att values vary by >1 order of magnitude at visible wavelengths due to differences in ice structure and composition"

Line 374: Change "to modelling light attenuation in glacier ice" to "to modelling light attenuation in near-surface glacier ice".

Author reply: We added "near-surface", as requested.

Line 394-397: This is a bit vague, please reformulate.

Author reply: We removed this sentence at the request of another reviewer and replaced it with a summary of the scattering and absorption coefficient values that we

quantify.

Bibliography (reviewer):

Tedstone, A. J., Cook, J. M., Williamson, C. J., Hofer, S., McCutcheon, J., Irvine-Fynn, T., Gribbin, T. and Tranter, M.: Algal growth and weathering crust state drive variability in western Greenland Ice Sheet ice albedo, The Cryosphere, 14, 521-538, https://doi.org/10.5194/tc-14-521-2020, 2020

Wientjes, I. G. M., Van de Wal, R. S. W., Reichart, G. J., Sluijs, A. and Oerlemans, J.: Dust from the dark region in the western ablation zone of the Greenland ice sheet, The Cryosphere, 5, 589-601, https://doi.org/10.5194/tc-5-589-2011, 2011.

Libois, Q., Picard, G., France, J. L., Arnaud, L., Dumont, M., Carmagnola, C. M., and King, M. D.: Influence of grain shape on light penetration in snow, Cryosphere, 7, 1803–1818, https://doi.org/10.5194/tc-7-1803-2013, 2013.

Gardner, A. S. and Sharp, M. J.: A review of snow and ice albedo and the development of a new physically based broadband albedo parameterization, Journal of Geophysical Research: Earth Surface, 115, https://doi.org/10.1029/2009JF001444, 2010.

He, C. and Flanner, M.: Snow Albedo and Radiative Transfer: Theory, Modeling, and Parameterization, pp. 67–133, Springer International Publishing, Cham, https://doi.org/10.1007/978-3-030-38696-2_3, 2020.

Bibliography (response):

Briegleb, B. P.: Delta-Eddington approximation for solar radiation in the NCAR community climate model, J. Geophys. Res., 97(DBohren, C. F.: Multiple scattering of light and some of its observable consequences, American Journal of Physics, 55(6), 524–533, doi:10.1119/1.15109, 1987.

Briegleb, B. P. and Light, B.: A Delta-Eddington Mutiple Scattering Parameterization for Solar Radiation in the Sea Ice Component of the Community Climate System

[Figure]

Model, Technical Note, National Center for Atmospheric Research, Boulder, Colorado. [online] Available from: http://dx.doi.org/10.5065/D6B27S71 (Accessed 18 February 2019), 2007.

Deems, J. S., Painter, T. H. and Finnegan, D. C.: Lidar measurement of snow depth: a review, Journal of Glaciology, 59(215), 467–479, doi:10.3189/2013JoG12J154, 2013.

Grenfell, T. C. and Maykut, G. A.: The Optical Properties of Ice and Snow in the Arctic Basin*, Journal of Glaciology, 18(80), 445–463, doi:10.3189/S0022143000021122, 1977.

He, C., Takano, Y., Liou, K.-N., Yang, P., Li, Q. and Chen, F.: Impact of Snow Grain Shape and Black Carbon–Snow Internal Mixing on Snow Optical Properties: Parameterizations for Climate Models, J. Climate, 30(24), 10019–10036, doi:10.1175/JCLI-D-17-0300.1, 2017.

Libois, Q., Picard, G., France, J. L., Arnaud, L., Dumont, M., Carmagnola, C. M. and King, M. D.: Influence of grain shape on light penetration in snow, The Cryosphere, 7(6), 1803–1818, doi:10.5194/tc-7-1803-2013, 2013.

Libois, Q., Picard, G., Dumont, M., Arnaud, L., Sergent, C., Pougatch, E., Sudul, M. and Vial, D.: Experimental determination of the absorption enhancement parameter of snow, Journal of Glaciology, 60(222), 714–724, doi:10.3189/2014JoG14J015, 2014.

Liston, G. E. and Winther, J.-G.: Antarctic Surface and Subsurface Snow and Ice Melt Fluxes, J. Climate, 18(10), 1469–1481, doi:10.1175/JCLI3344.1, 2005.

Mishchenko, M. I.: 125 years of radiative transfer: Enduring triumphs and persisting misconceptions, pp. 11–18, Dahlem Cube, Free University, Berlin., 2013.

Price, P. B. and Bergström, L.: Optical properties of deep ice at the South Pole: scattering, Appl. Opt., AO, 36(18), 4181–4194, doi:10.1364/AO.36.004181, 1997.

Schuster, A.: Radiation through a foggy atmosphere, The Astrophysical Journal,

XX1(1), 1–22, 1905.

Smith, B. E., Gardner, A., Schneider, A. and Flanner, M.: Modeling biases in laser-altimetry measurements caused by scattering of green light in snow, Remote Sensing of Environment, 215, 398–410, doi:10.1016/j.rse.2018.06.012, 2018. 7), 7603, doi:10.1029/92JD00291, 1992.

Deems, J. S., Painter, T. H. and Finnegan, D. C.: Lidar measurement of snow depth: a review, Journal of Glaciology, 59(215), 467–479, doi:10.3189/2013JoG12J154, 2013.

Liston, G. E. and Winther, J.-G.: Antarctic Surface and Subsurface Snow and Ice Melt Fluxes, J. Climate, 18(10), 1469–1481, doi:10.1175/JCLI3344.1, 2005.

Smith, B. E., Gardner, A., Schneider, A. and Flanner, M.: Modeling biases in laser-altimetry measurements caused by scattering of green light in snow, Remote Sensing of Environment, 215, 398–410, doi:10.1016/j.rse.2018.06.012, 2018.

Please also note the supplement to this comment:
https://tc.copernicus.org/preprints/tc-2020-53/tc-2020-53-AC1-supplement.pdf

―――――――――――――――――

**Fig. 1.** Attenuation coefficient k_att spectra from measurements of light transmission collected on 20 July, 2018, compared with average k_att values from four simulations with a 3-dimensional Monte Carlo radi

[Figure]

[Figure]

**Fig. 2.**

**Supplement:**

**Table of contents:**

**S1 Monte Carlo radiative transfer model**

The Monte Carlo method solves the radiative transfer equation (RTE) by simulating large ensembles of photon events represented by random samples from probability density functions (Ertürk and Howell, 2017). In this study and others, the Monte Carlo method is used to quantify relative uncertainties in imperfect optical measurements that are intractable with analytical or numerical solutions to the RTE (Gordon, 1985). We developed a Monte Carlo radiative transfer model to estimate the effect of detector interference on our irradiance measurements. The model closely follows methods developed to simulate light propagation in biological tissue, ocean waters, and sea ice (Leathers et al., 2004; Light et al., 2003; Wang et al., 1995). A general description of the model and particular modifications for this investigation are described below.

**S1.1 Probability functions for optical properties**

The fundamental ingredients of this and other Monte Carlo radiative transfer models are the inherent optical properties k, ω, and g (see Sect. 2.3 of the main), the geometric boundary conditions, and the probabilistic rules that govern the system. The cumulative probability of occurrence for an event x, with probability density function $p(x)$, is:

$$P(x) = \int_{-\infty}^{x} p(x)dx, \quad 0 \leq P(x) \leq 1. \tag{1}$$

To solve for x, the left-hand-side (LHS) of (1) is replaced with a random number:

$$P(x) = q \tag{2}$$

where $q$ is from the uniform distribution over [0,1]. The right-hand-side (RHS) lower limit of integration $-\infty$ is replaced with an appropriate limit (e.g., 0) and analytic or empirical expressions for $p(x)$ are specified.

In this study, x represents optical path length, scattering direction, and photon survival probability. Closed-form expressions for each of these terms are given in the following sections.

**S1.1.1 Optical path length**

The probability density function for the optical path length $l$ [m$^{-1}$] is given by the e-folding length:

$$p(l) = e^{-l}, \quad l \geq 0 \tag{3}$$

with the cumulative distribution function:

$$P(l) = \int_{0}^{l} e^{-l'}dl' = 1 - e^{-l}. \tag{4}$$

From Eq. (2), $q = 1 - e^{-l}$ and therefore:

$$l = -\ln q, \quad 0 \leq 1. \tag{5}$$

In this study, $q$ is generated with the MATLAB function `rand`.

The photon transport length [m] is the optical path length scaled by the extinction coefficient:

$$s = l/\sigma_e \tag{6}$$

where:

$$\sigma_e = \sigma_s + \sigma_a \tag{7}$$

is the single-scattering extinction coefficient, $\sigma_s$ [m$^{-1}$] is the scattering coefficient, and $\sigma_a$ [m$^{-1}$] is the absorption coefficient.

**S1.1.2 Scattering phase function**

The probability density function for a scattering phase function with azimuthal symmetry is:

$$p(\theta_s) = 2\pi\tilde{\beta}(\theta_s)\sin\theta \tag{8}$$

where $\tilde{\beta}(\theta_s)$ is the probability that a photon will scatter at polar angle $\theta_s$. We specify $\tilde{\beta}(\theta_s)$ with the Henyey-Greenstein scattering phase function, which is appropriate for strongly forward scattering by ice grains and air bubbles (Light et al., 2003):

$$\tilde{\beta}(g, \theta_s) = \frac{1}{4\pi}\frac{1 - g^2}{(1 + g^2 - 2g\cos\theta_s)^{\frac{3}{2}}}, \qquad -1 < g < 1. \tag{9}$$

where $g = 0$ reduces Eq. 9 to isotropic scattering and $g \to 1$ is strongly forward scattering. In this study, $g = 0.86$, as given by Mullen & Warren (1988) from Mie theory calculations for scattering by air bubbles in ice.

From Eq. (1):

$$P(\theta_s) = -\frac{1 - g^2}{2}\int_0^{\theta_s}\frac{\sin\theta_s'}{(1 + g^2 - 2g\cos\theta_s')^{\frac{3}{2}}}d\theta_s' = q \tag{10}$$

which evaluates to:

$$q = \frac{1 - g^2}{2g}\left[\frac{1}{1 - g} - \frac{1}{\sqrt{1 + g^2 - 2g\cos\theta_s}}\right] \tag{11}$$

yielding the scattering angle:

$$\cos\theta_s = \frac{1}{2g}\left[1 + g^2 - \left(\frac{1 - g^2}{1 - g + 2gq}\right)^2\right], \qquad g \neq 0; \ 0 \leq \theta_s \leq \pi/2. \tag{12}$$

The probability density function for scattering azimuth angle $\phi_s$ in a spherical coordinate system with azimuthal symmetry is $1/2\pi$. From Eq. (1):

$$P(\phi_s) = \frac{\phi_s}{2\pi}, \qquad 0 \leq \phi_s \leq 2\pi \tag{13}$$

and from Eq. (2):

$$\phi_s = 2\pi q. \tag{14}$$

**S1.1.3 Photon termination**

Monte Carlo simulations are computationally expensive. To improve computational performance, photons are treated as packets of photons with initial weight $w = 1$. At each interaction, photons are scattered and absorbed according to their respective statistical probabilities, parameterized by $\sigma_s$ and $\sigma_a$. Accordingly, at each interaction the weight is updated as:

$$w = (1 - \overline{\omega}) \cdot w \tag{15}$$

where:

$$\overline{\omega} = \sigma_s / \sigma_e \tag{16}$$

is the single-scattering albedo [-]. Each $1 - \overline{\omega}$ reduction in photon packet weight is proportional to the probability of an individual photon absorption event. After many interactions, if $w$ drops below a very small value it contributes very little to the solution. The so-called "Russian roulette" technique is used to improve computational performance, where photon packet weights below a specified threshold $w < w_{min}$ are increased in proportion to a survival probability function and are re-released into the medium, or otherwise terminated:

$$w = \begin{cases} m \cdot w, & q \leq 1/m \\ 0, & q > 1/m \end{cases} \tag{17}$$

where $1/m$ is the probability of photon survival and $q$ is a random number as previously defined. This technique conserves energy and is unbiased (Wang et al., 1995). In this study, $w_{min} = 10^{-5}$ and $m = 10$.

At each interaction, the absorbed fraction $\overline{\omega} \cdot w$ is scored into an absorption array in a cylindrical coordinate system that is used to compute observable quantities of absorption and photon fluence. If a photon packet exits the medium, it is scored into a transmittance or reflectance array in an azimuthally independent spherical coordinate system that is used to compute observable quantities of irradiance, radiant intensity, and power. These scoring systems follow the definitions in Wang et al. (1995) Eq. 4.1–4.32.

The preceding sections describe the fundamental processes of photon transport, scattering direction, and survival probability. Similar probability density functions that describe the detector rod interference are described next.

**S1.2 Monte Carlo experiment**

The detector rod interference is estimated with a "backward" Monte Carlo (BMC) simulation, which simulates photon trajectories starting from the detector backward to the target (Leathers et al., 2004; Light et al., 2003). Here, the target is the ice surface. The simulation domain is a 3-dimensional ice slab with one boundary, the ice surface, and otherwise infinite horizontal and vertical extent. A cylinder with dimensions identical to the detector rod is placed at positions identical to the measurement depths reported in this paper, and photon packets are released from the irradiance sensor ("remote cosine receptor") located on the detector rod (Fig. S1).

[Figure]

**Fig. S1: Example Monte Carlo photon tracking simulation from model output used in this study, with interference by cylindrical detector rod. (a) ~14,000 random photon interactions are traced within a 3-dimensional ice volume. The cylindrical object represents the detector rod, here inserted at 1 m below the ice surface. The photon packet is released from the position of the irradiance sensor ("remote cosine receptor") located on the rod and traced backward to the ice surface ("backward Monte Carlo"). (b) Magnified view of the detector rod in the y-z plane shows photon packets scattering off of the rod. The color-bar represents the number of cumulative interactions experienced by this photon packet.**

As described above, each interaction within the ice volume is defined by absorption and scattering of the photon by ice. Absorption reduces the photon energy density by an amount $1 - \overline{\omega}_{ice}$. Scattering redirects the photon trajectory according to the Henyey-Greenstein scattering phase function with asymmetry parameter $g$ and transport distance $l$. Photon interactions with the detector rod require additional specifications that are described next.

**S1.2.1 Source function for cosine detector**

The scattering phase function for an irradiance sensor with a cosine response is:

$$\tilde{\beta}(\theta) = \frac{\cos\theta}{\pi}, \qquad 0 \le \theta \le \frac{\pi}{2} \tag{18}$$

with probability density function:

$$p(\theta) = 2\pi\tilde{\beta}(\theta)\sin\theta \tag{19}$$

and cumulative distribution function:

$$P(\theta) = 2\int_0^\theta \cos\theta'\sin\theta'\,d\theta' = q\,. \tag{20}$$

Substituting $\mu = \cos(\theta)$ the scattering angle is:

$$\cos\theta = \sqrt{1-q}. \tag{21}$$

For a forward Monte Carlo simulation, Eq. 21 gives the probability of photon receipt by an irradiance sensor with an ideal cosine response. For a BMC simulation, the form of Equation 21 that gives the initial launch trajectory of photons from the irradiance sensor surface is:

$$\cos \theta = -\sqrt{q}. \tag{22}$$

In reality, irradiance sensors do not have an ideal cosine response to radiance. In this experiment, the non-ideal cosine response of the irradiance sensor is estimated by replacing Eq. 22 with uniform sampling from an empirical probability density function derived from laboratory measurements of the cosine receptor angular response function provided by Ocean Optics (Fig. S2). The source azimuth angle $\phi$ is determined with Eq. (14).

[Figure]

Fig. S2: (a) Comparison of ideal angular response function (ideal cosine) with the empirical angular response function used to estimate the non-ideal response of the irradiance sensor used in this study. The empirical angular response function was developed by Ocean Optics from laboratory measurements on the same irradiance sensor type used in this study. (b) Same as (a) but normalized. The red line in (b) is the empirical probability density function used as the irradiance source function for our backward-Monte Carlo simulations (see Eq. 21–22).

**S1.2.2 Scattering and absorption by detector rod**

If a photon trajectory crosses the 3-dimensional position of the detector rod, the photon energy density is reduced by an amount $1 - \omega_{\text{rod}}$ and the photon is scattered away from the rod (Fig. S1) with an isotropic scattering phase function:

$$\theta_s = 1 - 2q, \tag{23}$$

$$\phi_s = 2\pi q. \tag{24}$$

The collision point is determined with ray tracing formulas that equate the vector equation of the photon trajectory with the parametric equation for the cylindrical detector rod surface following Ertürk and Howell (2017) Sect. 7.1 Eq. 59–66.

The polyvinyl chloride (PVC) detector rod albedo $\omega_{\text{rod}}$ is estimated from values for the complex refractive index of PVC (Zhang et al., 2020). Let $\mu = \cos \theta$ be the cosine zenith angle of incident radiation with $\mu = +1$ vertically

downward. Following Modest (2013) Section 2.5 Eq. 2.89–2.98, the Fresnel reflectivity and transmissivity to incident (downward) radiation are:

$$R_\text{F}(\mu) = \frac{1}{2}\left[\left(\frac{\mu - n\mu_\text{n}}{\mu + n\mu_\text{n}}\right)^2 + \left(\frac{n\mu - \mu_\text{n}}{n\mu + \mu_\text{n}}\right)^2\right] \tag{25}$$

$$T_\text{F}(\mu) = 1 - R_\text{F}(\mu) \tag{26}$$

where $n + ik$ and $n_0 + ik_0$ are the complex refractive indices of PVC and air, respectively, and:

$$\mu_\text{n} = \sqrt{1 - (1 - \mu^2)/n^2} \tag{27}$$

is the refracted cosine zenith angle in the PVC pipe. Radiation transmitted into the PVC is attenuated exponentially:

$$a(\mu_\text{n}) = e^{-\tau/\mu_\text{n}} \tag{28}$$

where:

$$\tau = 4\pi k L/\lambda \tag{29}$$

is the optical thickness of the PVC pipe with wall thickness $L = 0.004$ m. Radiation that transmits through $L$ is internally reflected upward from the inner wall in the direction $\mu_\text{n}$ and attenuated exponentially along path length $\tau$. Radiation that reaches the outer wall at $\mu_\text{n} < \mu_\text{c}$ is transmitted across the outer wall according to $T_\text{F}(\mu_\text{n})$ and reflected back into the PVC according to $R_\text{F}(\mu_\text{n})$, where $\mu_\text{c}$ is the critical angle given by Snell's law:

$$\mu_\text{c} = \sqrt{1 - 1/n^2}. \tag{30}$$

Formulas for $T_\text{F}(\mu_\text{n})$ and $R_\text{F}(\mu_\text{n})$ are similar to Eq. 25 and Eq. 26 with modifications for total internal reflection about $\mu_\text{c}$ and are given elsewhere (Briegleb and Light, 2007; Liou, 2002).

The total reflectivity is estimated with the successive-order-of-scattering method (van de Hulst, 1980), which accounts for the multiple internal reflections and absorption within the PVC described by Eq. 25–30. We model the PVC pipe as a plane, which is justified because the radius of curvature is much larger than all wavelengths of light considered here. For the geometry and optical properties of the detector rod, the total reflectivity has the closed-form solution:

$$R_\text{d} = R_{F,\mu} + \frac{T_{F,\mu} R_{F,\mu_\text{n}} T_{F,\mu_\text{n}}\, a_{\mu_\text{n}}^2}{1 - R_{F,\mu_\text{n}} a_{\mu_\text{n}}^2} \tag{31}$$

where the subscripts $\mu$ and $\mu_\text{n}$ on $R$, $T$, and $a$ indicate the direction of incident radiance.

**S2 Model validation**

The Monte Carlo model described above is verified by comparison with benchmark values for total diffuse reflectance $R_\text{d}$ [W m$^{-2}$], total transmittance $T_\text{t}$ [W m$^{-2}$], diffuse angular reflectance $R_\text{d}(\alpha)$ [W sr$^{-1}$] and diffuse angular transmittance $T_\text{d}(\alpha)$ [W sr$^{-1}$] tabulated by van de Hulst (1980). The angular quantities, which have units of radiant intensity, are defined with respect to the exiting angle normal to the surface $\alpha$ [rad]. For a plane-parallel slab with optical properties $\sigma_\text{s} = 0.9$ m$^{-1}$, $\sigma_\text{a} = 0.1$ m$^{-1}$, $g = 0.75$, and optical thickness $\tau = 2$, the van de Hulst (1980) solutions are $R_\text{d} = 0.09739$ and $T_\text{t} = 0.66096$. For an ensemble of $N = 100$ simulations, the Monte Carlo model described

above gives $R_d = 0.09740 \pm 0.00034$ and $T_t = 0.66098 \pm 0.00049$ ($\mu \pm 1\sigma$). The model closely reproduces the benchmark solutions for $R_d(\alpha)$ and $T_d(\alpha)$ (Fig. S3).

[Figure]

**Fig. S3: Values of diffuse angular reflectance (radiant intensity), $R_d(\alpha)$ and transmittance $T_d(\alpha)$ vs. the photon exiting angle with respect to the surface normal α (after Wang et al. 1995 Fig. 3). Solid circles are benchmark solutions from Table 35 in van de Hulst (1980), obtained with the doubling method of solution to the radiative transfer equation.**

**S3 Monte Carlo uncertainty estimate**

The Monte Carlo model is used to estimate the effect of detector interference on our irradiance measurements and, in turn, the asymptotic flux attenuation coefficients $k_{att}$ that are estimated from them. To this end, we designed four experiments that isolate two forms of detector interference: 1) the non-ideal cosine response of the irradiance detector, and 2) absorption and scattering by the PVC detector rod. The four experiments, including a base simulation with no detector interference, are summarized in Table S1.

**Table S1: Summary of four Monte Carlo experiments that simulate the effect of the detector rod interference on in-ice irradiance measurements. The baseline simulation (ideal diffusion, no rod) has no detector interference.**

| Experiment | Source function | Detector absorption | Detector scattering |
|---|---|---|---|
| Ideal Diffusion, No Rod | Eq. 23-24 | - | - |
| Ideal Cosine, No Rod | Eq. 22 | - | - |
| Ideal Cosine, With Rod | Eq. 22 | $\omega_{rod}$ | Eq. 23-24 |
| Non-ideal Cosine, With Rod | Empirical (Fig. S2) | $\omega_{rod}$ | Eq. 23-24 |

For each experimental setup, the Monte Carlo is integrated across 10,000 interactions at four wavelengths (400 nm, 500 nm, 600 nm, and 700 nm) with detector rod positions that are identical to the measurement depths reported in this paper (c.f. Fig. S1). For the 20 July experiment, these depths are 9.35 cm, 30.0 cm, 50.45 cm, and 68.60 cm, in units of solid-ice equivalent (i.e., physical thickness scaled by measured ice density). For the 21 July experiment, these

depths are 45.93 cm, 58.98 cm, 73.40 cm, and 114.5 cm. Monte Carlo $k_{att}$ values are estimated for each wavelength with the same method used for the field-estimates, i.e., by least-squares linear regression:

$$-\log T(z, \lambda) = T_0 + k_{att}(\lambda)z + \varepsilon \tag{32}$$

where $T$ is the total diffuse transmittance from Monte Carlo simulation (see Section S2), $T_0$ is a parameter (y-intercept) that represents $T(z = 0)$ and $\varepsilon$ is an error term.

These simulations provide two measures of $k_{att}$ uncertainty: 1) the difference between the average Monte Carlo $k_{att}$ value $\mu_{MC}$ and the field-estimated $k_{att}$ value at each wavelength, and 2) the spread among Monte Carlo $k_{att}$ values at each wavelength. The spread among Monte Carlo $k_{att}$ values is an estimate of uncertainty due to the irradiance sensor angular response function and the detector rod interference. The spectrometer dark-light sensitivity is an additional source of instrumental uncertainty that is estimated from field measurements as described in Sect. X. These instrumental uncertainties (irradiance sensor angular response, detector rod interference, and dark-light sensitivity) are compared with the statistical variations in the high-frequency irradiance measurements and with the statistical uncertainty in the $k_{att}$ linear regression model (Eq. 32).

If we take the spread among the Monte Carlo $k_{att}$ values as independent of the uncertainty estimates obtained from analysis of field datasets and the uncertainty in the linear regression model, a combined uncertainty is estimated as:

$$\varepsilon = \sqrt{\varepsilon_{MC}^2 + \varepsilon_D^2 + \varepsilon_{2\sigma}^2 + \varepsilon_{LM}^2} \tag{33}$$

where $\varepsilon_{MC}$, $\varepsilon_D$, $\varepsilon_{2\sigma}$, and $\varepsilon_{LM}$ are the uncertainty from Monte Carlo, dark-current sensitivity, high-frequency statistical variability, and linear model statistical uncertainty, defined as one standard error in the $k_{att}$ linear regression (Fig. S4).

[Figure]

**Fig S4: Attenuation coefficient $k_{att}$ spectra from measurements of light transmission collected on 20 July, 2018, compared with average $k_{att}$ values from four simulations with a 3-dimensional Monte Carlo radiative transfer model ($\mu_{MC}$) and with two measures of uncertainty: 1) statistical linear model uncertainty $\varepsilon_{LM}$ (shaded uncertainty bounds; $\pm 1$ standard error in the linear regression) and, 2) $\varepsilon_{LM}$ combined with instrumental and measurement uncertainty (error bars; $\mu_{MC} \pm \varepsilon$). The combined estimate combines $\varepsilon_{LM}$ with uncertainty due to spectrometer dark-light sensitivity, non-ideal cosine response of the irradiance sensor, detector rod interference, and statistical variations in the high-frequency raw data ($\pm 2$ standard deviations).**

**References**

Briegleb, B. P. and Light, B.: A Delta-Eddington Mutiple Scattering Parameterization for Solar Radiation in the Sea Ice Component of the Community Climate System Model, Technical Note, National Center for Atmospheric Research, Boulder, Colorado. [online] Available from: http://dx.doi.org/10.5065/D6B27S71 (Accessed 18 February 2019), 2007.

Ertürk, H. and Howell, J. R.: Monte Carlo Methods for Radiative Transfer, in Handbook of Thermal Science and Engineering, edited by F. A. Kulacki, pp. 1–43, Springer International Publishing, Cham., 2017.

Gordon, H. R.: Ship perturbation of irradiance measurements at sea 1: Monte Carlo simulations, Appl. Opt., 24(23), 4172, doi:10.1364/AO.24.004172, 1985.

van de Hulst, H. C.: Multiple light scattering: tables, formulas, and applications, Academic Press, New York., 1980.

Leathers, R. A., Downes, T. V., Davis, C. O. and Mobley, C. D.: Monte Carlo Radiative Transfer Simulations for Ocean Optics: A Practical Guide, Memorandum, Naval Research Laboratory, Washington, D.C. [online] Available from: https://www.oceanopticsbook.info/packages/iws_l2h/conversion/files/Leathersetal_NRL2004.pdf (Accessed 11 October 2020), 2004.

Light, B., Maykut, G. A. and Grenfell, T. C.: A two-dimensional Monte Carlo model of radiative transfer in sea ice, J. Geophys. Res. Oceans, 108(C7), 3219, doi:10.1029/2002JC001513, 2003.

Liou, K.-N.: An introduction to atmospheric radiation, 2nd ed., Academic Press, Amsterdam ; Boston., 2002.

Modest, M. F.: Radiative heat transfer, Third Edition., Academic Press, New York., 2013.

Mullen, P. C. and Warren, S. G.: Theory of the optical properties of lake ice, J. Geophys. Res. Atmospheres, 93(D7), 8403–8414, doi:10.1029/JD093iD07p08403, 1988.

Wang, L., Jacques, S. L. and Zheng, L.: MCML—Monte Carlo modeling of light transport in multi-layered tissues, Comput. Methods Programs Biomed., 47(2), 131–146, doi:10.1016/0169-2607(95)01640-F, 1995.

Zhang, X., Qiu, J., Li, X., Zhao, J. and Liu, L.: Complex refractive indices measurements of polymers in visible and near-infrared bands, Appl. Opt., 59(8), 2337, doi:10.1364/AO.383831, 2020.

---

## Author Comment (AC2) · 15 Dec 2020

**Table of contents:**

**1 Response to Anonymous Referee #1**

**General comments**

Within the manuscript, spectral measurements of light attenuation in Greenland Ice Sheet bare ice are presented. For this purpose, the authors employed spectral irradiance measurements between 350-900 nm wavelength at the surface and at four different depths below the ice surface and calculated the spectral transmittance within the ice. From this, spectral flux attenuation and absorption coefficients are derived and compared to previous studies

The manuscript is clearly structured and the figures are of good quality, which helps to convey the arguments of the authors. The measurements of in-ice spectral transmittance are very valuable, and the author's efforts to put them into context and to identify possible future applications should be acknowledged. However, there are some aspects that need further focus in my opinion. After some general comments, the more specific comments and suggestions for technical corrections follow below.

**Author reply: Thank you for your detailed comments on our manuscript. We believe we have addressed all of your concerns. In particular, we include a new detailed assessment of instrumental, measurement, and statistical uncertainty that we believe will substantially improve the usefulness of our findings.**

**Reviewer comment:** In my point of view, the most pressing aspect is the lack of accounting for measurement uncertainties within the manuscript. So far, only statistical variations along the 30 measured irradiance spectra are considered, which need to be clearly separated from instrument uncertainties. However, the latter are not mentioned at all within the manuscript.

I suggest to include a new subsection within the 'Methods'-section that is devoted to instrument and measurement uncertainties. I understand the instrument is calibrated for irradiance measurements, but instrument errors such as the influence of dark and stray light, the wavelength calibration of the spectrometer, the non-ideal cosine response of the RCR diffuser, and the uncertainty of the absolute calibration (to get calibrated irradiance measurements) need to be quantified. Furthermore, as the transmittance is measured calculating the ratio of measured irradiance from two different instruments (the one in the ice, and the permanent one at the surface), differences between the two instruments need to be quantified (e.g., by means of a cross-calibration with the same light source). These instrumental errors need to be put in relation to uncertainties regarding the measurement setup (e.g., is the cosine receptor really in close contact with the ice during the in-ice irradiance measurements?) and the statistical variations as deduced from the 30 subsequently measured spectra.

**Author reply:** We appreciate this request for an uncertainty analysis. We addressed this by comparison with additional field datasets (Fig. 1) and with simulations from a 3-dimensional Monte Carlo radiative transfer model (Sect. S1 of this document).

Based on our Monte Carlo, we estimate an uncertainty on $k_{att}$ due to the combined effects of detector rod interference and the non-ideal cosine response of the irradiance sensor diffusing element on the order 0.09–0.31 m$^{-1}$ for the wavelength range 400–700 nm. Stated in terms of e-folding length scale, this uncertainty range is 9–90 mm.

Based on analysis of field datasets, we estimate an uncertainty on $k_{att}$ due to dark-light sensitivity on the order 0.02–0.2 m$^{-1}$ for the wavelength range 350–700 nm, or 5–20 mm in e-folding length scale (Fig. 2 and Fig. 3).

For comparison, the uncertainty deduced from statistical variations in the measured in-ice irradiance is of the order 0.007–0.23 m$^{-1}$ at the two-sigma level, or 8–25 mm in e-folding length scale. The higher end of this uncertainty range applies to a small wavelength range near 700 nm. In the 400–600 nm range, the uncertainty is 0.005–0.007 m$^{-1}$ (Fig. 4).

If the combined uncertainty addressed by Monte Carlo is taken as independent of the other estimates, an overall systematic uncertainty on $k_{\mathrm{att}}$ is 0.09–0.38 m$^{-1}$ for the wavelength range 350–700 nm, estimated as:

$$\varepsilon = \sqrt{\varepsilon_{MC}^2 + \varepsilon_D^2 + \varepsilon_{2\sigma}^2}$$

where $\varepsilon_{MC}$, $\varepsilon_D$, and $\varepsilon_{2\sigma}$ are the uncertainty estimates for detector interference (estimated with Monte Carlo), dark-light sensitivity (estimated with field datasets), and statistical variations in the measured irradiances (estimated with field datasets), respectively. For comparison, the uncertainty in the linear regression, quantified as one standard error in the regression slope, is 0.05–0.22 m$^{-1}$. The specific methods used to estimate $\varepsilon_{MC}$, $\varepsilon_D$, and $\varepsilon_{2\sigma}$ are described in the new uncertainty analysis section in the Methods and Results, as requested.

For additional insight into experimental uncertainty, the revised manuscript includes a second attenuation coefficient ($k_{\mathrm{att}}$) spectrum from measurements collected on 21 July, one day after the 20 July experiment described in the submitted manuscript. The 21 July in-ice irradiance data were collected at depths between 53–124 cm below the ice surface, whereas the 20 July data were collected at depths between 12–77 cm below the ice surface (Fig. 1).

The 21 July $k_{\mathrm{att}}$ values are nearly identical to the 20 July values at wavelengths between 600–650 nm but are lower by 0.06–0.13 m$^{-1}$ in the 400–600 nm range. The lower $k_{\mathrm{att}}$ values at these visible wavelengths is consistent with the expectation that attenuation in deeper ice is less influenced by light absorbing material near the ice surface, whereas the nearly identical values at ~600 nm is consistent with the expectation that attenuation is dominated by ice absorption at near-infrared wavelengths. Differences between the 20 July and 21 July $k_{\mathrm{att}}$ spectra therefore likely reflect differences in the vertical variation of light absorbing material within the ice volume rather than instrumental and/or measurement uncertainty.

Overall, this comparison supports our experimental conclusions and gives additional insight into the relative influence of light absorbing material on variations in attenuation magnitude. More details are given in relevant sections below. The Monte Carlo model is described in the attached supplementary document.

**Reviewer comment:** This uncertainty analysis should eventually lead to vertical uncertainty bars in Figure 3, together with the horizontal uncertainty bars stemming from the depth measurements with the ruler (this second paragraph of Section 3.5 should also be moved to the new uncertainty subsection in Section 'Methods'). The errors in the linear regression to derive the flux attenuation coefficient should consider both depth and transmittance uncertainty estimates and, eventually, should lead to an uncertainty range attributed to each k$_{\mathrm{att}}$ value. A thorough treatment of the measurement uncertainties will definitely increase the value of the measurements for further applications.

**Author reply: As requested, a new uncertainty subsection has been added to Methods that describes the new measurement uncertainty estimates given above. These new values have been added as vertical uncertainty on the reported $k_{\mathrm{att}}$ spectra (Fig 8). Please also see Section S1 of this document (where Fig. 8 is located) that summarizes the new uncertainty estimate, as requested.**

**Reviewer comment:** Another remark is more structural and concerns the introduction of figures in the text: Some figures need to be described and explained in more detail within the text already. So far, some figures are mentioned for the first time in the text in brackets after an interpretation of the figure is done already. In contrast, the figure captions explain the figures in a lot of detail. I suggest providing the information of the figure captions already within the text of the manuscript.

**Author reply: We moved the detailed descriptions from the figure captions to the main text, as requested.**

**Reviewer comment:** It is definitely fair to point out that the presented measurements are a valuable contribution to the field as (to my knowledge) such experimental values do not exist for glacier ice. However, the authors should consider to not oversell their work in mentioning it many times throughout the manuscript (e.g., the title, and on Page 2 Line 41, P3 L72, P3 L89, ...). In a somewhat related issue, the title reads a bit 'bulky' and would benefit from being shortened in my opinion. However, this is of course only something to consider for the authors. In addition,

some shortcomings in explaining the applied methodology should be fixed in order to increase reading comprehension and reproducibility of the measurements (see specific comments below).

**Author reply: We removed the claim "First" from the title and throughout the text, to reduce perceptions of overselling the work. We also removed the reference to ICESat-2. The new title reads: "Spectral attenuation coefficients from measurements of light transmission in bare ice on the Greenland Ice Sheet".**

**Specific comments**

**Introduction**

• Page 2 Line 46: at this point of the introduction, it is helpful to give some values for typical ranges of air bubble and ice grain sizes.

**Author reply: We added values for near-surface glacier ice air bubble and grain-size radii, which are of the order $10^{-2}$–$10^{-3}$ m, or $10^{-1}$–$10^{-3}$ m in terms of optically-equivalent grain size (Dadic et al., 2013), as requested.**

• P3 L66: While it is true that analytical models typically assume spherical scatterers to calculate the inherent optical properties, the introduction should also point at different approaches and mention efforts by Kokhanovsky and Zege (2004) as well as Malinka (2014) and Malinka et al. (2016) which provide analytical solutions for single-scattering properties but model snow and ice grains as nonspherical.

**Author reply: Thank for these suggestions. We added a discussion of these important studies to the Introduction, as requested:**

Kokhanovsky, A. A. and Zege, E. P.: Scattering optics of snow, Appl. Opt., 43, 1589–1602, https://doi.org/10.1364/AO.43.001589, 2004.

Malinka, A.: Light scattering in porous materials: Geometrical optics and stereological approach, J. Quant. Spectrosc. Radiat. Transf., 141,3514–23, https://doi.org/10.1016/j.jqsrt.2014.02.022,, 2014.

Malinka, A., Zege, E., Heygster, G., and Istomina, L.: Reflective properties of white sea ice and snow, Cryosphere, 10, 2541–2557,https://doi.org/10.5194/tc10-2541-2016, 2016.

**Methods**

• How did you make sure the cosine receptor is in direct contact with the ice to avoid another ice-air interface? Shimming the ruler underneath the PVC tube seems to help, but have you done any testing in that regard?

**Author reply: We did not perform optical measurements to test this. Our Monte Carlo simulations quantify the impact of scattering and absorption by the rod, which include the RCR as a scattering surface with the same optical properties as the rod (Sect. S1).**

• P4 L123: This sentence reads a bit confusing to me and needs to be reformulated. Also: I don't understand the integration time given in Hertz I think a conversion to an actual time period is useful here.

**Author reply: The 44 Hz corresponds to 0.0228 s integration time per scan. We agree this is confusing and the units for integration time are now reported in seconds, as requested.**

**The revised sentence reads: "**Spectral irradiance was recorded at 1 Hz frequency. Each recorded measurement is a 20-scan average with 0.0228 s integration time per scan, yielding 0.4 s total integration time per irradiance measurement."

• I have some suggestions for Figure 1 that would help the reader in my opinion:

- the photograph of the measurement setup within Figure 1 should be enlarged (and maybe put to the right of the schematic) as, right now, it is a bit small.
- a horizontal line should clearly mark the ice surface, as due to the color gradient it is hard to distinguish from the end of the schematic.
- Can you indicate the other vertical positions of the transmittance measurements maybe with some dashed horizontal lines and then also draw an arrow indicating that the measurements were conducted from the bottom to the top?

**Author reply: Each of these changes will be made to the revised figure, as requested.**

• P5 L134: The weather situation during the measurements needs to be specified with respect to clouds, temperature, etc.

**Author reply: The weather situation is expanded upon in the revised manuscript. The conditions were overcast with light rain on 20 July and overcast with partial cloud cover during some periods of the experiment on 21 July.**

Results

• Figure 2

- It would be interesting to include the surface downwelling irradiance at $z_0$ into Figure 2a for comparison.

   **Author reply: We added the surface downwelling irradiance to Figure 2a, as requested.**

- The unit for the standard deviation is missing in Figure 2b.

   **Author reply: We added the units, as requested.**

- Figure caption: instead of naming it relative irradiance for the first time, I would name it Transmittance like in the rest of the manuscript.

   **Author reply: We changed relative irradiance to transmittance, as requested.**

• Figure 2b is not discussed in Section 3.1. Instead of stating in the Figure caption of Fig. 2b that the standard deviation is below 1 W m-2 nm-1 at all wavelengths, I would move this statement to the main text.

**Author reply: We moved this statement to the main text in Section 3.1, as requested.**

• P8 L216: Albedo is mentioned for the first time. Please explain at this point how it was calculated from the surface measurements.

**Author reply: We added the definition of albedo and how we calculated it to the Methods Sect. 2.2, as requested.**

**The sentence reads:** "The ice surface albedo was estimated as the ratio of the 2 m background upwelling spectral irradiance to the downwelling spectral irradiance. These irradiance data were smoothed with the same 1 nm

interpolation filter described above [for the in-ice irradiance measurements]. The ice surface albedo is presented in Sect. 3 to qualitatively discuss the in-ice irradiance measurements and the $k_{att}(\lambda)$ estimates."

• P8 L231-...: Warren et al. (2006) also show the snow transmission measurements before removing the absorption by impurities wouldn't using these measurements lead to a similar discrepancy between the theory and field estimate for clean snow than for the glacier ice at smaller wavelengths? I suggest to use the uncorrected measurements for snow as well within Figure 4a, so that it is consistent with the glacier ice measurements.

**Author reply: We removed this figure because it is redundant with Fig. 8 (comparison of $k_{att}$ spectra from sea ice and snowpack), as requested by reviewer 2.**

• P9 L252: The derivation of as the root-mean-squared difference between measured and predicted transmitted irradiance should already be explained in the respective Section 2.5 in the Methods.

**Author reply: We added the derivation to Sect. 2.5, as requested.**

**The sentence reads** "We estimate $\chi(\lambda)$ as the value that minimizes the root-mean-squared-difference between measured and predicted transmitted irradiance, weighted equally at all depths. In addition, we estimate a single broadband value of $\chi$ that minimizes the root-mean-squared-difference between measured and predicted transmitted irradiance, weighted equally at all depths and all $\lambda$.".

• Again P9 L252: I have a general question to the -value you applied. As you state in the caption of Figure 5, 'the spectral dependence [of the relative error] suggests a contribution of absorption to near-surface attenuation enhancement'. This calls for a spectral value of , and indeed you mention at P9 L253, 'weighted equally at all depths and all ' indicating that you derived for each individual wavelength separately. Is this the case? If so, please state already in Equation (7) that is dependent on (which is the main difference to i$_0$ in my understanding). However, I get confused with the last sentence of the paragraph on P9 L253: it reads as if you apply only one value of = 15 % to all wavelengths. Please clarify this in the text. The same applies to P12 L354.

**Author reply: We include the spectrally averaged value because large-scale models often need a single value for the visible and a single value for the infrared, or a single broadband value (Briegleb and Light, 2007; Liston and Winther, 2005). We now report spectral $\chi(\lambda)$ values in addition to the average value, as requested.**

• Figure 5: The second empirical model you use, applying $I(z_0) = I(z_{12cm})$, seems to perform best as it is only applied in the isotropic region of transmittance. Looking at the formulas in Figs. 5b and 5c for $I_z$, one could think the best possible solution for would be such that $(1) = I_{12cm}$. Which is not possible in my understanding as the exponential part of the equation (namely the 'z') is different. To avoid this confusion, the equation in Fig. 5b needs to be adapted accordingly: $I_z = I_{12cm} \exp [k(z \ 12 \ cm)]$

**Author reply: It is correct that the formula in Fig. 5b needs to include the z-12cm offset. We corrected the equation, as requested.**

• P9 L256: it is unclear to the reader how the effective $k_{att}$-values are derived using a finite-difference solution to Eq. (2) (also: contradictory, in the caption of Figure 6 it says Eq. 1)?

**Author reply: We corrected the caption of Fig. 6 to point to Eq. 2, as requested.**

**The finite difference solution takes the form:**

$$I_{\lambda,0-12 \ cm} = -\frac{1}{\Delta z}\log[I_{\lambda,z=12 \ cm} - I_{\lambda,z_0}]$$

**where $\Delta z = 12$cm.**

**We use a centered finite difference (i.e. $1/\Delta z$) to estimate a bulk attenuation value for the 0–12 cm region. We specified this in the revised text.**

• P9 L260: please state how exactly the effective $k_{att}$-values were combined for calculation of the effective penetration depth, e.g. give an equation for that.

**Author reply: We added a new Eq. 8 that shows how these values are calculated, as requested.**

**Here is the equation:**

From the definition of optical depth:

$$\tau_{\lambda,z} = \int_0^{z'} k_{\lambda,z}dz$$

we define a piecewise optical depth:

$$\tau_{\lambda,z} = \int_0^{12\ cm} k_{\lambda,0-12cm}dz + \int_{12\ cm}^{z'} k_{\lambda}dz$$

where $k_{\lambda,0-12cm}$ is the centered finite-difference estimate of $k_{att}$ for $z_{0-12cm}$ and $k_{\lambda}$ is the asymptotic $k_{att}$ for $z_{12-77cm}$ estimated from the irradiance measurements.

The e-folding depth is the depth at which $\tau = 1$. Setting $\tau_{\lambda} = 1$ in Eq. X and solving for $z'$ yields:

$$z' = \frac{1 - (k_{\lambda,0-12\ cm} \times 0.12) + (k_{\lambda} \times 0.12)}{k_{\lambda}} = 49 \text{ cm}$$

• P10 L280: Please clarify in more detail how the external diffuse specular reflectivity for a flat ice surface was calculated in this case.

**Author reply: The formulas used to calculate the external diffuse specular reflectivity are described in Section S1.2.2 of the attached supplementary document: Monte Carlo uncertainty analysis, where we use the same formulas to calculate the specular reflection from the PVC detector rod. The formulas are repeated in brief here:**

Let $\theta$ be the polar (zenith) angle between the flat ice surface and the unit normal and $\mu = \cos\theta$ be the cosine zenith angle. Following Modest (2013) Section 2.5 Eq. 2.89–2.98, the Fresnel reflectivity $R_F(\mu)$ and transmissivity $T_F(\mu)$ to incident (downward) radiation are:

$$R_F(\mu) = \frac{1}{2}\left[\left(\frac{\mu - n\mu_n}{\mu + n\mu_n}\right)^2 + \left(\frac{n\mu - \mu_n}{n\mu + \mu_n}\right)^2\right]$$

$$T_F(\mu) = 1 - R_F(\mu)$$

where $n + ik$ is the complex refractive index of ice and:

$$\mu_n = \sqrt{1 - (1 - \mu^2)/n^2}$$

is the refracted cosine zenith angle in the ice. The diffuse external reflectivity is obtained by integrating $R_F(\mu)$ across the incident hemisphere normalized by the incident flux:

$$R_F = \frac{1}{2}\int_0^1 \mu R_F(\mu)d\mu.$$

• Section 3.5: The title 'Uncertainty analysis' is misleading. It is true that the second paragraph of this section is a valuable uncertainty estimate, that should already be part of an 'Uncertainty analysis' subsection of the Section

'Methods' (compare general comment). The first paragraph of 3.5 is well-placed at this point of the manuscript, but I suggest to rename the subsection to e.g. 'Influence of ice density' after moving the second paragraph to the 'Methods'-section.

**Author reply: As requested, we have renamed Section 3.5 and we have added an additional uncertainty analysis as a new Sect. 3.6: Monte Carlo uncertainty analysis, which is included as a supplement to this document.**

• Figure 9: I suggest including an additional subsection that compares the $k_{abs}$ values of this study with previous estimates. The first time the authors mention Figure 9 is in the 'Suggestions for further work' part, which definitely undersells this comparison. This is also a very specific case for what I was mentioning in the 'General comments' section: the Figure is mostly described and discussed in the Figure caption and not in the text at all. This should be changed.

**Author reply: We added the additional subsection, as requested.**

• P13 L394: The 'Further work ...'-part of the last sentence is not useful in my opinion, as Section 4.4 already gives suggestions for future studies. I would end the 'Conclusions' section with the new values of attenuation and absorption coefficients that are provided in this study.

**Author reply: We replaced the last sentence with the new values of attenuation and absorption coefficients, as requested. The scattering coefficient is ~1.0 m$^{-1}$ in the 350–500 nm range and increases to ~5.0 m$^{-1}$ from 500–700 nm.**

• Data availability: please provide the doi of the published dataset in Pangaea.

**Author reply: The URL to the dataset is printed below. Please note that the dataset is considered "in review" as the parent work (this manuscript) remains in review. As per the policies of the data repository the DOI will remain unregistered until the work is published.**

**https://doi.pangaea.de/10.1594/PANGAEA.913508**

**Technical corrections**

• P2 Eq. (1): please indicate the spectral dependence already within the equation.

**Author reply: We added the spectral dependence to the equation.**

• P2 L47: please make sure the exponents are not split up at a line break.

**Author reply: Thank you for noting this. We replaced all units with non-breaking hyphens.**

• P2 L61: as you specify the spectral dependence of m, you should include it also for $m_{re}$ and $m_{im}$. In addition, naming them the real part and imaginary part of the complex index of refraction seems more appropriate than denoting them 'real and imaginary index'.

**Author reply: We made the requested corrections.**

• P2 L62: The authors should consider to give $k_{abs,ice}$ a separate, numbered equation.

**Author reply: As requested, we added a separate numbered equation.**

• P5 L139: The equation for the spectral transmittance should become a separate equation instead of an in-text equation.

**Author reply: As requested, we made this a separate numbered equation.**

• P6 L179: do you mean Warren and Brandt (2008)?

**Author reply: Yes, thank you.**

• P7 L183: Equation (2) does not give a direct relation to calculate $k_{att}()$. The authors should consider providing a separate equation for this purpose.

**Author reply: We added the following relation as a separate equation, as requested:**

"Estimates of $k_{att}(\lambda)$ for each 1 nm band are estimated by solving a linear equation of the form:
$$-\log T(z, \lambda) = T_0 + k_{att}(\lambda) \cdot z + \varepsilon$$
where $T_0$ is a parameter (y-intercept) that represents $T(z = 0)$ and $\varepsilon$ is an error term."

• Figure 3:

- The y-axes of Figs. 3a and 3b don't show the Transmittance T but its logarithm ln T please adjust the axes titles accordingly.

  **Author reply: The data shown in Fig. 3a and 3b are transmittance plotted on log-scale axes. The axes are labeled correctly (consider that from any point on the line, trace horizontally to the y-axis, and the value obtained in this way is transmittance, not log transmittance).**

- Legend for black line: 350 nm are stated in the text is '351 nm' a typing error?

  **Author reply: Yes, this typing error has been corrected.**

• P8 L220: the superscript 1 belongs to the unit of $k_{att}$, please keep it in one line.

**Author reply: We replaced all units with non-breaking hyphens.**

• P8 L234: reference should be to Figure 4a not Figure 5.

**Author reply: We corrected the reference.**

• P8 L236: 4 μm instead of 4 um.

**Author reply: We corrected the symbol as requested.**

• P9 L252: missing closing bracket after Eq. (7)

**Author reply: We corrected this as requested.**

• Figure 6: add y-axis title to Fig. 6b.

**Author reply:**

• P9 L273-274: don't split the unit on different lines.

**Author reply: We have replaced all units with non-breaking hyphens.**

• P10 L281: I think this should be a reference to Fig. 3b.

**Author reply: We corrected this reference to Fig. 3b.**

• P10 L287: I guess you don't mean $\omega$?

**Author reply: Thank you, this typo has been corrected. We intended to say "values of $\omega$ at wavelengths >800 nm".**

• P11 L310: to six previously published [...], not seven

**Author reply: Thank you, this has been corrected.**

• Bibliography: please add how to access the Mätzler (2002) and Perovich (1996) references.

**Author reply: These URLs have been added to the references. They are:**
Mätzler: https://boris.unibe.ch/146550/
Perovich: https://apps.dtic.mil/dtic/tr/fulltext/u2/a310586.pdf

• Appendix 1, P29 L693: the minus '-' in the unit is missing

**Author reply: Thank you, this has been corrected.**

**2 New datasets**

[Figure]

**Fig. 1:** $k_{att}$ spectra calculated with in-ice irradiance collected on 20 July at depths between 12–77 cm below the ice surface and on 21 July at depths between 53–124 cm below the ice surface. The higher attenuation in the 12–77 cm depth region is consistent with the expectation that impurities have a larger impact on visible light attenuation nearer the ice surface. The close agreement in the 600–700 nm region is consistent with the expectation that ice absorption dominates attenuation in the near infrared. The noise in the 21 July spectrum at longer wavelengths is due to the lower light levels at those depths.

**3 Dark light sensitivity**

[Figure]

**Fig. 2: Measured values of dark-light sensitivity collected with the reference spectrometer that was used to measure downwelling irradiance at 2 m height above the ice surface and with the spectrometer that was used to measure in-ice irradiance. The Ocean View software requires that a measurement of dark light is made prior to each absolute irradiance measurement, and automatically removes the dark light from the measured irradiance. The values shown in this figure represent the residual dark-light sensitivity that remained after the automated software correction. The dark light spectra shown here is subtracted from the irradiance measurements prior to fitting our experimental $k_{att}$ values. Comparison of $k_{att}$ computed with and without this dark-light correction is shown in Fig. 3.**

[Figure]

Fig. 3: (a) $k_{att}$ spectra with (I*) and without (I₀) subtracting the dark-light response spectra shown in Fig. 2. (b) the difference between the two spectra in (a), which is used as an estimate of uncertainty due to dark light response $\varepsilon_D$.

**4 Statistical variations in the high frequency measurements**

[Figure]

**Fig. 4:** (a) $k_{att}$ spectra calculated by adding and subtracting two standard deviations in the high-frequency in-ice irradiance data at each depth from the mean value at each depth and re-calculating $k_{att}$ with these modified irradiance values. (b) The difference between the two spectra in (a), which is used as an estimate of uncertainty due to high frequency variability in the irradiance measurements, $\varepsilon_{2\sigma}$.

**S1 Monte Carlo radiative transfer model**

The Monte Carlo method solves the radiative transfer equation (RTE) by simulating large ensembles of photon events represented by random samples from probability density functions (Ertürk and Howell, 2017). In this study and others, the Monte Carlo method is used to quantify relative uncertainties in imperfect optical measurements that are intractable with analytical or numerical solutions to the RTE (Gordon, 1985). We developed a Monte Carlo radiative transfer model to estimate the effect of detector interference on our irradiance measurements. The model closely follows methods developed to simulate light propagation in biological tissue, ocean waters, and sea ice (Leathers et al., 2004; Light et al., 2003; Wang et al., 1995). A general description of the model and particular modifications for this investigation are described below.

**S1.1 Probability functions for optical properties**

The fundamental ingredients of this and other Monte Carlo radiative transfer models are the inherent optical properties k, ω, and g (see Sect. 2.3 of the main), the geometric boundary conditions, and the probabilistic rules that govern the system. The cumulative probability of occurrence for an event x, with probability density function $p(x)$, is:

$$P(x) = \int_{-\infty}^{x} p(x) dx, \quad 0 \leq P(x) \leq 1. \tag{1}$$

To solve for x, the left-hand-side (LHS) of (1) is replaced with a random number:

$$P(x) = q \tag{2}$$

where $q$ is from the uniform distribution over [0,1]. The right-hand-side (RHS) lower limit of integration $-\infty$ is replaced with an appropriate limit (e.g., 0) and analytic or empirical expressions for $p(x)$ are specified.

In this study, x represents optical path length, scattering direction, and photon survival probability. Closed-form expressions for each of these terms are given in the following sections.

**S1.1.1 Optical path length**

The probability density function for the optical path length $l$ [m$^{-1}$] is given by the e-folding length:

$$p(l) = e^{-l}, \quad l \geq 0 \tag{3}$$

with the cumulative distribution function:

$$P(l) = \int_{0}^{l} e^{-l'} dl' = 1 - e^{-l}. \tag{4}$$

From Eq. (2), $q = 1 - e^{-l}$ and therefore:

$$l = -\ln q, \quad 0 \leq 1. \tag{5}$$

In this study, $q$ is generated with the MATLAB function rand.

The photon transport length [m] is the optical path length scaled by the extinction coefficient:

$$s = l/\sigma_e \tag{6}$$

where:

$$\sigma_e = \sigma_s + \sigma_a \tag{7}$$

is the single-scattering extinction coefficient, $\sigma_s$ [m$^{-1}$] is the scattering coefficient, and $\sigma_a$ [m$^{-1}$] is the absorption coefficient.

**S1.1.2 Scattering phase function**

The probability density function for a scattering phase function with azimuthal symmetry is:

$$p(\theta_s) = 2\pi\tilde{\beta}(\theta_s)\sin\theta \tag{8}$$

where $\tilde{\beta}(\theta_s)$ is the probability that a photon will scatter at polar angle $\theta_s$. We specify $\tilde{\beta}(\theta_s)$ with the Henyey-Greenstein scattering phase function, which is appropriate for strongly forward scattering by ice grains and air bubbles (Light et al., 2003):

$$\tilde{\beta}(g, \theta_s) = \frac{1}{4\pi}\frac{1 - g^2}{(1 + g^2 - 2g\cos\theta_s)^{\frac{3}{2}}}, \qquad -1 < g < 1. \tag{9}$$

where $g = 0$ reduces Eq. 9 to isotropic scattering and $g \to 1$ is strongly forward scattering. In this study, $g = 0.86$, as given by Mullen & Warren (1988) from Mie theory calculations for scattering by air bubbles in ice.

From Eq. (1):

$$P(\theta_s) = -\frac{1 - g^2}{2}\int_0^{\theta_s}\frac{\sin\theta_s'}{(1 + g^2 - 2g\cos\theta_s')^{\frac{3}{2}}}d\theta_s' = q \tag{10}$$

which evaluates to:

$$q = \frac{1 - g^2}{2g}\left[\frac{1}{1 - g} - \frac{1}{\sqrt{1 + g^2 - 2g\cos\theta_s}}\right] \tag{11}$$

yielding the scattering angle:

$$\cos\theta_s = \frac{1}{2g}\left[1 + g^2 - \left(\frac{1 - g^2}{1 - g + 2gq}\right)^2\right], \qquad g \neq 0;\ 0 \leq \theta_s \leq \pi/2. \tag{12}$$

The probability density function for scattering azimuth angle $\phi_s$ in a spherical coordinate system with azimuthal symmetry is $1/2\pi$. From Eq. (1):

$$P(\phi_s) = \frac{\phi_s}{2\pi}, \qquad 0 \leq \phi_s \leq 2\pi \tag{13}$$

and from Eq. (2):

$$\phi_s = 2\pi q. \tag{14}$$

**S1.1.3 Photon termination**

Monte Carlo simulations are computationally expensive. To improve computational performance, photons are treated as packets of photons with initial weight $w = 1$. At each interaction, photons are scattered and absorbed according to their respective statistical probabilities, parameterized by $\sigma_s$ and $\sigma_a$. Accordingly, at each interaction the weight is updated as:

$$w = (1 - \overline{\omega}) \cdot w \qquad (15)$$

where:

$$\overline{\omega} = \sigma_s/\sigma_e \qquad (16)$$

is the single-scattering albedo [-]. Each $1 - \overline{\omega}$ reduction in photon packet weight is proportional to the probability of an individual photon absorption event. After many interactions, if $w$ drops below a very small value it contributes very little to the solution. The so-called "Russian roulette" technique is used to improve computational performance, where photon packet weights below a specified threshold $w < w_{min}$ are increased in proportion to a survival probability function and are re-released into the medium, or otherwise terminated:

$$w = \begin{cases} m \cdot w, & q \leq 1/m \\ 0, & q > 1/m \end{cases} \qquad (17)$$

where $1/m$ is the probability of photon survival and $q$ is a random number as previously defined. This technique conserves energy and is unbiased (Wang et al., 1995). In this study, $w_{min} = 10^{-5}$ and $m = 10$.

At each interaction, the absorbed fraction $\overline{\omega} \cdot w$ is scored into an absorption array in a cylindrical coordinate system that is used to compute observable quantities of absorption and photon fluence. If a photon packet exits the medium, it is scored into a transmittance or reflectance array in an azimuthally independent spherical coordinate system that is used to compute observable quantities of irradiance, radiant intensity, and power. These scoring systems follow the definitions in Wang et al. (1995) Eq. 4.1–4.32.

The preceding sections describe the fundamental processes of photon transport, scattering direction, and survival probability. Similar probability density functions that describe the detector rod interference are described next.

**S1.2 Monte Carlo experiment**

The detector rod interference is estimated with a "backward" Monte Carlo (BMC) simulation, which simulates photon trajectories starting from the detector backward to the target (Leathers et al., 2004; Light et al., 2003). Here, the target is the ice surface. The simulation domain is a 3-dimensional ice slab with one boundary, the ice surface, and otherwise infinite horizontal and vertical extent. A cylinder with dimensions identical to the detector rod is placed at positions identical to the measurement depths reported in this paper, and photon packets are released from the irradiance sensor ("remote cosine receptor") located on the detector rod (Fig. 5).

[Figure]

**Fig. 5: Example Monte Carlo photon tracking simulation from model output used in this study, with interference by cylindrical detector rod. (a) ~14,000 random photon interactions are traced within a 3-dimensional ice volume. The cylindrical object represents the detector rod, here inserted at 1 m below the ice surface. The photon packet is released from the position of the irradiance sensor ("remote cosine receptor") located on the rod and traced backward to the ice surface ("backward Monte Carlo"). (b) Magnified view of the detector rod in the y-z plane shows photon packets scattering off of the rod. The color-bar represents the number of cumulative interactions experienced by this photon packet.**

As described above, each interaction within the ice volume is defined by absorption and scattering of the photon by ice. Absorption reduces the photon energy density by an amount $1 - \overline{\omega}_{\text{ice}}$. Scattering redirects the photon trajectory according to the Henyey-Greenstein scattering phase function with asymmetry parameter $g$ and transport distance $l$. Photon interactions with the detector rod require additional specifications that are described next.

**S1.2.1 Source function for cosine detector**

The scattering phase function for an irradiance sensor with a cosine response is:

$$\tilde{\beta}(\theta) = \frac{\cos\theta}{\pi}, \qquad 0 \leq \theta \leq \frac{\pi}{2} \tag{18}$$

with probability density function:

$$p(\theta) = 2\pi\tilde{\beta}(\theta)\sin\theta \tag{19}$$

and cumulative distribution function:

$$P(\theta) = 2\int_0^\theta \cos\theta'\sin\theta'\,d\theta' = q\,. \tag{20}$$

Substituting $\mu = \cos(\theta)$ the scattering angle is:

$$\cos\theta = \sqrt{1-q}. \tag{21}$$

For a forward Monte Carlo simulation, Eq. 21 gives the probability of photon receipt by an irradiance sensor with an ideal cosine response. For a BMC simulation, the form of Equation 21 that gives the initial launch trajectory of photons from the irradiance sensor surface is:

$$\cos\theta = -\sqrt{q}. \tag{22}$$

In reality, irradiance sensors do not have an ideal cosine response to radiance. In this experiment, the non-ideal cosine response of the irradiance sensor is estimated by replacing Eq. 22 with uniform sampling from an empirical probability density function derived from laboratory measurements of the cosine receptor angular response function provided by Ocean Optics (Fig. 6). The source azimuth angle $\phi$ is determined with Eq. (14).

[Figure]

**Fig. 6: (a) Comparison of ideal angular response function (ideal cosine) with the empirical angular response function used to estimate the non-ideal response of the irradiance sensor used in this study. The empirical angular response function was developed by Ocean Optics from laboratory measurements on the same irradiance sensor type used in this study. (b) Same as (a) but normalized. The red line in (b) is the empirical probability density function used as the irradiance source function for our backward-Monte Carlo simulations (see Eq. 21–22).**

**S1.2.2 Scattering and absorption by detector rod**

If a photon trajectory crosses the 3-dimensional position of the detector rod, the photon energy density is reduced by an amount $1 - \omega_{rod}$ and the photon is scattered away from the rod (Fig. 5) with an isotropic scattering phase function:

$$\theta_s = 1 - 2q, \tag{23}$$

$$\phi_s = 2\pi q. \tag{24}$$

The collision point is determined with ray tracing formulas that equate the vector equation of the photon trajectory with the parametric equation for the cylindrical detector rod surface following Ertürk and Howell (2017) Sect. 7.1 Eq. 59–66.

The polyvinyl chloride (PVC) detector rod albedo $\omega_{rod}$ is estimated from values for the complex refractive index of PVC (Zhang et al., 2020). Let $\mu = \cos\theta$ be the cosine zenith angle of incident radiation with $\mu = +1$ vertically

downward. Following Modest (2013) Section 2.5 Eq. 2.89–2.98, the Fresnel reflectivity and transmissivity to incident (downward) radiation are:

$$R_F(\mu) = \frac{1}{2}\left[\left(\frac{\mu - n\mu_n}{\mu + n\mu_n}\right)^2 + \left(\frac{n\mu - \mu_n}{n\mu + \mu_n}\right)^2\right]$$

(25)

$$T_F(\mu) = 1 - R_F(\mu)$$

(26)

where $n + ik$ and $n_0 + ik_0$ are the complex refractive indices of PVC and air, respectively, and:

$$\mu_n = \sqrt{1 - (1 - \mu^2)/n^2}$$

(27)

is the refracted cosine zenith angle in the PVC pipe. Radiation transmitted into the PVC is attenuated exponentially:

$$a(\mu_n) = e^{-\tau/\mu_n}$$

(28)

where:

$$\tau = 4\pi kL/\lambda$$

(29)

is the optical thickness of the PVC pipe with wall thickness $L = 0.004$ m. Radiation that transmits through $L$ is internally reflected upward from the inner wall in the direction $\mu_n$ and attenuated exponentially along path length $\tau$. Radiation that reaches the outer wall at $\mu_n < \mu_c$ is transmitted across the outer wall according to $T_F(\mu_n)$ and reflected back into the PVC according to $R_F(\mu_n)$, where $\mu_c$ is the critical angle given by Snell's law:

$$\mu_c = \sqrt{1 - 1/n^2}.$$

(30)

Formulas for $T_F(\mu_n)$ and $R_F(\mu_n)$ are similar to Eq. 25 and Eq. 26 with modifications for total internal reflection about $\mu_c$ and are given elsewhere (Briegleb and Light, 2007; Liou, 2002).

The total reflectivity is estimated with the successive-order-of-scattering method (van de Hulst, 1980), which accounts for the multiple internal reflections and absorption within the PVC described by Eq. 25–30. We model the PVC pipe as a plane, which is justified because the radius of curvature is much larger than all wavelengths of light considered here. For the geometry and optical properties of the detector rod, the total reflectivity has the closed-form solution:

$$R_d = R_{F,\mu} + \frac{T_{F,\mu}R_{F,\mu_n}T_{F,\mu_n}\,a_{\mu_n}^2}{1 - R_{F,\mu_n}a_{\mu_n}^2}$$

(31)

where the subscripts $\mu$ and $\mu_n$ on $R$, $T$, and a indicate the direction of incident radiance.

**S2 Model validation**

The Monte Carlo model described above is verified by comparison with benchmark values for total diffuse reflectance $R_d$ [W m$^{-2}$], total transmittance $T_t$ [W m$^{-2}$], diffuse angular reflectance $R_d(\alpha)$ [W sr$^{-1}$] and diffuse angular transmittance $T_d(\alpha)$ [W sr$^{-1}$] tabulated by van de Hulst (1980). The angular quantities, which have units of radiant intensity, are defined with respect to the exiting angle normal to the surface $\alpha$ [rad]. For a plane-parallel slab with optical properties $\sigma_s = 0.9$ m$^{-1}$, $\sigma_a = 0.1$ m$^{-1}$, $g = 0.75$, and optical thickness $\tau = 2$, the van de Hulst (1980) solutions are $R_d = 0.09739$ and $T_t = 0.66096$. For an ensemble of $N = 100$ simulations, the Monte Carlo model described

above gives $R_d = 0.09740 \pm 0.00034$ and $T_t = 0.66098 \pm 0.00049$ ($\mu \pm 1\sigma$). The model closely reproduces the benchmark solutions for $R_d(\alpha)$ and $T_d(\alpha)$ (Fig. 7).

[Figure]

**Fig. 7: Values of diffuse angular reflectance (radiant intensity), $R_d(\alpha)$ and transmittance $T_d(\alpha)$ vs. the photon exiting angle with respect to the surface normal α (after Wang et al. 1995 Fig. 3). Solid circles are benchmark solutions from Table 35 in van de Hulst (1980), obtained with the doubling method of solution to the radiative transfer equation.**

**S3 Monte Carlo uncertainty estimate**

The Monte Carlo model is used to estimate the effect of detector interference on our irradiance measurements and, in turn, the asymptotic flux attenuation coefficients $k_{att}$ that are estimated from them. To this end, we designed four experiments that isolate two forms of detector interference: 1) the non-ideal cosine response of the irradiance detector, and 2) absorption and scattering by the PVC detector rod. The four experiments, including a base simulation with no detector interference, are summarized in Table S1.

**Table S1: Summary of four Monte Carlo experiments that simulate the effect of the detector rod interference on in-ice irradiance measurements. The baseline simulation (ideal diffusion, no rod) has no detector interference.**

| Experiment | Source function | Detector absorption | Detector scattering |
|---|---|---|---|
| Ideal Diffusion, No Rod | Eq. 23-24 | - | - |
| Ideal Cosine, No Rod | Eq. 22 | - | - |
| Ideal Cosine, With Rod | Eq. 22 | $\omega_{rod}$ | Eq. 23-24 |
| Non-ideal Cosine, With Rod | Empirical (Fig. 6) | $\omega_{rod}$ | Eq. 23-24 |

For each experimental setup, the Monte Carlo is integrated across 10,000 interactions at four wavelengths (400 nm, 500 nm, 600 nm, and 700 nm) with detector rod positions that are identical to the measurement depths reported in this paper (c.f. Fig. 5). For the 20 July experiment, these depths are 9.35 cm, 30.0 cm, 50.45 cm, and 68.60 cm, in units of solid-ice equivalent (i.e., physical thickness scaled by measured ice density). For the 21 July experiment, these depths

are 45.93 cm, 58.98 cm, 73.40 cm, and 114.5 cm. Monte Carlo $k_{\text{att}}$ values are estimated for each wavelength with the same method used for the field-estimates, i.e., by least-squares linear regression:

$$-\log T(z, \lambda) = T_0 + k_{\text{att}}(\lambda)z + \varepsilon \tag{32}$$

where $T$ is the total diffuse transmittance from Monte Carlo simulation (see Section S2), $T_0$ is a parameter (y-intercept) that represents $T(z = 0)$ and $\varepsilon$ is an error term.

These simulations provide two measures of $k_{\text{att}}$ uncertainty: 1) the difference between the average Monte Carlo $k_{\text{att}}$ value $\mu_{\text{MC}}$ and the field-estimated $k_{\text{att}}$ value at each wavelength, and 2) the spread among Monte Carlo $k_{\text{att}}$ values at each wavelength. The spread among Monte Carlo $k_{\text{att}}$ values is an estimate of uncertainty due to the irradiance sensor angular response function and the detector rod interference. The spectrometer dark-light sensitivity is an additional source of instrumental uncertainty that is estimated from field measurements as described in Sect. X. These instrumental uncertainties (irradiance sensor angular response, detector rod interference, and dark-light sensitivity) are compared with the statistical variations in the high-frequency irradiance measurements and with the statistical uncertainty in the $k_{\text{att}}$ linear regression model (Eq. 32).

If we take the spread among the Monte Carlo $k_{\text{att}}$ values as independent of the uncertainty estimates obtained from analysis of field datasets and the uncertainty in the linear regression model, a combined uncertainty is estimated as:

$$\varepsilon = \sqrt{\varepsilon_{MC}^2 + \varepsilon_D^2 + \varepsilon_{2\sigma}^2 + \varepsilon_{LM}^2} \tag{33}$$

where $\varepsilon_{MC}$, $\varepsilon_D$, $\varepsilon_{2\sigma}$, and $\varepsilon_{LM}$ are the uncertainty from Monte Carlo, dark-current sensitivity, high-frequency statistical variability, and linear model statistical uncertainty, defined as one standard error in the $k_{\text{att}}$ linear regression (Fig 8).

[Figure]

**Fig 8: Attenuation coefficient $k_{\text{att}}$ spectra from measurements of light transmission collected on 20 July, 2018, compared with average $k_{\text{att}}$ values from four simulations with a 3-dimensional Monte Carlo radiative transfer model ($\mu_{MC}$) and with two measures of uncertainty: 1) statistical linear model uncertainty $\varepsilon_{LM}$ (shaded uncertainty bounds; $\pm 1$ standard error in the linear regression) and, 2) $\varepsilon_{LM}$ combined with instrumental and measurement uncertainty (error bars; $\mu_{MC} \pm \varepsilon$). The combined estimate combines $\varepsilon_{LM}$ with uncertainty due to spectrometer dark-light sensitivity, non-ideal cosine response of the irradiance sensor, detector rod interference, and statistical variations in the high-frequency raw data ($\pm 2$ standard deviations).**

**References**

Briegleb, B. P. and Light, B.: A Delta-Eddington Mutiple Scattering Parameterization for Solar Radiation in the Sea Ice Component of the Community Climate System Model, Technical Note, National Center for Atmospheric Research, Boulder, Colorado. [online] Available from: http://dx.doi.org/10.5065/D6B27S71 (Accessed 18 February 2019), 2007.

Ertürk, H. and Howell, J. R.: Monte Carlo Methods for Radiative Transfer, in Handbook of Thermal Science and Engineering, edited by F. A. Kulacki, pp. 1–43, Springer International Publishing, Cham., 2017.

Gordon, H. R.: Ship perturbation of irradiance measurements at sea 1: Monte Carlo simulations, Appl. Opt., 24(23), 4172, doi:10.1364/AO.24.004172, 1985.

van de Hulst, H. C.: Multiple light scattering: tables, formulas, and applications, Academic Press, New York., 1980.

Leathers, R. A., Downes, T. V., Davis, C. O. and Mobley, C. D.: Monte Carlo Radiative Transfer Simulations for Ocean Optics: A Practical Guide, Memorandum, Naval Research Laboratory, Washington, D.C. [online] Available from: https://www.oceanopticsbook.info/packages/iws_l2h/conversion/files/Leathersetal_NRL2004.pdf (Accessed 11 October 2020), 2004.

Light, B., Maykut, G. A. and Grenfell, T. C.: A two-dimensional Monte Carlo model of radiative transfer in sea ice, J. Geophys. Res. Oceans, 108(C7), 3219, doi:10.1029/2002JC001513, 2003.

Liou, K.-N.: An introduction to atmospheric radiation, 2nd ed., Academic Press, Amsterdam ; Boston., 2002.

Modest, M. F.: Radiative heat transfer, Third Edition., Academic Press, New York., 2013.

Mullen, P. C. and Warren, S. G.: Theory of the optical properties of lake ice, J. Geophys. Res. Atmospheres, 93(D7), 8403–8414, doi:10.1029/JD093iD07p08403, 1988.

Wang, L., Jacques, S. L. and Zheng, L.: MCML—Monte Carlo modeling of light transport in multi-layered tissues, Comput. Methods Programs Biomed., 47(2), 131–146, doi:10.1016/0169-2607(95)01640-F, 1995.

Zhang, X., Qiu, J., Li, X., Zhao, J. and Liu, L.: Complex refractive indices measurements of polymers in visible and near-infrared bands, Appl. Opt., 59(8), 2337, doi:10.1364/AO.383831, 2020.

---

## Author Response (AR1)

**1 Response to Anonymous Referee #1**

**Author reply: Thank you for your detailed comments on our manuscript. Below, please find the original reviewer concern/question in italic typeface followed by our response in regular typeface.**

**Please note: Response to Anonymous Referee #2 is included within this document, after this reponse.**

**General comments**

*Within the manuscript, spectral measurements of light attenuation in Greenland Ice Sheet bare ice are presented. For this purpose, the authors employed spectral irradiance measurements between 350-900 nm wavelength at the surface and at four different depths below the ice surface and calculated the spectral transmittance within the ice. From this, spectral flux attenuation and absorption coefficients are derived and compared to previous studies*

*The manuscript is clearly structured and the figures are of good quality, which helps to convey the arguments of the authors. The measurements of in-ice spectral transmittance are very valuable, and the author's efforts to put them into context and to identify possible future applications should be acknowledged. However, there are some aspects that need further focus in my opinion. After some general comments, the more specific comments and suggestions for technical corrections follow below.*

*In my point of view, the most pressing aspect is the lack of accounting for measurement uncertainties within the manuscript. So far, only statistical variations along the 30 measured irradiance spectra are considered, which need to be clearly separated from instrument uncertainties. However, the latter are not mentioned at all within the manuscript.*

*I suggest to include a new subsection within the 'Methods'-section that is devoted to instrument and measurement uncertainties. I understand the instrument is calibrated for irradiance measurements, but instrument errors such as the influence of dark and stray light, the wavelength calibration of the spectrometer, the non-ideal cosine response of the RCR diffuser, and the uncertainty of the absolute calibration (to get calibrated irradiance measurements) need to be quantified. Furthermore, as the transmittance is measured calculating the ratio of measured irradiance from two different instruments (the one in the ice, and the permanent one at the surface), differences between the two instruments need to be quantified (e.g., by means of a cross-calibration with the same light source). These instrumental errors need to be put in relation to uncertainties regarding the measurement setup (e.g., is the cosine receptor really in close contact with the ice during the in-ice irradiance measurements?) and the statistical variations as deduced from the 30 subsequently measured spectra.*

**Author reply:** We added a new subsection within the 'Methods' devoted to instrument and measurement uncertainties. Specifically, we address:
- Dark noise sensitivity
- Cross-calibration of the two spectrometers
- Non-ideal cosine response of the RCR diffuser
- Absorption interference by the PVC detector rod

To address the last two points, we developed a 3-dimensional Monte Carlo radiative transfer model described in the attached Supplementary Material.

Based on analysis of field datasets and Monte Carlo simulation, we estimate a combined statistical and systematic uncertainty on our $k_{att}$ values that is <20% for wavelengths between 350–450 nm and as low as ~5% for wavelengths >450 nm.

**Reviewer comment:** *This uncertainty analysis should eventually lead to vertical uncertainty bars in Figure 3, together with the horizontal uncertainty bars stemming from the depth measurements with the ruler (this second paragraph of Section 3.5 should also be moved to the new uncertainty subsection in Section 'Methods'). The errors in the linear regression to derive the flux attenuation coefficient should consider both depth and transmittance uncertainty estimates and, eventually, should lead to an uncertainty range attributed to each $k_{att}$ value. A thorough treatment of the measurement uncertainties will definitely increase the value of the measurements for further applications.*

**Author reply:** As requested, the second paragraph of Section 3.5 was moved to the 'Methods' (in this case, we added a new subheading 'Ice thickness and density').

As requested, Figure 3 (revised Figure 5) now contains both vertical and horizontal error bars representing uncertainty in depth and transmittance. These new estimates of uncertainty in both depth and transmittance are used in the revised analysis to obtain an unbiased estimate of the slope ($k_{att}$) and associated standard errors of the linear relationship between depth and transmittance using Maximum Likelihood Estimation, rather than Ordinary Least Squares (York et al., 2004).

**Reviewer comment:** *Another remark is more structural and concerns the introduction of figures in the text: Some figures need to be described and explained in more detail within the text already. So far, some figures are mentioned for the first time in the text in brackets after an interpretation of the figure is done already. In contrast, the figure captions explain the figures in a lot of detail. I suggest providing the information of the figure captions already within the text of the manuscript.*

**Author reply:** We moved the detailed descriptions from the figure captions to the main text, as requested.

**Reviewer comment:** *It is definitely fair to point out that the presented measurements are a valuable contribution to the field as (to my knowledge) such experimental values do not exist for glacier ice. However, the authors should consider to not oversell their work in mentioning it many times throughout the manuscript (e.g., the title, and on Page 2 Line 41, P3 L72, P3 L89, ...). In a somewhat related issue, the title reads a bit 'bulky' and would benefit from being shortened in my opinion. However, this is of course only something to consider for the authors. In addition, some shortcomings in explaining the applied methodology should be fixed in order to increase reading comprehension and reproducibility of the measurements (see specific comments below).*

**Author reply:** We removed the claim "First" from the title and throughout the text, to reduce perceptions of overselling the work. We also removed the reference to ICESat-2. The new title reads: "Spectral attenuation coefficients from measurements of light transmission in bare ice on the Greenland Ice Sheet".

**Specific comments**

**Introduction**

• *Page 2 Line 46: at this point of the introduction, it is helpful to give some values for typical ranges of air bubble and ice grain sizes.*

**Author reply:** As requested, we added values for near-surface glacier ice air bubble and grain-size radii, which are of the order $10^{-2}$–$10^{-3}$ m, or $10^{-1}$–$10^{-3}$ m in terms of optically-equivalent grain size (Dadic et al., 2013).

• *P3 L66: While it is true that analytical models typically assume spherical scatterers to calculate the inherent optical properties, the introduction should also point at different approaches and mention efforts by Kokhanovsky and Zege (2004) as well as Malinka (2014) and Malinka et al. (2016) which provide analytical solutions for single-scattering properties but model snow and ice grains as nonspherical.*

**Author reply:** Thank for these suggestions. We added a discussion of these important studies to the Introduction, as requested:

Kokhanovsky, A. A. and Zege, E. P.: Scattering optics of snow, Appl. Opt., 43, 1589–1602, https://doi.org/10.1364/AO.43.001589, 2004.

Malinka, A.: Light scattering in porous materials: Geometrical optics and stereological approach, J. Quant. Spectrosc. Radiat. Transf., 141,3514–23, https://doi.org/10.1016/j.jqsrt.2014.02.022,, 2014.

Malinka, A., Zege, E., Heygster, G., and Istomina, L.: Reflective properties of white sea ice and snow, Cryosphere, 10, 2541–2557,https://doi.org/10.5194/tc10-2541-2016, 2016.

**Methods**

• *How did you make sure the cosine receptor is in direct contact with the ice to avoid another ice-air interface? Shimming the ruler underneath the PVC tube seems to help, but have you done any testing in that regard?*

**Author reply:** We did not perform optical measurements to test this. Our Monte Carlo simulations quantify the impact of scattering and absorption by the rod, which include the RCR as a scattering surface with the same optical properties as the rod (Sect. S1).

• *P4 L123: This sentence reads a bit confusing to me and needs to be reformulated. Also: I don't understand the integration time given in Hertz I think a conversion to an actual time period is useful here.*

**Author reply:** The 44 Hz corresponds to 0.0228 s integration time per scan. We agree this is confusing. As requested, the units for integration time are now reported in seconds:

"Spectral irradiance was recorded at 1 Hz frequency. Each recorded measurement is a 20-scan average with 0.0228 s integration time per scan, yielding 0.4 s total integration time per irradiance measurement."

• *I have some suggestions for Figure 1 that would help the reader in my opinion:*

- *the photograph of the measurement setup within Figure 1 should be enlarged (and maybe put to the right of the schematic) as, right now, it is a bit small.*
- *a horizontal line should clearly mark the ice surface, as due to the color gradient it is hard to distinguish from the end of the schematic.*
- *Can you indicate the other vertical positions of the transmittance measurements maybe with some dashed horizontal lines and then also draw an arrow indicating that the measurements were conducted from the bottom to the top?*

**Author reply:** We enlarged the figure and added a horizontal line at the ice surface, which is also labeled on the figure in text as "ice surface". We did not add the dashed lines, they detract from the clarity of the figure.

• *P5 L134: The weather situation during the measurements needs to be specified with respect to clouds, temperature, etc.*

**Author reply:** As requested, we expanded our description of the weather situation in the revised manuscript in the new 'Methods' subsection 2.2 'Weather Conditions'. The conditions were overcast with light rain on 20 July and overcast with partial cloud cover during some periods of the experiment on 21 July.

**Results**

• *Figure 2*

- *It would be interesting to include the surface downwelling irradiance at $z_0$ into Figure 2a for comparison.*

  **Author reply:** As requested, we added the surface downwelling irradiance to Figure 2a (revised Fig. 4a).

- *The unit for the standard deviation is missing in Figure 2b.*

  **Author reply:** The standard deviation is now shown as shaded uncertainty bounds on the plotted values.

- *Figure caption: instead of naming it relative irradiance for the first time, I would name it Transmittance like in the rest of the manuscript.*

  **Author reply:** As requested, we changed relative irradiance to transmittance.

• *Figure 2b is not discussed in Section 3.1. Instead of stating in the Figure caption of Fig. 2b that the standard deviation is below 1 W m-2 nm-1 at all wavelengths, I would move this statement to the main text.*

**Author reply:** As requested, we moved this statement to the main text in Section 3.1.

• *P8 L216: Albedo is mentioned for the first time. Please explain at this point how it was calculated from the surface measurements.*

**Author reply:** As requested, we added the definition of albedo and how we calculated it to the Methods Sect. 2.4.

• *P8 L231-...: Warren et al. (2006) also show the snow transmission measurements before removing the absorption by impurities wouldn't using these measurements lead to a similar discrepancy between the theory and field estimate for clean snow than for the glacier ice at smaller wavelengths? I suggest to use the uncorrected measurements for snow as well within Figure 4a, so that it is consistent with the glacier ice measurements.*

**Author reply:** As requested, we added the uncorrected (contaminated) snow values to Fig. 9 (comparison of $k_{att}$ spectra from literature review). We removed the figure that this comment references (Fig. 4b) because it is redundant with Fig. 8 (comparison of $k_{abs}$ spectra, revised Fig. 12), as requested by reviewer 2.

• *P9 L252: The derivation of as the root-mean-squared difference between measured and predicted transmitted irradiance should already be explained in the respective Section 2.5 in the Methods.*

**Author reply:** We removed this method altogether. Rather than compute a best-fit spectral value, we use the 0-12 cm effective attenuation coefficient in a piece-wise Bouguer model to demonstrate the impact of the near-surface attenuation, see below for further detail.

• *Again P9 L252: I have a general question to the -value you applied. As you state in the caption of Figure 5, 'the spectral dependence [of the relative error] suggests a contribution of absorption to near-surface attenuation enhancement'. This calls for a spectral value of , and indeed you mention at P9 L253, 'weighted equally at all depths and all ' indicating that you derived for each individual wavelength separately. Is this the case? If so, please state already in Equation (7) that is dependent on (which is the main difference to $i_0$ in my understanding). However, I get confused with the last sentence of the paragraph on P9 L253: it reads as if you apply only one value of = 15 % to all wavelengths. Please clarify this in the text. The same applies to P12 L354.*

**Author reply:** As requested, we clarified these issues in the revised manuscript. Specifically, we use the effective attenuation coefficient for the 0-12 cm layer to demonstrate the spectral variation in near-surface attenuation. In

addition, we report values for the $i_o$ parameter using the formal definition given by Grenfell and Maykut, (1977). This parameter is a broadband value. It is simply the fraction of irradiance absorbed in the upper 10 cm of ice, weighted by the incoming irradiance (integrated over a given spectral range). This is described clearly in the revised methods. The reason we report this value is because several large-scale models apply simple radiative transfer parameterizations that use $i_o$ to put the correct amount of absorbed radiation in the upper grid cell of the vertical ice column (Briegleb and Light, 2007; Liston and Winther, 2005). With our reported values, any user will be able to compute their own $i_o$ for any other ice thickness or spectral region.

• *Figure 5: The second empirical model you use, applying $I(z_0) = I(z_{12cm})$, seems to perform best as it is only applied in the isotropic region of transmittance. Looking at the formulas in Figs. 5b and 5c for $I_z$, one could think the best possible solution for would be such that $(1\ ) = I_{12cm}$. Which is not possible in my understanding as the exponential part of the equation (namely the 'z') is different. To avoid this confusion, the equation in Fig. 5b needs to be adapted accordingly: $I_z = I_{12cm} \exp[k(z\ 12\ cm)]$*

**Author reply:** It is correct that the formula in Fig. 5b needs to include the z-12cm offset. However, we rewrote this equation in piecewise form. Note that the new form is functionally identical to the one used previously:

$$I_z(\lambda) = I_0(\lambda) \exp\left[-\int_0^{z'} k'(\lambda)\,dz - \int_{z'}^{z} k_{att}(\lambda)\,dz\right]$$

• *P9 L256: it is unclear to the reader how the effective $k_{att}$-values are derived using a finite-difference solution to Eq. (2) (also: contradictory, in the caption of Figure 6 it says Eq. 1)?*

**Author reply:** As requested, we clarified how the effective $k_{att}$ values are computed (see below) and we corrected the caption of Fig. 6 to point to the correct equation, as requested.

• *P9 L260: please state how exactly the effective $k_{att}$-values were combined for calculation of the effective penetration depth, e.g. give an equation for that.*

**Author reply:** As requested, we added a new equation that shows exactly how these values are calculated.

• *P10 L280: Please clarify in more detail how the external diffuse specular reflectivity for a flat ice surface was calculated in this case.*

**Author reply:** The external diffuse specular reflectivity for a flat ice surface was calculated using Eq. 20–24 of Briegleb and Light, (2007). The formulas are described in more detail Section S1.2.2 of the attached supplementary document that describes the Monte Carlo model, where we use the same formulas to calculate the reflectivity of the PVC detector rod.

• *Section 3.5: The title 'Uncertainty analysis' is misleading. It is true that the second paragraph of this section is a valuable uncertainty estimate, that should already be part of an 'Uncertainty analysis' subsection of the Section 'Methods' (compare general comment). The first paragraph of 3.5 is well-placed at this point of the manuscript, but I suggest to rename the subsection to e.g. 'Influence of ice density' after moving the second paragraph to the 'Methods'-section.*

**Author reply:** As requested, we moved the second paragraph of Section 3.5 to the revised Methods Section 2.3: "Ice thickness and density".

• *Figure 9: I suggest including an additional subsection that compares the kabs values of this study with previous estimates. The first time the authors mention Figure 9 is in the 'Suggestions for further work' part, which definitely undersells this comparison. This is also a very specific case for what I was mentioning in the 'General comments'*

*section: the Figure is mostly described and discussed in the Figure caption and not in the text at all. This should be changed.*

**Author reply:** As requested, we added an in-depth discussion of the $k_{abs}$ values and compare them with the previous estimates shown in Fig. 9 (revised Fig. 12). The revised Discussion section takes on a narrative format without dedicated subsections as each paragraph is logically linked, therefore we did not add a dedicated subsection.

*• P13 L394: The 'Further work ...'-part of the last sentence is not useful in my opinion, as Section 4.4 already gives suggestions for future studies. I would end the 'Conclusions' section with the new values of attenuation and absorption coefficients that are provided in this study.*

**Author reply:** As requested, we replaced the last sentence with the new values of attenuation and absorption coefficients.

*• Data availability: please provide the doi of the published dataset in Pangaea.*

**Author reply:** The URL to the dataset is printed below. Please note that the dataset is considered "in review" as the parent work (this manuscript) remains in review. As per the policies of the data repository the DOI will remain unregistered until the work is published.

https://doi.pangaea.de/10.1594/PANGAEA.913508

**Technical corrections**

*• P2 Eq. (1): please indicate the spectral dependence already within the equation.*

**Author reply:** We added the spectral dependence to the equation.

*• P2 L47: please make sure the exponents are not split up at a line break.*

**Author reply:** Thank you for noting this; we replaced all units with non-breaking hyphens.

*• P2 L61: as you specify the spectral dependence of m, you should include it also for $m_{re}$ and $m_{im}$. In addition, naming them the real part and imaginary part of the complex index of refraction seems more appropriate than denoting them 'real and imaginary index'.*

**Author reply:** We made the requested corrections.

*• P2 L62: The authors should consider to give $k_{abs,ice}$ a separate, numbered equation.*

**Author reply:** As requested, we added a separate numbered equation.

*• P5 L139: The equation for the spectral transmittance should become a separate equation instead of an in-text equation.*

**Author reply:** As requested, we made this a separate numbered equation.

*• P6 L179: do you mean Warren and Brandt (2008)?*

**Author reply:** Yes, thank you, we corrected this.

• *P7 L183: Equation (2) does not give a direct relation to calculate $k_{att}()$. The authors should consider providing a separate equation for this purpose.*

**Author reply:** As requested, we added a dedicated equation for this purpose.

• *Figure 3:*

- *The y-axes of Figs. 3a and 3b don't show the Transmittance T but its logarithm ln T please adjust the axes titles accordingly.*

  **Author reply:** The data shown in Fig. 3a and 3b are transmittance plotted on log-scale axes.

- *Legend for black line: 350 nm are stated in the text is '351 nm' a typing error?*

  **Author reply:** Yes, this typing error has been corrected.

• *P8 L220: the superscript 1 belongs to the unit of $k_{att}$, please keep it in one line.*

**Author reply:** We replaced all units with non-breaking hyphens.

• *P8 L234: reference should be to Figure 4a not Figure 5.*

**Author reply:** As noted above, we removed this figure because it is redundant with Fig. 8 and 9 (revised Fig. 10 and 12) as requested by reviewer 2.

• *P8 L236: 4 μm instead of 4 um.*

**Author reply:** We corrected the symbol as requested.

• *P9 L252: missing closing bracket after Eq. (7)*

**Author reply:** We corrected this as requested.

• *Figure 6: add y-axis title to Fig. 6b.*

**Author reply:** We added the y-axis, as requested.

• *P9 L273-274: don't split the unit on different lines.*

**Author reply:** We have replaced all units with non-breaking hyphens.

• *P10 L281: I think this should be a reference to Fig. 3b.*

**Author reply:** We corrected this reference to Fig. 3b (revised Fig. 5b)

• *P10 L287: I guess you don't mean $\omega$?*

**Author reply:** Thank you, this typo has been corrected. We intended to say "values of $\omega$ at wavelengths >800 nm".

• *P11 L310: to six previously published [...], not seven*

**Author reply:** Thank you. We added an additional published value. Seven is now correct.

• *Bibliography: please add how to access the Mätzler (2002) and Perovich (1996) references.*

**Author reply:** These URLs have been added to the references. They are:
Mätzler: https://boris.unibe.ch/146550/
Perovich: https://apps.dtic.mil/dtic/tr/fulltext/u2/a310586.pdf

• *Appendix 1, P29 L693: the minus '-' in the unit is missing*

**Author reply:** Thank you, this has been corrected.

**References**

Briegleb, B. P. and Light, B.: A Delta-Eddington Mutiple Scattering Parameterization for Solar Radiation in the Sea Ice Component of the Community Climate System Model, Technical Note, National Center for Atmospheric Research, Boulder, Colorado. http://dx.doi.org/10.5065/D6B27S71, last access: 18 February 2019, 2007.

Dadic, R., Mullen, P. C., Schneebeli, M., Brandt, R. E. and Warren, S. G.: Effects of bubbles, cracks, and volcanic tephra on the spectral albedo of bare ice near the Transantarctic Mountains: Implications for sea glaciers on Snowball Earth, J. Geophys. Res. Earth Surf., 118(3), 1658–1676, https://doi.org/10.1002/jgrf.20098, 2013.

Ertürk, H. and Howell, J. R.: Monte Carlo Methods for Radiative Transfer, in Handbook of Thermal Science and Engineering, edited by F. A. Kulacki, pp. 1–43, Springer International Publishing, Cham, https://doi.org/10.1007/978-3-319-32003-8_57-1, , 2017.

Gordon, H. R.: Ship perturbation of irradiance measurements at sea 1: Monte Carlo simulations, Appl. Opt., 24(23), 4172, https://doi.org/10.1364/AO.24.004172, 1985.

van de Hulst, H. C.: Multiple light scattering: tables, formulas, and applications, Academic Press, New York., 1980.

Leathers, R. A., Downes, T. V., Davis, C. O. and Mobley, C. D.: Monte Carlo Radiative Transfer Simulations for Ocean Optics: A Practical Guide, Memorandum, Naval Research Laboratory, Washington, D.C. https://www.oceanopticsbook.info/packages/iws_l2h/conversion/files/Leathersetal_NRL2004.pdf, last access: 11 October 2020, 2004.

Light, B., Maykut, G. A. and Grenfell, T. C.: A two-dimensional Monte Carlo model of radiative transfer in sea ice, J. Geophys. Res. Oceans, 108(C7), 3219, https://doi.org/10.1029/2002JC001513, 2003.

Liou, K.-N.: An introduction to atmospheric radiation, 2nd ed., Academic Press, Amsterdam ; Boston., 2002.

Liston, G. E. and Winther, J.-G.: Antarctic Surface and Subsurface Snow and Ice Melt Fluxes, J. Clim., 18(10), 1469–1481, https://doi.org/10.1175/JCLI3344.1, 2005.

Modest, M. F.: Radiative heat transfer, Third Edition., Academic Press, New York., 2013.

Mullen, P. C. and Warren, S. G.: Theory of the optical properties of lake ice, J. Geophys. Res. Atmospheres, 93(D7), 8403–8414, https://doi.org/10.1029/JD093iD07p08403, 1988.

Wang, L., Jacques, S. L. and Zheng, L.: MCML—Monte Carlo modeling of light transport in multi-layered tissues, Comput. Methods Programs Biomed., 47(2), 131–146, https://doi.org/10.1016/0169-2607(95)01640-F, 1995.

York, D., Evensen, N. M., Martínez, M. L. and De Basabe Delgado, J.: Unified equations for the slope, intercept, and standard errors of the best straight line, Am. J. Phys., 72(3), 367–375, https://doi.org/10.1119/1.1632486, 2004.

Zhang, X., Qiu, J., Li, X., Zhao, J. and Liu, L.: Complex refractive indices measurements of polymers in visible and near-infrared bands, Appl. Opt., 59(8), 2337, https://doi.org/10.1364/AO.383831, 2020.

**2 Response to Anonymous Referee #2**

**Author reply: Thank you for your detailed comments on our manuscript. Please note that Reviewer 1 requested a comprehensive instrumental and measurement uncertainty analysis. Please see the attached supplementary document that describes the Monte Carlo radiative transfer simulations that we performed for this purpose. Please also see our response to Reviewer 1 for more information.**

**Below, please find the original reviewer concern/question in italic typeface followed by our response in regular typeface.**

*This study presents attenuation flux coefficients using spectral irradiance measurements of bare glacial ice in western Greenland. These coefficients are compared with theory and other data sets. The authors conclude that attenuation is enhanced due to a semi-granular near-surface ice layer and by light absorbing impurities at their measurement location. As attenuation flux coefficients for glacial ice are scarce, the data set presented in this study is therefore an important addition for the scientific community. The manuscript is generally well written with clear figures. The following issues should be addressed before publication:*

**General comments:**

*1) The title and part of the introduction (line 81-91) looked a bit strange to me. I got the impression that a significant part of the study is about the ICEsat satellite, while it is only briefly discussed in Sect. 4.3. Therefore, I would suggest to shorten the title and leave out the ICEsat part and shorten the part of the introduction about ICEsat to take away the confusion.*

**Author reply:** As requested, we shortened the title and removed the ICESat part. We retained the ICESat paragraph in the Introduction to demonstrate the broader motivation for our study. We report salient results in the Results section but we removed the ICESat paragraph from the revised Discussion and only briefly touch on the relevant results. We hope this satisfies your request while maintaining the broader impact intent of the submitted version.

*2) The Kangerlussuaq region is well known to have a high LAP concentration (e.g., Wientjes et al., 2011; Tedstone et al., 2020). Therefore, it is not surprising that impurities impact the results. The authors, however, state on various occasions in the manuscript that there might be LAPs involved, almost like it is a new finding (e.g., line 331-333: "Comparison with the spectral coefficient for pure ice (Figure 4c) suggests the discrepancy we find is likely due to LAPs present in the measured volume, which appear to disproportionately enhance energy absorption near the ice surface" or on line 385-386: "This suggest light absorbing particles enhance visible light absorption and reduce optical penetration depth at our field site"). I think that the manuscript would benefit if more literature is used to determine if the results are in agreement with the observed LAP concentration for this region.*

**Author reply:** As requested, we strengthened the literature review regarding LAP concentrations. We paid special attention to this request and, as a result, the Discussion section was reorganized. Briefly, our investigation revealed that the ice we measured was optically quite pure, with equivalent black carbon concentrations comparable to pre-industrial values. We infer that the ice is most likely of Pleistocene provenance, which (for this sector of the ice sheet) is generally less dusty than Holocene ice associated with the 'dark zone'. This is discussed in detail in the revised Discussion.

*3) The authors state that no asymptotic flux attenuation coefficients are available for glacial ice (e.g., line 72-73), but Ackermann et al. (2006) (which is cited in this manuscript) reported absorption coefficients for glacial ice in Antarctica. Although it is true that Ackermann et al. (2006) measured deep glacial ice in Antarctica while the authors measured bare glacial ice in Greenland, for some cases it compares relatively well with the results presented in the manuscript, as you have shown in Fig. 9. Furthermore, Ackermann et al. (2006) show the absorption coefficient for 532 nm (Fig. 16 of that paper), which does not seem to not match the statement on line 89-91. I would like these issues discussed on the relevant places in this study or an explanation why the authors think it is not comparable (For example, on line 72-73, line 89-91, line 226-229, line 305-307, Sect. 4.3, Fig. 4b, Fig. 9).*

**Author reply:** We removed the claim "first" everywhere to avoid confusion. As with the prior comment, we paid special attention to this request and the Discussion section was reorganized to accommodate a detailed new comparison with the results of Ackermann et al. (2006).

*4) Figure 4b should be replaced by Fig. 9, as Fig 4b seems redundant. Furthermore, a discussion in more detail about Fig. 9 is desirable. On one hand it shows that the results of this study are in agreement with AMANDA 1755 m, and support the claim that impurities are an important factor (which is mentioned on various places in the manuscript, like on line 226-229 or line 282-283). However, on the other hand it shows that the difference with the pure-ice estimate of Picard et al. 2016 is very small. This is confusing for me. I also think that Fig. 4a and Fig. 8 can be merged.*

**Author reply:** As requested, we removed Fig. 4b and we point to Fig. 9 (revised Fig. 12) where we previously pointed to Fig. 4b. We added values for contaminated snow (Warren's Layer B) to revised Fig. 10, as requested by Reviewer 1. In addition, we added a detailed discussion of Fig. 9 (revised Fig. 12) that compares our findings with those of AMANDA and Picard et al. (2016).

With regard to the stated confusion, the Picard et al. (2016) pure ice estimate is in fact an estimate of ice absorptivity in the presence of trace impurities. They argue that even if conservative assumptions are made about the impurity concentration of the snowpack they measured, they would still find higher absorptivity values in the blue-green than the Warren et al. (2006) values, meaning the Warren values are unreasonably low. They conclude by recommending that their values be used in lieu of Warren's values, a recommendation that has been adopted in the literature (Tuzet et al., 2019). The point we make is that the ice we measured likely had higher LAP concentration than the snowpack they measured, yet our absorptivity values are lower than theirs. This suggests their values are biased high, and cannot be considered a better estimate of pure ice absorptivity than the values reported by Warren et al. (2006). Rather, it seems sensible to consider their values, and our values (and the AMANDA values) as estimates of in-situ ice absorptivity in the presence of realistic LAP concentrations, whereas Warren's values are reasonable estimates of pure-ice absorptivity. We conclude with this recommendation.

*5) Most Figures are barely introduced in the manuscript, while a highly detailed description is provided in the caption. I would suggest to move some of the caption to the main text.*

**Author reply:** We addressed this throughout the manuscript, as requested.

*6) It would have been better for the $\chi$ term that is introduced in Sect. 2.5 to be wavelength dependent, as attenuation in the surface layer strongly depends on wavelength (e.g., Fig. 6 and 7 of (Grenfell and Maykut, 1977)). This maybe could explain the increasing difference with wavelength for the 12 cm depth fit in Fig. 5c. As the differences are still rather small, I do not think that it is necessary to adjust the results to a wavelength dependent $\chi$, but I think that the manuscript would benefit if the authors state the uncertainty that arises because of this choice. Also, I do not understand line 195-197. Isn't $\chi$ now practically the same as i0 due to the spectral integration?*

**Author reply:** We removed the best-fit $\chi$ method altogether. Rather than compute a best-fit value, we use the 0-12 cm effective attenuation coefficient in a piece-wise Bouguer model to demonstrate the impact of the near-surface attenuation. In addition, we report values for the $i_o$ parameter using the formal definition given by Grenfell and Maykut, (1977). This parameter is a broadband value. It is simply the fraction of irradiance absorbed in the upper 10 cm of ice, weighted by the incoming irradiance (integrated over a given spectral range). This is described clearly in the revised methods. The reason we report this value is because several large-scale models apply simple radiative transfer parameterizations that use $i_o$ to put the correct amount of absorbed radiation in the upper grid cell of the vertical ice column (Briegleb and Light, 2007; Liston and Winther, 2005). With our reported values, any user will be able to compute their own $i_o$ for any other ice thickness or spectral region.

*7) Figure 7, lines 268 – 275 and Sect. 3.5, except for line 285-287, should be moved to the methods.*

**Author reply:** As requested, we moved these sections and Fig. 7 (revised Fig. 3) to the methods.

*8) Use the abbreviation 'Fig.' when referring to a figure in running text, unless it is at the beginning of the sentence.*

**Author reply:** As requested, this has been corrected throughout the manuscript.

Minor comments:
*Line 65: Change "size > wavelength" to "size larger than wavelength"*

**Author reply:** This has been corrected, as requested.

*Line 66-69: Add references to this statement.*

**Author reply:** We added the following references to this statement:

Grenfell, T. C. and Warren, S. G.: Representation of a nonspherical ice particle by a collection of independent spheres for scattering and absorption of radiation, J. Geophys. Res., 104(D24), 31697–31709, https://doi.org/10.1029/1999JD900496, 1999.

Brandt, R. E. and Warren, S. G.: Solar-heating rates and temperature profiles in Antarctic snow and ice, Journal of Glaciology, 39(131), 99–110, doi:10.3189/S0022143000015756, 1993.

Wiscombe, W. J. and Warren, S. G.: A Model for the Spectral Albedo of Snow. I: Pure Snow, J. Atmos. Sci., 37(12), 2712–2733, doi:10.1175/1520-0469(1980)037<2712:AMFTSA>2.0.CO;2, 1980.

*For all equations: Use punctuation at the end of the equation, as the equation is part of a sentence.*

**Author reply:** This has been corrected, as requested.

*Line 148-149: What do you mean with "Solid ice-equivalent values?"*

**Author reply:** "Solid ice-equivalent values" refers to normalization of the values by the ratio of solid ice density to measured (sample) density. A strict application of Bouguer's Law would have the medium be homogeneous. In the original manuscript, we fit our $k_{att}$ values to the measured depths and then converted $k_{att}$ to 'solid ice equivalent'. In the revised manuscript, we first convert the measured ice thicknesses to 'solid ice equivalent' thickness, and then fit the linear models. Note that for our medium, the density is relatively constant, and we show that our results are insensitive to this decision. This is described in the revised manuscript in Sect. 2.3.

*Line 154: g is usually assumed to be independent of wavelength. Also call it the asymmetry factor and define the single-scattering albedo.*

**Author reply:** As requested, we now call it the asymmetry factor and we defined the single scattering albedo with a dedicated equation. We retained the wavelength dependence of g.

*Eq 5: Are you sure that Schuster, 1905 is the right reference for this equation? Libois et al. (2013) and Tuzet et al. (2019, cited in this manuscript) describe it relatively well. They also use the Delta-Eddington method, which should be mentioned in the manuscript.*

**Author reply:** Schuster, (1905) is usually credited with the asymptotic two-stream solution (Mishchenko, 2013). We cite Tuzet et al. (2019) and Libois et al. (2013) in the manuscript. In the original analysis, we inverted the Eddington approximation to obtain the theoretical attenuation coefficients. In the revised analysis, we invert the delta-Eddington approximation, and we cite Joseph et al. (1976).

*Fig. 2.: Please change "Relative irradiance" to "transmittance" and add the units for the standard deviation.*

**Author reply:** As requested, we changed "relative irradiance" to "transmittance" in the figure caption. The standard deviation plot was removed; the standard deviation is now shown as shaded bounds on the plotted transmittance.

*Fig. 3: Do the authors have any idea why k_att becomes increasingly smaller and very small around 850-900 nm, which does not seem to be in agreement with Warren et al. 2006? I know that in the manuscript it is stated that beyond 700 nm the flux is small and the results become less reliable, but it seems to be odd.*

**Author reply:** The values beyond ~700 nm are inaccurate. We show them to help the reader understand why we restrict our $k_{att}$ values to the range 350–700 nm, whereas transmittance is reported to 900 nm (and is plotted in this range in revised Fig. 3b).

*Line 216: Define the albedo, as e.g. the surface reflectivity of solar radiation .*

**Author reply:** As requested, we added a definition for albedo and explained how we calculate it.

*Line 218: Please use more recent references, e.g. Gardner and Sharp (2010), and/or He and Flanner (2020).*

**Author reply:** As requested, we added both of these references. Thank you for alerting us to the review by He and Flanner (2020), it was helpful.

*Line 230: Add more references.*

**Author reply:** As requested, we added the following references:

He, C., Takano, Y., Liou, K.-N., Yang, P., Li, Q. and Chen, F.: Impact of Snow Grain Shape and Black Carbon–Snow Internal Mixing on Snow Optical Properties: Parameterizations for Climate Models, J. Climate, 30(24), 10019–10036, doi:10.1175/JCLI-D-17-0300.1, 2017.

Libois, Q., Picard, G., France, J. L., Arnaud, L., Dumont, M., Carmagnola, C. M. and King, M. D.: Influence of grain shape on light penetration in snow, The Cryosphere, 7(6), 1803–1818, doi:10.5194/tc-7-1803-2013, 2013.

Libois, Q., Picard, G., Dumont, M., Arnaud, L., Sergent, C., Pougatch, E., Sudul, M. and Vial, D.: Experimental determination of the absorption enhancement parameter of snow, Journal of Glaciology, 60(222), 714–724, doi:10.3189/2014JoG14J015, 2014.

*Line 241. Do the authors mean Eq. 17 instead of Eq. 16 of Warren et al. (2006)?*

**Author reply:** Yes, thank you, we corrected this.

*Fig.6b: Please put k_att(0-12 cm) / k_att on the y-axis and remove the legend.*

**Author reply:** Corrected, as requested.

*Line 268: I assume the authors mean with "The field measurements" your observations, and not from Grenfell and Maykut (1977)? Please clarify.*

**Author reply:** Yes, we are referring to our measurements. We clarified this in the revised text.

*Line 287: "omega > 800 nm". 800 nm does not make sense, as omega is defined in this manuscript as the single-scattering albedo.*

**Author reply:** This typo has been corrected.

*Line 323-324: "The comparison demonstrates the tremendous variation in k_att values". The term 'tremendous' is a bit overexaggerated. Besides, the differences are not that large if the absorption coefficient is compared to glacial ice or pure ice (Fig. 9).*

**Author reply:** We revised the text as follows: "The comparison demonstrates that $k_{\mathrm{att}}$ values vary by nearly two orders of magnitude at visible wavelengths due to differences in ice structure and composition"

*Line 374: Change "to modelling light attenuation in glacier ice" to "to modelling light attenuation in near-surface glacier ice".*

**Author reply:** Changed, as requested.

*Line 394-397: This is a bit vague, please reformulate.*

**Author reply:** As requested, this paragraph has been reformulated.

**References:**

Briegleb, B. P. and Light, B.: A Delta-Eddington Mutiple Scattering Parameterization for Solar Radiation in the Sea Ice Component of the Community Climate System Model, Technical Note, National Center for Atmospheric Research, Boulder, Colorado. http://dx.doi.org/10.5065/D6B27S71, last access: 18 February 2019, 2007.

Ertürk, H. and Howell, J. R.: Monte Carlo Methods for Radiative Transfer, in Handbook of Thermal Science and Engineering, edited by F. A. Kulacki, pp. 1–43, Springer International Publishing, Cham, https://doi.org/10.1007/978-3-319-32003-8_57-1, , 2017.

Gordon, H. R.: Ship perturbation of irradiance measurements at sea 1: Monte Carlo simulations, Appl. Opt., 24(23), 4172, https://doi.org/10.1364/AO.24.004172, 1985.

Grenfell, T. C. and Maykut, G. A.: The Optical Properties of Ice and Snow in the Arctic Basin*, Journal of Glaciology, 18(80), 445–463, https://doi.org/10.3189/S0022143000021122, 1977.

van de Hulst, H. C.: Multiple light scattering: tables, formulas, and applications, Academic Press, New York., 1980.

Joseph, J. H., Wiscombe, W. J. and Weinman, J. A.: The Delta-Eddington Approximation for Radiative Flux Transfer, J. Atmos. Sci., 33(12), 2452–2459, https://doi.org/10.1175/1520-0469(1976)033<2452:TDEAFR>2.0.CO;2, 1976.

Leathers, R. A., Downes, T. V., Davis, C. O. and Mobley, C. D.: Monte Carlo Radiative Transfer Simulations for Ocean Optics: A Practical Guide, Memorandum, Naval Research Laboratory, Washington, D.C. https://www.oceanopticsbook.info/packages/iws_l2h/conversion/files/Leathersetal_NRL2004.pdf, last access: 11 October 2020, 2004.

Light, B., Maykut, G. A. and Grenfell, T. C.: A two-dimensional Monte Carlo model of radiative transfer in sea ice, Journal of Geophysical Research: Oceans, 108(C7), 3219, https://doi.org/10.1029/2002JC001513, 2003.

Liou, K.-N.: An introduction to atmospheric radiation, 2nd ed., Academic Press, Amsterdam ; Boston., 2002.

Liston, G. E. and Winther, J.-G.: Antarctic Surface and Subsurface Snow and Ice Melt Fluxes, J. Climate, 18(10), 1469–1481, https://doi.org/10.1175/JCLI3344.1, 2005.

Mishchenko, M. I.: 125 years of radiative transfer: Enduring triumphs and persisting misconceptions, pp. 11–18, Dahlem Cube, Free University, Berlin, https://doi.org/10.1063/1.4804696, , 2013.

Modest, M. F.: Radiative heat transfer, Third Edition., Academic Press, New York., 2013.

Mullen, P. C. and Warren, S. G.: Theory of the optical properties of lake ice, J. Geophys. Res., 93(D7), 8403–8414, https://doi.org/10.1029/JD093iD07p08403, 1988.

Picard, G., Libois, Q. and Arnaud, L.: Refinement of the ice absorption spectrum in the visible using radiance profile measurements in Antarctic snow, The Cryosphere, 10(6), 2655–2672, https://doi.org/10.5194/tc-10-2655-2016, 2016.

Schuster, A.: Radiation through a foggy atmosphere, The Astrophysical Journal, XX1(1), 1–22, 1905.

Tuzet, F., Dumont, M., Arnaud, L., Voisin, D., Lamare, M., Larue, F., Revuelto, J. and Picard, G.: Influence of light-absorbing particles on snow spectral irradiance profiles, The Cryosphere, 13(8), 2169–2187, https://doi.org/10.5194/tc-13-2169-2019, 2019.

Wang, L., Jacques, S. L. and Zheng, L.: MCML—Monte Carlo modeling of light transport in multi-layered tissues, Computer Methods and Programs in Biomedicine, 47(2), 131–146, https://doi.org/10.1016/0169-2607(95)01640-F, 1995.

Warren, S. G., Brandt, R. E. and Grenfell, T. C.: Visible and near-ultraviolet absorption spectrum of ice from transmission of solar radiation into snow, Appl. Opt., AO, 45(21), 5320–5334, https://doi.org/10.1364/AO.45.005320, 2006.

Zhang, X., Qiu, J., Li, X., Zhao, J. and Liu, L.: Complex refractive indices measurements of polymers in visible and near-infrared bands, Appl. Opt., 59(8), 2337, https://doi.org/10.1364/AO.383831, 2020.